# Robust Adversarial Quantification via Conflict-Aware Evidential Deep Learning

**Charmaine Barker** *¶**, Daniel Bethell** *¶**, Simos Gerasimou** §¶

¶ Department of Computer Science, University of York, UK
§ Department of Elect. Eng., and Computer Science and Eng., Cyprus University of Technology, Cyprus
`{charmaine.barker, daniel.bethell, simos.gerasimou}@york.ac.uk`

## Abstract

Reliability of deep learning models is critical for deployment in high-stakes applications, where out-of-distribution or adversarial inputs may lead to detrimental outcomes. Evidential Deep Learning, an efficient paradigm for uncertainty quantification, models predictions as Dirichlet distributions of a single forward pass. However, EDL is particularly vulnerable to adversarially perturbed inputs, making overconfident errors. Conflict-aware Evidential Deep Learning (C-EDL) is a lightweight post-hoc uncertainty quantification approach that mitigates these issues, enhancing adversarial and OOD robustness without retraining. C-EDL generates diverse, task-preserving transformations per input and quantifies representational disagreement to calibrate uncertainty estimates when needed. C-EDL's conflict-aware prediction adjustment improves detection of OOD and adversarial inputs, maintaining high in-distribution accuracy and low computational overhead. Our experimental evaluation shows that C-EDL significantly outperforms state-of-the-art EDL variants and competitive baselines, achieving substantial reductions in coverage for OOD data (up to $\approx 55\%$) and adversarial data (up to $\approx 90\%$), across a range of datasets, attack types, and uncertainty metrics.

## 1 Introduction

Advances in Artificial Intelligence (AI) have led to impressive performance in diverse domains such as computer vision (Dosovitskiy et al., 2020) and natural language processing (Achiam et al., 2023). Yet in high-stakes applications, such as healthcare (Seoni et al., 2023; Loftus et al., 2022) and autonomous driving (Wang et al., 2025; 2023), ensuring AI model reliability is critical for trustworthy decision-making. In such settings, models must recognise when their predictions are uncertain, particularly in the presence of out-of-distribution (OOD) inputs, which differ significantly from the training distribution, and adversarial input that are subtly perturbed to mislead the model.

Uncertainty Quantification (UQ) is a core research area (Abdar et al., 2021) aimed at equipping models with the ability to recognise when their predictions may be unreliable. In particular, UQ seeks to capture two uncertainty types: aleatoric uncertainty, arising from inherent noise in the data, and epistemic uncertainty, stemming from limited or biased training data (Marco et al., 2025). Popular UQ approaches include Bayesian neural networks (Goan & Fookes, 2020), variational inference (Blei et al., 2017), and Laplace approximations (Fortuin, 2022), which provide principled uncertainty estimates but often at significant computational cost. More scalable approaches, like Monte Carlo Dropout (Gal & Ghahramani, 2016), deep ensembles (Lakshminarayanan et al., 2017), and test-time augmentation (Ayhan & Berens, 2018), trade accuracy or interpretability for efficiency, while hybrid approaches capture both uncertainty types simultaneously and seek appropriate performance trade-offs (Pearce et al., 2018; Angelopoulos et al., 2023). Despite their merits, these alternative approaches are too costly for use in resource-constrained environments, demanding more lightweight solutions that can support the explicit efficiency and resiliency needs of edge AI-based systems (Moskalenko et al., 2023).

Evidential Deep Learning (EDL)(Sensoy et al., 2018) is an efficient alternative UQ paradigm by modelling class probabilities with a Dirichlet distribution, enabling simultaneous capture of epistemic

---

*Equal Contribution

and aleatoric uncertainty in a single deterministic pass. This unique EDL characteristic makes it well-suited for

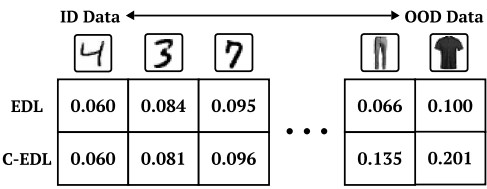

Figure 1: Uncertainty on ID (MNIST) vs. OOD (FashionMNIST). Both methods stay low on ID, but C-EDL assigns higher values to OOD where EDL remains low.

detecting OOD inputs (Figure 1) and for deployment in real-time or resource-constrained settings. However, its deterministic nature can lead to overconfident predictions under adversarial perturbations. Recent extensions to EDL (Deng et al., 2023; Qu et al., 2024; Chen et al., 2024; Yoon & Kim, 2024) aim to reduce overconfidence by encouraging alternative uncertainty estimation strategies.

While these improve OOD detection, gradient-based adversarial attacks can still mislead the model into treating inputs as in-distribution (ID), as illustrated in Figure 2. Post-hoc approaches (Yang et al., 2022) offer a promising alternative by decoupling uncertainty estimation from model training, making them more robust to such attacks and easier to deploy. For instance, Smoothed EDL (Kopetzki et al., 2021), regularises predictions against local perturbations to boost adversarial robustness. However, despite improvements, the approach continues to exhibit significant overconfidence under adversarial perturbations, indicating a clear need for more effective post-hoc defences that maintain the efficiency advantages of EDL.

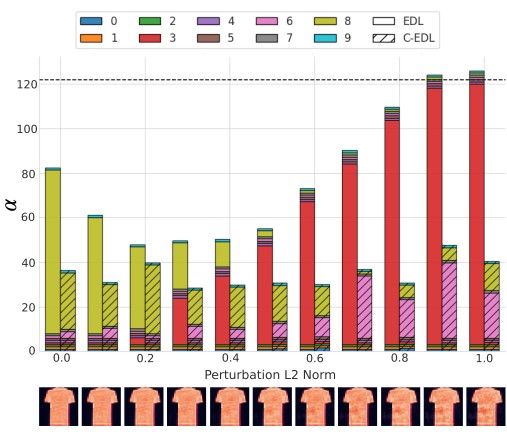

Figure 2: Evidence on a FashionMNIST OOD input under increasing L2PGD perturbations. Solid bars: EDL, gridded bars: C-EDL, dotted line: ID–OOD threshold. C-EDL stays low under attack, while EDL misclassifies as ID.

We introduce Conflict-aware Evidential Deep Learning (C-EDL), a lightweight post-hoc approach that enhances out-of-distribution (OOD) and adversarial detection by operating on pretrained EDL models. Inspired by the Dempster-Shafer theory (DST) principle that aggregating multiple sources of evidence yields more reliable beliefs (Shafer, 1976), C-EDL generates diverse views of the input and quantifies distributional disagreement to increase predictive uncertainty when appropriate. This mechanism retains in-distribution (ID) accuracy while boosting robustness to OOD inputs and adversarial perturbations. We conduct extensive experiments across various ID/OOD datasets, near- and far-OOD scenarios, and both gradient- and non-gradient-based attacks at multiple perturbation strengths. C-EDL achieves state-of-the-art results across the board, reducing OOD coverage by up to $\approx 55\%$ and adversarial coverage by up to $\approx 90\%$, while preserving ID accuracy with minimal reduction in ID coverage.

Our key contributions are: (1) the C-EDL post-hoc approach for enhancing EDL-based uncertainty estimation; (2) theoretical guarantees on the robustness of our conflict-awareness measure; and (3) a comprehensive benchmarking across diverse datasets, decision thresholds, and adversarial attacks, demonstrating that C-EDL reduces coverage for OOD and adversarial data up to 55% and 90%, respectively, significantly outperforming state-of-the-art EDL variants and competitive approaches.

## 2 RELATED WORK

**Uncertainty Quantification.** Uncertainty quantification (UQ) is essential for improving the reliability of deep learning models, particularly in safety-critical applications (He & Jiang, 2023). Epistemic uncertainty is commonly addressed by estimating distributions over predictions, with

Bayesian neural networks (Goan & Fookes, 2020), variational inference (Blei et al., 2017), and Laplace approximations (Fortuin, 2022) offering principled but computationally expensive solutions. Scalable alternatives such as Monte Carlo Dropout (Gal & Ghahramani, 2016), deep ensembles (Lakshminarayanan et al., 2017), adversarial perturbations (Schweighofer et al., 2023), and distance-aware models (Liu et al., 2020; Van Amersfoort et al., 2020) come with trade-offs in expressiveness or efficiency. Aleatoric uncertainty is typically modelled through discriminative methods predicting input-dependent variability (Kendall & Gal, 2017; Guo et al., 2017), non-parametric prediction intervals (Khosravi et al., 2010; Tagasovska & Lopez-Paz, 2019), or generative models that reconstruct data distributions (Kingma et al., 2013; Goodfellow et al., 2014). Test-time augmentation (Ayhan & Berens, 2018) offers a simple, model-agnostic alternative but does not explicitly separate uncertainty types. As both uncertainty types can coexist, it is often most effective to model them simultaneously (He & Jiang, 2023). Hybrid approaches, like ensembles with prediction intervals (Pearce et al., 2018) or conformal prediction (Angelopoulos et al., 2023; Bethell et al., 2024), albeit modelling both uncertainty types, they often incur high computational cost or require additional calibration.

**Evidential Deep Learning.** Evidential deep learning (EDL) leverages Dempster-Shafer theory (DST) (Dempster, 1968) to quantify uncertainty in a single forward pass. EDL models class probabilities using a Dirichlet distribution, where the model produces non-negative evidence for each class. The approach uses a single forward pass to estimate both epistemic and aleatoric uncertainty, making it computationally efficient. The full mathematical formulation of EDL is provided in Section 3. EDL is effective for out-of-distribution (OOD) detection, as it learns to avoid overconfident predictions on uncertain inputs, typically by thresholding uncertainty metrics such as mutual information or differential entropy. However, because it relies on a single deterministic forward pass, it cannot benefit from multiple stochastic perspectives of the same input as seen in methods like Monte Carlo Dropout or deep ensembles. As a result, if the model makes an overconfident error, it cannot correct for it, limiting its performance especially under adversarial perturbations where stronger uncertainty signals are often needed (Kopetzki et al., 2021). Figure 2 exemplifies this effect, where higher levels of L2 perturbation sees the EDL model seeing the OOD input as in-distribution (ID).

Subsequent work has extended EDL to improve OOD detection, with methods primarily proposing refinements that enhance the model's ability to recognise unfamiliar inputs (Deng et al., 2023; Qu et al., 2024; Chen et al., 2024; Yoon & Kim, 2024). However, these approaches largely retain EDL's main limitation under adversarial perturbations, as they do not address the deterministic nature of single-pass uncertainty estimation. Smoothed EDL (S-EDL)(Kopetzki et al., 2021) is one of the few methods that explicitly targets adversarial robustness by regularising predictions against local input perturbations, but it does not fully resolve the issue, particularly against stronger attacks. There is a clear need for an improvement to EDL that thoroughly addresses adversarial robustness while remaining lightweight and computationally efficient. Post-hoc approaches have also been shown to outperform approaches that modify the training process, offering developers easier integration into existing systems (Yang et al., 2022; Barker et al., 2025), and additionally allowing uncertainty improvements to be applied flexibly across a wide range of pre-trained models without retraining. To this end, the C-EDL approach introduced in this paper is designed to meet all these requirements.

## 3 PRELIMINARIES

We consider the standard supervised classification setting, aiming to learn a predictive model over a finite set of class labels. Let $\mathcal{X} \subseteq \mathbb{R}^d$ denote the input space and $\mathcal{Y} = \{1, \ldots, K\}$ the label space with $K$ classes. Given a dataset $\mathcal{D} = \{(x_i, y_i)\}_{i=1}^N$, where each pair $(x_i, y_i) \in \mathcal{X} \times \mathcal{Y}$ is sampled i.i.d. from an underlying distribution $p(x, y) = p(x) \, p(y \mid x)$, the objective is to learn the class-conditional probabilities $p(y \mid x)$. In addition to making accurate predictions, we are interested in quantifying the model's predictive uncertainty. In this work, we focus exclusively on the classification case.

EDL models class probabilities using a Dirichlet distribution, parameterised by $\boldsymbol{\alpha} = [\alpha_1, \ldots, \alpha_K]$. Given an input $x$, a neural network produces Dirichlet parameters $\alpha_k = e_k + 1$, where $e_k \geq 0$ represents the evidence collected for class $k$. The belief mass and uncertainty are derived as

$$b_k = \frac{\alpha_k - 1}{S}, \quad u = \frac{K}{S}, \tag{1}$$

where $S$ is the Dirichlet strength defined by $S = \sum_{k=1}^K \alpha_k$. A higher $S$ corresponds to stronger model confidence, leading to sharper probability distributions. The Dirichlet distribution itself is

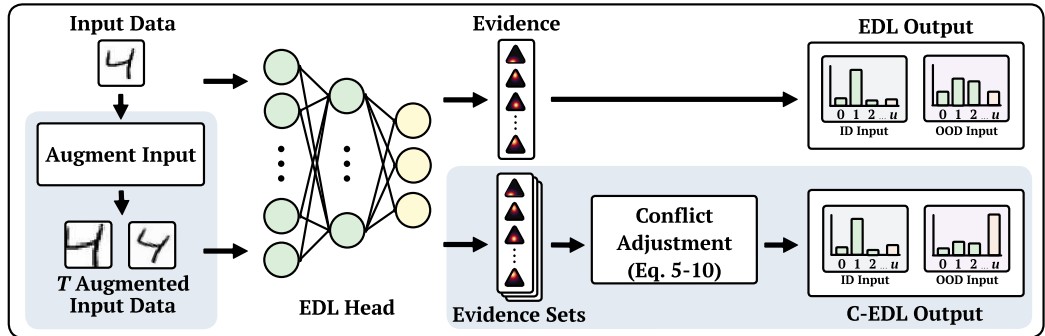

Figure 3: Overview of Conflict-aware Evidential Deep Learning (C-EDL) approach, with its key post-hoc steps that advance regular EDL highlighted in blue . For each new input, C-EDL performs $T$ metamorphic transformations, yielding a label-preserving evidence set, and then executes conflict adjustment on the accumulated evidence to calibrate the final prediction. When applied to in-distribution inputs, C-EDL closely matches the original EDL output, while given out-of-distribution inputs, C-EDL amplifies uncertainty to better reflect model disagreement.

expressed as

$$\mathrm{Dir}(p \mid \alpha) = \frac{\Gamma(S)}{\prod_{k=1}^{K} \Gamma(\alpha_k)} \prod_{k=1}^{K} p_k^{\alpha_k - 1}, \tag{2}$$

where $\Gamma(\cdot)$ is the gamma function. The expected class probability, corresponding to the mean of the Dirichlet distribution, is

$$\mathbb{E}[p_k] = \frac{\alpha_k}{S}. \tag{3}$$

While EDL allows for uncertainty estimation in a single forward pass, it struggles with out-of-distribution detection and adversarial robustness. The reliance on a single prediction can lead to overconfident outputs for unseen samples, as the method does not explicitly account for inconsistencies in evidence when the input is perturbed or transformed.

## 4 C-EDL

Our **C**onflict-aware **E**vidential **D**eep **L**earning (C-EDL) approach, whose high-level workflow is shown in Figure 3, improves robustness to OOD and adversarially attacked inputs. Motivated by the DST principle that multiple sources of evidence yield more reliable beliefs (Shafer, 1976), C-EDL generates diverse views of each input through label-preserving metamorphic transformations, coalesces the resulting evidence sets, and quantifies their discernment. When conflict is detected across these views, C-EDL reduces the overall evidence to signify greater uncertainty, resulting in better detection of OOD and adversarially attacked inputs without affecting the ID data detection and accuracy.

### 4.1 INPUT AUGMENTATION AND EVIDENCE SET GENERATION

To generate a diverse set of evidence for a given input, C-EDL applies a set of $T$ metamorphic transformations $\{\tau_1, \ldots, \tau_T\}$ to an input instance $x$, where each transformation satisfies the label-preserving constraint $f^*(\tau_t(x)) = f^*(x) \ \forall 1 \leq t \leq T$. Each $\tau_t(x)$ serves as a distinct but semantically equivalent view of the original input. Since each $\tau_t$ is label-preserving, a model that has learned robust decision features should produce consistent evidence across $\{\tau_t(x)\}_{t=1}^{T}$. Since the EDL Dirichlet output already encodes both epistemic and aleatoric uncertainty, cross-view disagreement is used as a post-hoc stability check that primarily indicates instability in the epistemic component (i.e., brittle or insufficient knowledge) while preserving the underlying uncertainty representation. Any transformation-induced stochasticity is kept small by bounding augmentation intensity, see Table 11. Although these transformations induce only small changes in input space, they can yield substantially

different responses in the model's internal feature representations due to its sensitivity to local structure (Goodfellow et al., 2016). This property enables C-EDL to probe the stability of the model's beliefs under controlled, task-instrumented perturbations, offering a principled way to elicit and assess representational discernment and disagreement. These transformations span an equivalence class of inputs that share the same label under the ground-truth function $f^*$, but differ in their statistics at the input level; for instance, in the case of images, the difference would be at the pixel-level. Since the pretrained evidential model may be susceptible to spurious features or lack generalisability capabilities to transformation invariance, its outputs across $\{\tau_t(x)\}_{t=1}^T$ can differ, providing a signal of uncertainty that we later quantify through conflict analysis and adjustment.

Each transformed instance $\tau_t(x)$ is independently passed through a pretrained evidential model, which produces a corresponding Dirichlet vector $\alpha^{(t)} = (\alpha_1^{(t)}, \ldots, \alpha_K^{(t)})$, where each $\alpha_k^{(t)}$ encodes the strength of belief assigned to class $k$. This results in a set of $T$ evidence sets $\mathcal{A} = \{\alpha^{(1)}, \alpha^{(2)}, \ldots, \alpha^{(T)}\}$, which captures the model's output variability across representational shifts in the input data.

## 4.2 CONFLICT ADJUSTMENT

After collecting diverse evidence sets $\mathcal{A}$, we quantify disagreement across these views through two complementary measures of conflict: intra-class variability and inter-class contradiction. Using disagreement across multiple views as a reliability cue is conceptually aligned with prior work in multi-view learning (Xu et al., 2024). However, the ECML approach is orthogonal as it assumes distinct input views (e.g., an image and its caption), trains a separate evidential model per view, and uses conflict among confident but divergent view-specific predictions during training. In contrast, C-EDL is post-hoc and needs a single modality, inducing views at test time via label-preserving transformations and measuring conflict in evidential space. Intra-class variability captures how much the evidence for each class fluctuates across the applied transformations. For each class $k$, the standard deviation and mean of the Dirichlet parameters $\alpha_k^{(t)}$ across the transformations $T$. The coefficient of this is then averaged across all classes:

$$C_{\text{intra}} = \frac{1}{K} \sum_{k=1}^K \frac{\sigma(\{\alpha_k^{(t)}\}_{t=1}^T)}{\mu(\{\alpha_k^{(t)}\}_{t=1}^T) + \epsilon}, \tag{4}$$

where $\sigma(\cdot)$ and $\mu(\cdot)$ denote the standard deviation and mean respectively, and $\epsilon$ is a small positive constant used for numerical stability. In particular, $\epsilon$ prevents division by zero by ensuring the denominator is strictly positive, which is used in Appendix A when establishing boundedness and continuity of $C_{\text{intra}}$. The rationale underpinning $C_{intra}$ is that its value increases when the model assigns inconsistent beliefs to the same class across the performed transformations.

Inter-class conflict measures instances where the model supports competing classes (e.g., two or more classes have equally high probability/evidence), highlighting cases where the model is unsure about its prediction. For each set of Dirichlet parameters $\alpha_k^{(t)}$, the pairwise contradictions between classes $k$ and $j$ are computed, highlighting cases where both classes are supported with high evidence. This is formalised as:

$$C_{\text{inter}} = \frac{1}{T} \sum_{t=1}^T \left( 1 - \exp\left( -\beta \sum_{k=1}^K \sum_{j=k+1}^K \left( \frac{\min(\alpha_k^{(t)}, \alpha_j^{(t)})}{\max(\alpha_k^{(t)}, \alpha_j^{(t)})} \times \frac{\min(\alpha_k^{(t)}, \alpha_j^{(t)})}{\sum_{k=1}^K \alpha_k^{(t)}} \times 2 \right)^2 \right) \right), \tag{5}$$

where $\beta > 0$ is a scaling parameter that adjusts the sharpness of the penalty assigned to the contradiction, and the final multiplication by 2 ensures the combined term is bounded within $[0, 1]$. We design $C_{\text{inter}}$ to be symmetric, bounded, and to increase only when competing classes are both (i) similarly supported and (ii) supported with non-trivial evidence. The ratio term captures the relative balance between classes, while the strength normalisation penalises contradictions arising from uniformly low evidence. Simpler formulations can overstate conflict under uniformly low evidence; Eq. 5 is a minimal construction that avoids this edge case while remaining tractable for Theorem 1.

To obtain the final measurement of conflict, both terms are combined into a single total score $C$:

$$C = C_{\text{inter}} + C_{\text{intra}} - C_{\text{inter}} C_{\text{intra}} - \lambda (C_{\text{inter}} - C_{\text{intra}})^2 \tag{6}$$

using the inclusion-exclusion principle, which ensures a combined measure of conflict by accounting for overlap between $C_{intra}$ and $C_{inter}$, where $\lambda \in [0,1]$ controls the penalisation of asymmetric disagreement. This formulation ensures $C \in (0,1]$ tends towards $0$ if and only if all transformations produce identical Dirichlet parameters concentrated on a single class, and increases monotonically with either source of conflict.

**Theorem 1.** *The conflict measure $C$ is bounded between $(0,1]$, tends towards $0$ if and only if all transformations produce identical Dirichlet parameters concentrated on a single class, and monotonically non-decreasing with increasing intra and inter-class conflict with $\lambda \in [0, \frac{1}{2}]$.*

See the corresponding proof in Appendix A. This conflict score $C$ serves as the basis for the post-hoc uncertainty augmentation and evidence reduction of C-EDL. Firstly, the Dirichlet parameters across all transformations are aggregated:

$$\bar{\alpha}_k = \frac{1}{T} \sum_{t=1}^{T} \alpha_k^{(t)}, \quad \forall k \in \{1, ..., K\}. \tag{7}$$

C-EDL applies an exponential decay to each element of the aggregated Dirichlet parameters, specifically, each parameter $\alpha_k$ is scaled to reduce overconfident predictions when high conflict is detected and preserve predictions when low conflict is detected:

$$\tilde{\alpha}_k = \bar{\alpha}_k \times \exp(-\delta C). \tag{8}$$

where $\delta > 0$ is a tunable hyperparameter controlling the sensitivity of the adjustment. This decay operation ensures that the shape of the original distribution remains unchanged, while the magnitude of the evidence is reduced proportionally to the conflict. Doing so retains the model's most likely prediction but expresses reduced certainty. The remaining EDL calculations, specifically the Dirichlet strength, belief per class, uncertainty mass, and expected probabilities, are only modified to use the reduced Dirichlet parameters:

$$\tilde{S} = \sum_{k=1}^{K} \tilde{\alpha}_k, \quad \tilde{b}_k = \frac{\tilde{\alpha}_k - 1}{\tilde{S}}, \quad \tilde{u} = \frac{K}{\tilde{S}}, \quad \mathbb{E}[\tilde{p}_k] = \frac{\tilde{\alpha}_k}{\tilde{S}}. \tag{9}$$

In cases where the conflict $C$ is high, the total Dirichlet strength $\tilde{S}$ is reduced and, consequently, the uncertainty mass $\tilde{u}$ is amplified. In contrast, when conflict is low, the total Dirichlet strength $\tilde{S}$ resembles the original strength $S$ and uncertainty is therefore minimally affected.

## 5 EVALUATION

We evaluate C-EDL in a comprehensive series of experiments comparing it against state-of-the-art EDL-based and other competitive UQ approaches over 10 independent runs. Our evaluation focuses on both performance and uncertainty estimates produced per approach for OOD and adversarially attacked data. The C-EDL code and replication package are available at our open-source repository [1].

**Comparative Approaches.** We compare C-EDL against Posterior Networks (Charpentier et al., 2020), Evidential Deep Learning (EDL) (Sensoy et al., 2018), Fisher Information-based EDL ($\mathcal{I}$-EDL) (Deng et al., 2023), Smoothed EDL (S-EDL) (Kopetzki et al., 2021), Hyper-Opinion EDL (H-EDL) (Qu et al., 2024) Relaxed EDL (R-EDL) (Chen et al., 2024), and Density-Aware EDL (DA-EDL) (Yoon & Kim, 2024), to represent a range of approaches in EDL and UQ that allow for a fair comparison.

**Datasets.** Adopting the procedure on EDL-based evaluation from recent research (Deng et al., 2023; Chen et al., 2024), we evaluate all approaches on the MNIST (LeCun et al., 1998), FashionMNIST (Xiao et al., 2017), KMNIST (Clanuwat et al., 2018), EMNIST (Cohen et al., 2017), CIFAR10 (Krizhevsky et al., 2009), CIFAR100 (Krizhevsky et al., 2009), SVHN (Netzer et al., 2011), Oxford Flowers (Netzer et al., 2011), Deep Weeds (Olsen et al., 2019), Tiny-ImageNet (Le & Yang, 2015), and CUB (Welinder et al., 2010) datasets which were selected to cover a diverse set of domains

---

[1]Our open-source repository is available at: https://github.com/team-daniel/cedl

Figure 4: Visualised adversarial AUROC plots including the binary decision threshold for OOD/Adv rejection, for comparative methods where the ID dataset is MNIST and the OOD dataset is Fashion-MNIST. Full plots are in Appendix B.1.

and challenges. In the following experiments, near-OOD datasets are those that share some degree of class overlap with the ID dataset (Yang et al., 2022).

Alongside C-EDL, we evaluate three variants to isolate component effects: EDL++, which omits conflict adjustment and averages Dirichlet parameters after input transformations; and MC versions of both EDL++ and C-EDL, denoted EDL++ (MC) and C-EDL (MC), which replace metamorphic transformations with Monte Carlo Dropout for diverse evidence. Full training, hyperparameters, attack setups, and datasets are detailed in Appendix C. We also compare C-EDL to other UQ methods (Table 4) and assess Tiny-ImageNet with CUB few-shot on ResNet50 (Table 5, Appendix B.1). Training and inference time results are in Appendix B.4.

## 5.1 Core Results

For our core set of experiments, we evaluate the accuracy, ID coverage, OOD detection, including both near- and far-OOD settings, and adversarial robustness of both C-EDL and its variants and comparative baselines across a diverse suite of ID/OOD dataset pairs, in order to assess overall reliability under distributional shift and attack. These results are summarised in Table 1 and further analysis can be found in Appendix B.1. We report *both* coverage and AUROC: coverage gives the acceptance rate at a fixed abstention threshold (a single operating point), while AUROC summarises separability across all thresholds (Figure 4).

Firstly, across all datasets, every approach (including C-EDL and its variants) maintains near-ceiling accuracy across all datasets (e.g., 95-99%), showing that none of the UQ approaches compromise classification performance on clean ID data. This key outcome provides concrete evidence reassuring that any observed gains in robustness through using C-EDL are not due to underlying degradation in classification performance in ID data.

In terms of ID coverage (the proportion of ID inputs retained after abstention), approaches such as EDL, H-EDL, and R-EDL attain higher coverage ($< 96\%$ on MNIST $\rightarrow$ FashionMNIST), but this comes at a cost of worse or only a marginal improvement in OOD and adversarial coverage. This shows that there still remains a large overconfidence in predictions. In contrast, C-EDL and its variants, offer a significantly better trade-off. More specifically, C-EDL achieves a substantially lower OOD and adversarial coverage, often halving, tripling or more compared to state-of-the-art approaches, with only a marginal reduction in ID coverage. For example, on the MNIST $\rightarrow$ FashionMNIST datasets pair, EDL achieves an adversarial coverage of $52.21\% \pm 9.49$ whilst C-EDL (Meta) reduces this substantially to $15.51\% \pm 6.09$. On more complex datasets, such as the CIFAR10 $\rightarrow$ SVHN pairing, EDL achieves an adversarial coverage of $20.00\% \pm 6.89$ whilst C-EDL (Meta) reduces this substantially to $1.25\% \pm 0.56$. To better visualise the trade-off and the superior performance of the C-EDL variants, Figure 4 provides adversarial AUROC plots for the adversarially attacked FashionMNIST dataset.

The EDL++ variants isolate the effect of the metamorphic and MC Dropout transformations with the conflict-aware adjustment. While the EDL++ variants can reduce OOD and adversarial coverage to some degree, the C-EDL variants reduce them even further. For example, on the CIFAR10 $\rightarrow$ SVHN pairing, EDL++ (Meta) achieves OOD coverage $6.59\% \pm 1.17$ and adversarial coverage $2.35\% \pm 0.96$ whilst C-EDL (Meta) achieves OOD coverage $4.69\% \pm 1.00$ and adversarial coverage $1.25\% \pm 0.56$. The improved performance of C-EDL across nearly every setting corroborates that it

---

[2]Details of the maximum L2PGD perturbation for each dataset are discussed in Appendix C.6

Table 1: The accuracy, OOD detection, and adversarial attack detection performance of the comparative approaches with a variety of ID and OOD datasets in order of dataset difficulty. The adversarial attack is an L2PGD attack [2] **Highlighted** cells denote the best performance for each metric. * indicates datasets classed as Near-OOD.

| | Comparative Methods | | | | | | | Ablated C-EDL Methods | | | Proposed C-EDL |
|---|---|---|---|---|---|---|---|---|---|---|---|
| | Posterior Network | EDL | $\mathcal{I}$-EDL | S-EDL | H-EDL | R-EDL | DA-EDL | EDL++ (MC) | C-EDL (MC) | EDL++ (Meta) | C-EDL (Meta) |
| MNIST → FashionMNIST | | | | | | | | | | | |
| ID Acc (%) ↑ | 99.96 ± 0.01 | 99.96 ± 0.02 | 99.95 ± 0.02 | 99.95 ± 0.02 | 99.96 ± 0.02 | 99.96 ± 0.01 | 99.89 ± 0.05 | 99.97 ± 0.01 | **99.98 ± 0.02** | 99.97 ± 0.02 | 99.96 ± 0.01 |
| ID Cov (%) ↑ | 94.72 ± 0.96 | **96.61 ± 0.57** | 96.08 ± 0.67 | 96.06 ± 0.72 | 96.18 ± 0.92 | 96.27 ± 0.48 | 95.16 ± 0.76 | 93.35 ± 1.28 | 92.14 ± 1.43 | 94.93 ± 0.60 | 94.18 ± 1.03 |
| OOD Cov (%) ↓ | 3.55 ± 0.67 | 2.52 ± 0.68 | 2.74 ± 0.96 | 2.41 ± 0.66 | 2.28 ± 0.98 | 2.34 ± 0.86 | 2.98 ± 2.02 | 7.55 ± 1.33 | **1.77 ± 0.63** | 1.96 ± 0.59 | 2.00 ± 0.80 |
| Adv Cov (%) ↓ | 61.14 ± 9.38 | 52.21 ± 9.49 | 47.58 ± 8.79 | 48.80 ± 8.75 | 50.40 ± 14.60 | 49.05 ± 11.90 | 28.74 ± 7.93 | 38.81 ± 4.23 | 21.62 ± 4.89 | 16.43 ± 5.52 | **15.51 ± 6.09** |
| MNIST → KMNIST | | | | | | | | | | | |
| ID Acc (%) ↑ | 99.97 ± 0.01 | 99.97 ± 0.01 | 99.96 ± 0.01 | 99.97 ± 0.01 | 99.97 ± 0.01 | 99.96 ± 0.01 | 99.90 ± 0.03 | **99.98 ± 0.01** | 99.98 ± 0.02 | **99.98 ± 0.01** | 99.98 ± 0.01 |
| ID Cov (%) ↑ | 94.72 ± 0.51 | 95.42 ± 0.62 | 95.47 ± 0.55 | 95.08 ± 0.78 | **95.59 ± 0.48** | 95.32 ± 0.53 | 94.12 ± 1.20 | 91.94 ± 0.99 | 90.20 ± 1.66 | 93.73 ± 0.82 | 92.42 ± 0.65 |
| OOD Cov (%) ↓ | 3.78 ± 0.81 | 3.23 ± 0.76 | 3.54 ± 0.74 | 3.16 ± 0.56 | 3.33 ± 0.59 | 3.21 ± 0.37 | 3.37 ± 2.35 | 11.67 ± 1.19 | 3.69 ± 1.01 | 2.21 ± 0.58 | **1.90 ± 0.32** |
| Adv Cov (%) ↓ | 23.91 ± 3.70 | 20.88 ± 5.95 | 19.57 ± 4.27 | 14.96 ± 4.11 | 20.03 ± 2.91 | 16.40 ± 4.49 | 10.06 ± 4.21 | 27.79 ± 2.66 | 12.40 ± 2.09 | 4.20 ± 1.93 | **3.01 ± 0.94** |
| MNIST → EMNIST* | | | | | | | | | | | |
| ID Acc (%) ↑ | 99.98 ± 0.01 | 99.98 ± 0.01 | **99.99 ± 0.01** | **99.99 ± 0.01** | 99.98 ± 0.01 | 99.98 ± 0.02 | 99.92 ± 0.04 | **99.99 ± 0.01** | **99.99 ± 0.01** | **99.99 ± 0.01** | 99.98 ± 0.01 |
| ID Cov (%) ↑ | 93.14 ± 0.87 | 92.73 ± 0.94 | 93.17 ± 0.46 | 92.65 ± 1.10 | 93.18 ± 1.16 | **93.27 ± 0.82** | 90.53 ± 1.30 | 86.11 ± 2.86 | 82.75 ± 2.96 | 90.65 ± 1.34 | 89.89 ± 1.44 |
| OOD Cov (%) ↓ | 14.96 ± 1.02 | 10.74 ± 1.34 | 10.62 ± 0.93 | 10.31 ± 1.33 | 11.29 ± 1.59 | 10.06 ± 1.02 | 14.14 ± 8.47 | 17.16 ± 1.34 | **6.83 ± 1.37** | 8.88 ± 1.04 | 8.93 ± 0.75 |
| Adv Cov (%) ↓ | 15.76 ± 3.95 | 7.81 ± 3.53 | 6.53 ± 1.72 | 6.60 ± 2.02 | 9.05 ± 4.11 | 6.40 ± 2.96 | 12.30 ± 10.93 | 22.63 ± 2.74 | 7.14 ± 1.61 | 1.47 ± 0.59 | **1.41 ± 0.40** |
| CIFAR10 → SVHN | | | | | | | | | | | |
| ID Acc (%) ↑ | 95.63 ± 0.72 | 95.88 ± 0.54 | 96.19 ± 0.70 | 96.07 ± 0.69 | 96.09 ± 0.55 | 96.82 ± 0.54 | 91.25 ± 0.96 | 97.39 ± 0.53 | **98.40 ± 0.39** | 97.29 ± 0.31 | 97.98 ± 0.42 |
| ID Cov (%) ↑ | 65.74 ± 3.11 | **67.34 ± 3.15** | 66.57 ± 2.41 | 65.57 ± 3.36 | 66.74 ± 2.75 | 62.59 ± 2.41 | 43.83 ± 16.97 | 56.65 ± 3.01 | 47.04 ± 3.64 | 60.53 ± 2.32 | 54.70 ± 2.11 |
| OOD Cov (%) ↓ | 11.82 ± 2.52 | 10.91 ± 2.23 | 9.68 ± 1.84 | 10.40 ± 1.94 | 9.84 ± 1.63 | 8.92 ± 1.76 | 20.40 ± 9.25 | 12.01 ± 1.88 | 6.66 ± 1.06 | 6.59 ± 1.17 | **4.69 ± 1.00** |
| Adv Cov (%) ↓ | 15.57 ± 6.19 | 20.00 ± 6.89 | 19.25 ± 6.37 | 3.32 ± 1.46 | 19.64 ± 4.86 | 14.61 ± 6.79 | 21.48 ± 8.98 | 15.22 ± 2.32 | 9.39 ± 1.18 | 2.35 ± 0.96 | **1.25 ± 0.56** |
| CIFAR10 → CIFAR100* | | | | | | | | | | | |
| ID Acc (%) ↑ | 96.17 ± 0.36 | 96.66 ± 0.70 | 96.67 ± 0.45 | 96.69 ± 0.54 | 96.76 ± 0.46 | 97.40 ± 0.41 | 89.28 ± 0.81 | 97.87 ± 0.28 | **98.64 ± 0.40** | 97.51 ± 0.40 | 98.22 ± 0.27 |
| ID Cov (%) ↑ | 62.91 ± 2.71 | 63.33 ± 3.89 | 63.84 ± 2.75 | 63.13 ± 2.35 | 62.25 ± 2.11 | **66.98 ± 3.51** | 60.21 ± 2.39 | 54.50 ± 2.37 | 45.02 ± 3.37 | 59.47 ± 2.31 | 53.18 ± 2.31 |
| OOD Cov (%) ↓ | 17.73 ± 2.42 | 17.24 ± 4.01 | 17.73 ± 2.72 | 17.48 ± 2.56 | 16.12 ± 2.01 | 14.46 ± 2.02 | 33.85 ± 4.14 | 19.50 ± 1.74 | 11.44 ± 1.81 | 13.34 ± 1.83 | **9.61 ± 1.26** |
| Adv Cov (%) ↓ | 12.30 ± 2.71 | 14.02 ± 5.06 | 14.10 ± 4.05 | 7.73 ± 1.88 | 12.73 ± 3.17 | 9.40 ± 2.49 | 31.88 ± 4.39 | 16.32 ± 2.01 | 7.20 ± 1.70 | 5.46 ± 1.20 | **3.17 ± 0.51** |
| Oxford Flowers (low-shot) → Deep Weeds | | | | | | | | | | | |
| ID Acc (%) ↑ | 98.59 ± 0.38 | 99.00 ± 0.57 | 98.89 ± 0.49 | 99.09 ± 0.50 | 98.97 ± 0.46 | 99.78 ± 0.15 | 92.38 ± 3.17 | 99.72 ± 0.25 | **100.00 ± 0.00** | 99.20 ± 0.51 | 99.59 ± 0.58 |
| ID Cov (%) ↑ | 69.86 ± 2.32 | 63.67 ± 4.40 | 60.73 ± 8.23 | 61.70 ± 5.89 | 63.17 ± 6.37 | 64.78 ± 2.51 | **71.40 ± 5.39** | 54.20 ± 8.25 | 42.33 ± 5.71 | 51.73 ± 4.12 | 47.50 ± 7.44 |
| OOD Cov (%) ↓ | 5.27 ± 1.26 | 8.87 ± 4.80 | 8.17 ± 1.78 | 8.13 ± 1.73 | 6.01 ± 1.34 | 2.34 ± 1.64 | 3.76 ± 2.28 | 6.00 ± 2.23 | **1.02 ± 0.41** | 5.61 ± 2.80 | 1.78 ± 1.92 |
| Adv Cov (%) ↓ | 8.24 ± 2.53 | 13.57 ± 5.92 | 12.74 ± 3.86 | 11.56 ± 3.65 | 8.61 ± 3.26 | 1.25 ± 0.78 | 4.69 ± 2.90 | 5.55 ± 1.84 | **1.14 ± 0.38** | 6.94 ± 2.81 | 1.93 ± 2.19 |

Figure 5: Adversarial coverage (circle size) compared to mean difference between the computed abstention metric (Δ, circle colour; Table 3) for all approaches on all evaluated datasets. Positive values indicate predictions tend to be above the threshold, while negative values indicate predictions fall below the threshold (as expected for adversarially attacked data). Smaller circle and more negative Δ values (darker green) are ideal, reflecting stronger confidence that the input is adversarial.

is not just the diversity of evidence that matters. Instead, the conflict-aware adjustment of C-EDL that quantifies the disagreement and uses this information to reduce the magnitude of belief is also highly important.

These core results also reinforce the strength of post-hoc uncertainty calibration strategies in comparison to in-training strategies. Among all evaluated approaches, post-hoc approaches (S-EDL and our C-EDL variants) outperform those that modify the training process (e.g., DA-EDL, H-EDL, R-EDL), highlighting the advantages of decoupling prediction and uncertainty estimation.

The comparison between C-EDL (MC) and C-EDL (Meta) shows that metamorphic transformations consistently outperform Monte Carlo sampling. While both generate diverse evidence to assess belief stability, the structured and semantically controlled perturbations in C-EDL (Meta) yield better OOD and adversarial detection. This supports the intuition that task-preserving augmentations form a more principled basis for detecting epistemic uncertainty, providing concrete evidence for the benefit of C-EDL's input augmentation and evidence set generation step (Section 4.1).

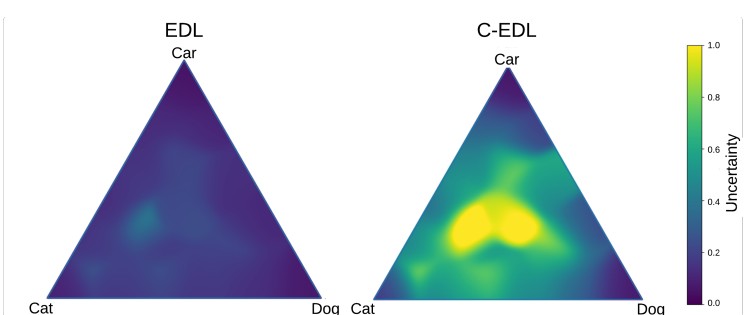

Figure 6: 3-class simplex visualisation for the CIFAR-10 dataset comparing EDL and C-EDL.

To further highlight the quality of abstention decisions, particularly in the adversarial context, we examine the computed scoring metrics ($\Delta$) against the coverage for each approach (Figure 5 and Table 3). These metrics combined represent the mean difference between the model's uncertainty scores and the ID-OOD decision threshold; positive for retained (ID) samples, negative for abstained (OOD or adversarial) samples. A large positive $\Delta$ for ID data and large negative $\Delta$ for adversarial or OOD inputs indicate well-separated and calibrated decisions. As shown in Figure 5, C-EDL (Meta) consistently shows the most desirable profile. For instance, on MNIST $\rightarrow$ FashionMNIST, its adversarial $\Delta$ is $-5.50 \pm 1.24$, compared to $-3.55 \pm 1.46$ and $-2.89 \pm 1.13$ for $\mathcal{I}$-EDL and EDL, respectively, showing a much stronger rejection on adversarial examples. These results yield strong evidence highlighting that C-EDL is not only effective in reducing adversarial coverage, but also reliably confident when rejecting uncertain inputs. Extended analysis of the core results can be found in Appendix B.1, and ablation results of the hyperparameters introduced by the C-EDL and its variants can be found in Appendix B.4.

To visualise how C-EDL improves upon quantifying uncertainty over the baseline, we also provide a 3-class simplex in Figure 6. Whilst EDL produces strong uncertainty regions across the simplex, C-EDL further boost these uncertain regions and in the centre outputs uncertainty close to 1 (full uncertainty). We also observe that C-EDL has higher uncertainty when there is equal disagreement between classes (directly our $C_{inter}$ conflict). However, C-EDL preserves the low uncertainty regions in the corners of the simplex. Evidently, C-EDL preserves low uncertainty in high-evidence and low-conflict regions just like EDL, but increases uncertainty in uncertain and high-conflict regions, unlike EDL. This insight provides an intuitive illustration of how conflict-aware adjustment yields more faithful uncertainty estimates than the baseline.

## 5.2 ADVERSARIAL ATTACK ANALYSIS

We evaluate adversarial coverage for the investigated approaches with increasing perturbation strengths $\epsilon$ for three distinct gradient (L2PGD and FGSM) and non-gradient (Salt-and-Pepper noise) adversarial attack types in Figure 7. Across all attack types and $\epsilon$ values, C-EDL (Meta) consistently achieves the lowest adversarial coverage, confirming robustness to multiple attacks and attack types.

For the most challenging gradient-based attack (L2PGD - left), adversarial coverage for EDL rises sharply to nearly 70% at $\epsilon = 1.0$, while C-EDL (Meta) remains below 20%, signifying a considerable improvement. The gap is similarly large for C-EDL (MC). For a weaker gradient-based attack (FGSM - middle), adversarial coverage is lower on average and, as expected, multiple comparative approaches struggle as $\epsilon$ increases. Both C-EDL variants remain very close to 0%. Finally, for a non-gradient-based attack (Salt-and-Pepper noise - right), all comparative approaches struggle at smaller $\epsilon$, which is to be expected with non-gradient-based attacks. All C-EDL variants maintain minimal coverage throughout. This result shows that C-EDL's conflict-aware strategy generalises across fundamentally different types of adversarial attacks, not just those tailored for gradient manipulation. Detailed results for the performance of all approaches under each attack are provided in Appendix B.2.

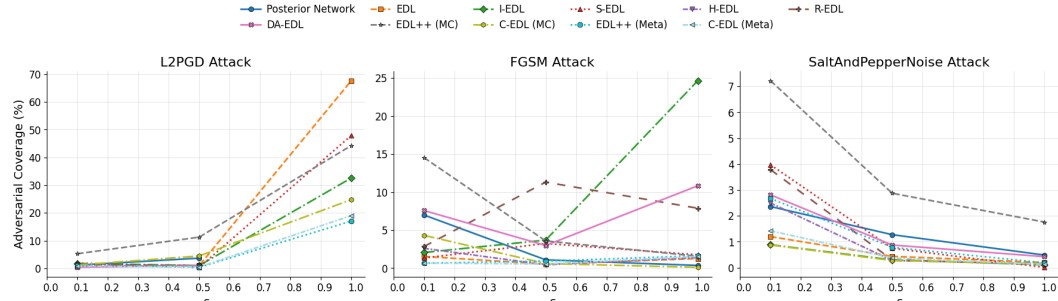

Figure 7: Adversarial coverage (%) across varying perturbation strengths ($\epsilon$) for three attack types (L2PGD, FGSM, and Salt and Pepper noise), where the ID dataset is MNIST, and the OOD dataset is FashionMNIST. Lower coverage indicates better robustness.

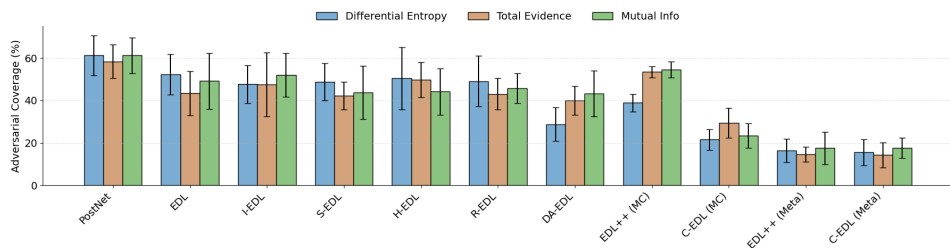

Figure 8: Adversarial coverage (%) for different ID-OOD threshold metrics where the ID dataset is MNIST, the OOD dataset is FashionMNIST, and the adversarial attack is L2PGD ($\epsilon = 1.0$).

## 5.3 THRESHOLD ANALYSIS

Similar to (Sensoy et al., 2018; Wang et al., 2024; Deng et al., 2023), we evaluate the performance of the investigated approaches under varying ID-OOD decision threshold methods, i.e., differential entropy, total evidence and mutual information. Details on each threshold are provided in Appendix C.2. The adversarial coverage per approach for each decision threshold is shown in Figure 8 and the results obtained reveal interesting trends. Firstly, most of the majority (e.g., Posterior Network, EDL, $\mathcal{I}$-EDL) have limited sensitivity to the decision threshold, and further perform poorly, consistently predicting on more than half of the adversarial data. In contrast, C-EDL and its variants achieve a significantly lower coverage across all three thresholds. These results demonstrate that the performance of C-EDL is robust to the choice of decision threshold methods. Further analysis can be found in Appendix B.3.

## 6 CONCLUSION AND FUTURE WORK

We introduced Conflict-aware Evidential Deep Learning (C-EDL), a post-hoc uncertainty quantification method that augments any pre-trained EDL classifier with principled, transformation-driven conflict analysis. By generating label-preserving metamorphic variants of each input, quantifying intra- and inter-class disagreement, and scaling evidence accordingly, C-EDL reliably detects OOD and adversarial inputs, thereby improving robustness. Experiments on diverse datasets show that C-EDL reduces prediction coverage on adversarial inputs by up to six-fold over state-of-the-art methods. Despite the added transformations, inference overhead remains negligible, making C-EDL lightweight and practical for deployment. These findings highlight C-EDL as a generalisable and robust solution against OOD and adversarial data across a variety of attacks and decision thresholds, with future work aimed at extending it to detection tasks and minimising augmentation requirements.

ACKNOWLEDGMENTS

This work was supported the European Union's Horizon Europe research and innovation programme under grant agreement 101168067 (GuardAI - Enhancing Robustness and Security of Edge AI Systems for Safety-Critical Applications).

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

# A    THEORETICAL ANALYSIS

This appendix provides the proofs to complement Theorem 1 presented in Section 4. More specifically, we provide proofs regarding the $C$, $C_{intra}$, and $C_{inter}$ bounds, and the quality of the conflict measurement $C$ in terms of monotonicity and how it behaves in specific scenarios.

## A.1    THEOREM 1

**Theorem 1.** The conflict measure $C$ is bounded between $(0, 1]$, tends towards $0$ if and only if all transformations produce identical Dirichlet parameters concentrated on a single class, and is monotonically non-decreasing with increasing intra and inter-class conflict with $\lambda \in [0, \frac{1}{2}]$.

*Proof.* We prove this in three parts: (1) $C$ is bounded within $(0, 1]$; (2) $C \to 0$ in the case of identical Dirichlet parameters; and (3) $C$ is monotonically non-decreasing with increasing $C_{intra}$ and $C_{inter}$.

Firstly, to prove that $C$ is bound within $(0, 1]$ we recall the $C_{intra}$ and $C_{inter}$ definitions from Equations equation 4 and equation 5, respectively:

$$C_{\text{intra}} = \frac{1}{K} \sum_{k=1}^{K} \frac{\sigma(\{\alpha_k^{(t)}\}_{t=1}^T)}{\mu(\{\alpha_k^{(t)}\}_{t=1}^T) + \epsilon}, \tag{10}$$

$$C_{\text{inter}} = \frac{1}{T} \sum_{t=1}^{T} \left( 1 - \exp\left( -\beta \sum_{k=1}^{K} \sum_{j=k+1}^{K} \left( \frac{\min(\alpha_k^{(t)}, \alpha_j^{(t)})}{\max(\alpha_k^{(t)}, \alpha_j^{(t)})} \times \frac{\min(\alpha_k^{(t)}, \alpha_j^{(t)})}{\sum_{k=1}^{K} \alpha_k^{(t)}} \times 2 \right)^2 \right) \right) \tag{11}$$

where $\beta > 0$ and $\alpha_k^{(t)} > 0$.

Concerning $C_{intra}$, since by construction each Dirichlet parameter $\alpha_k^{(t)}$ is strictly positive and $\epsilon$ is a small positive constant (used for numerical stability), the mean $\mu(\{\alpha_k^{(t)}\}_{t=1}^T)$ is itself strictly positive. Thus, the denominator in $C_{intra}$ is always positive and bounded above $0$. The numerator is the standard deviation of the Dirichlet parameters $\sigma(\{\alpha_k^{(t)}\}_{t=1}^T)$, thus, it is always non-negative or exactly zero in the case of identical Dirichlet parameters. The ratio $\frac{\sigma(...)}{\mu(...)+\epsilon}$ represents a normalised measure of dispersion. This ratio is well-known to lie between $0$ (no variability) and $1$, due to the nature of variance-to-mean ratios of strictly positive distributions. As a result, we establish:

$$0 \le C_{intra} \le 1 \tag{12}$$

Concerning $C_{inter}$ which is comprised of $1 - \exp(-x)$ and the argument:

$$x = -\beta \sum_{k=1}^{K} \sum_{j=k+1}^{K} \left( \frac{\min(\alpha_k^{(t)}, \alpha_j^{(t)})}{\max(\alpha_k^{(t)}, \alpha_j^{(t)})} \times \frac{\min(\alpha_k^{(t)}, \alpha_j^{(t)})}{\sum_{k=1}^{K} \alpha_k^{(t)}} \times 2 \right)^2 \tag{13}$$

Since by construction $\alpha_k^{(t)} \ge 1$, therefore $\min(\alpha_k^{(t)}, \alpha_j^{(t)})$ and $\max(\alpha_k^{(t)}, \alpha_j^{(t)})$ must be $\ge 1$ and $S^{(t)} \ge K$. Every squared factor inside the double sum us strictly positive, making the argument $x > 0$. Since $0 < \exp(-x) < 1$ for all $x > 0$, each term $1 - \exp(-x)$ lie strictly between $0$ and $1$. Thus, we establish:

$$0 < C_{inter} \le 1 \tag{14}$$

To quantify the tightness of the lower bound, we analyse the case where nearly all evidence is placed in a single class (the case that will bring $C_{inter}$ closest to $0$. Assume every transformation produces the same Dirichlet vector:

$$a_c = A >> 1, \quad a_j = 1(j \neq c), \quad S = A + K - 1 \tag{15}$$

where class $c$ is the favoured class, $A$ is the Dirichlet parameter assigned to class $c$, and all other class carry the implied prior constructed by the EDL process. Only the $K - 1$ pairs $\{c, j\}$ contribute the $C_{inter}$. For any such pair:

$$\min(\alpha_c, \alpha_j) = 1, \quad \max(\alpha_c, \alpha_j) = A \tag{16}$$

the squared factor becomes:

$$z = (\frac{1}{A}\frac{1}{A + K - 1}2)^2$$
$$= (K - 1)(\frac{2}{A(A + K - 1)})^2 \tag{17}$$

as there are $K - 1$ pairs. Substituting this into $C_{inter}$ gives:

$$C_{inter} = 1 - \exp(-\beta z) = 1 - \exp(-\beta(K - 1)(\frac{2}{A(A + K - 1)})^2) \tag{18}$$

assuming $K = 2$, the second term vanishes and $z = \mathcal{O}(A^{-4})$, hence in the case of a dominating single class, $C_{inter} = \mathcal{O}(A^{-4})$. Therefore, it can be said that $C_{inter}$ is strictly positive but can be driven arbitrarily closer to 0 at a quartic rate.

Given that both $C_{intra}$ is bounded $[0, 1]$ and $C_{inter}$ is bounded $(0, 1]$, we can determine the bounds of $C$.

In the case of the lower bound ($C_{intra} = 0$ and $C_{inter} = \epsilon$) where $\epsilon \in (0, 1]$, the conflict measure $C$ is:

$$C = \epsilon + 0 - (\epsilon)(0) - \lambda(\epsilon - 0)^2$$
$$= \epsilon - \lambda\epsilon^2 \tag{19}$$
$$= \epsilon(1 - \lambda\epsilon)$$

Since we have shown that $\epsilon \to 0^+$, it is strictly positive. In this case of the upper bound ($C_{intra} = 1$ and $C_{inter} = 1$), the conflict measure $C$ is:

$$C = 1 + 1 - (1)(1) - \lambda(1 - 1)^2$$
$$= 2 - 1 - 0 \tag{20}$$
$$= 0$$

Thus, $C$ is bounded on $(0, 1]$, satisfying the first part of the theorem.

Next, we prove that $C \to 0$ if and only if the Dirichlet parameters across transformations are identical and concentrated on a single class. Given a set of transformation $t \in \{1, \ldots, T\}$ and classes $k \in \{1, \ldots, K\}$, a set of identical Dirichlet parameters $\alpha_k^{(t)} = \alpha_k$ for all $t$ are produced. We have shown above that in this case $C_{intra} \to 0$ with the decay rate $\mathcal{O}(A^{-4})$ when $K = 2$. Because the Dirichlet parameters are identical, the intra-class conflict vanishes, thus $C_{intra} = 0$. Hence, $C$ shares the same properties. In this case of identical concentrated Dirichlet parameters, it tends towards 0 with the same decay rate. As a result, $C \to 0$ has little effect on the final evidence reduction/uncertainty boosting methodology.

Finally, we prove that $C$ is monotonically non-decreasing with increasing $C_{intra}$ and $C_{inter}$ when $\lambda \in [0, \frac{1}{2}]$. Since $C$ is continuously differentiable, monotonicity is equivalent to non-negativity of its partial derivatives. This is:

$$\frac{\partial C}{\partial C_{inter}} = 1 - C_{intra} - 2\lambda(C_{inter} - C_{intra})$$
$$\frac{\partial C}{\partial C_{intra}} = 1 - C_{inter} - 2\lambda(C_{inter} - C_{intra})$$

$$(21)$$

Each derivative is affine, therefore, its minimum over the rectangle $[0,1]^2$ is attained at one of the four corners $0,1^2$. Given $\frac{\partial C}{\partial C_{inter}}$, evaluating the corners gives:

$$(0,0):1, \quad (1,0):1-2\lambda, \quad (0,1):2\lambda, \quad (1,1):0 \tag{22}$$

with the smallest corner being $(1,0)$ with the value $1 - 2\lambda$. Under $\lambda \in [0, \frac{1}{2}]$, we get $1 - 2\lambda \geq 0$, thus:

$$\frac{\partial C}{\partial C_{inter}} \geq 0 \quad \text{for every } (C_{inter}, C_{intra}) \in (0,1]^2 \tag{23}$$

Increasing $C_{inter}$ while holding $C_{intra}$ never decreases $C$ when $\lambda \leq \frac{1}{2}$. Given $\frac{\partial C}{\partial C_{intra}}$, evaluating the corners gives:

$$(0,0):1, \quad (1,0):0, \quad (0,1):1-2\lambda, \quad (1,1):2\lambda \tag{24}$$

with the smallest corner being $(0,1)$ with the value $1 - 2\lambda$. Under $\lambda \in [0, \frac{1}{2}]$, we get $1 - 2\lambda \geq 0$, thus:

$$\frac{\partial C}{\partial C_{intra}} \geq 0 \quad \text{for every } (C_{inter}, C_{intra}) \in (0,1]^2 \tag{25}$$

Increasing $C_{intra}$ while holding $C_{inter}$ never decreases $C$ when $\lambda \leq \frac{1}{2}$. This proves that $C$ is monotonic for $\lambda \leq \frac{1}{2}$, this is qualitatively backed up with the $C$ visualisation w.r.t $C_{inter}$, $C_{intra}$, and $\lambda$ in Figure 16.

It is to be noted that $C$ remains non-negative on $[0,1]^2$ for $\lambda \in [0,1]$, yet monotonicity is lost when $\lambda > \frac{1}{2}$. Looking at Figure 16, the quadratic penalty $-\lambda(C_{inter} - C_{intra})^2$ bends the surface down along the diagonal. Restricting $\lambda$ to at most $\frac{1}{2}$ limits this curvature so that it does not overpower the linear ascent from $C_{inter} + C_{intra} - C_{inter}C_{intra}$. This proof is also part of the justification why we chose $\lambda = \frac{1}{2}$ in our experimental evaluation.

$\square$

## B  ADDITIONAL EXPERIMENTS

This appendix provides additional experimental insights and analyses to complement the core results presented in Section 5. Specifically, we include extended analysis of metrics, adversarial attacks, thresholds, and ablations of hyperparameters introduced in C-EDL.

### B.1  EXTENDED ANALYSIS OF CORE RESULTS

Table 2 shows the difference in ID accuracy, ID, OOD, and adversarial coverage of all approaches from the baseline (EDL) across the different datasets to complement Table 1.

Analysing this table shows us that EDL serves as a strong improvement to the Posterior Network, not only improving ID coverage but also OOD coverage. Thus, EDL learns to create a better uncertainty split between ID and OOD data across all datasets.

Notably, C-EDL (Meta) consistently achieves the largest reductions in OOD and adversarial coverage compared to EDL, while retaining competitive ID coverage across all dataset pairings. For instance,

Table 2: Difference from EDL baseline. Positive numbers improve ↑ metrics and worsen ↓ metrics; negative numbers do the opposite. The adversarial attack is an L2PGD attack [3] **Highlighted** cells denote the best performance for each metric. * indicates datasets classed as Near-OOD.

| | Comparative Methods | | | | | | | Ablated C-EDL Methods | | | Proposed C-EDL |
|---|---|---|---|---|---|---|---|---|---|---|---|
| | Posterior Network | EDL | $\mathcal{I}$-EDL | S-EDL | H-EDL | R-EDL | DA-EDL | EDL++ (MC) | C-EDL (MC) | EDL++ (Meta) | C-EDL (Meta) |
| MNIST → FashionMNIST | | | | | | | | | | | |
| ID Acc ↑ | $+0.00 \pm -0.01$ | 0 | $-0.01 \pm +0.00$ | $-0.01 \pm +0.00$ | $+0.00 \pm -0.01$ | $+0.00 \pm -0.01$ | $-0.07 \pm +0.03$ | $+0.01 \pm -0.01$ | $+0.02 \pm +0.00$ | $+0.01 \pm +0.00$ | $+0.00 \pm -0.01$ |
| ID Cov ↑ | $-1.89 \pm +0.39$ | **0** | $-0.53 \pm +0.10$ | $-0.55 \pm +0.15$ | $-0.43 \pm +0.35$ | $-0.34 \pm -0.09$ | $-1.45 \pm -0.20$ | $-3.26 \pm +0.72$ | $-4.47 \pm +0.86$ | $-1.68 \pm +0.03$ | $-2.43 \pm +0.46$ |
| OOD Cov ↓ | $+1.03 \pm -0.01$ | 0 | $+0.22 \pm +0.28$ | $-0.11 \pm -0.02$ | $-0.24 \pm +0.30$ | $-0.18 \pm +0.18$ | $+0.46 \pm +1.34$ | $+5.03 \pm +0.65$ | $-0.75 \pm -0.05$ | $-0.56 \pm -0.09$ | $-0.52 \pm +0.12$ |
| Adv Cov ↓ | $+8.93 \pm -0.11$ | 0 | $-4.63 \pm -0.70$ | $-3.41 \pm -0.74$ | $-1.81 \pm +5.11$ | $-3.16 \pm +2.41$ | $-23.47 \pm -1.56$ | $-13.40 \pm -5.26$ | $-30.59 \pm -4.60$ | $-35.78 \pm -3.97$ | $-36.70 \pm -3.40$ |
| MNIST → KMNIST | | | | | | | | | | | |
| ID Acc ↑ | $+0.00 \pm +0.00$ | 0 | $-0.01 \pm +0.00$ | $+0.00 \pm +0.00$ | $+0.00 \pm +0.00$ | $-0.01 \pm +0.00$ | $-0.07 \pm +0.02$ | $+0.01 \pm +0.01$ | $+0.01 \pm +0.01$ | $+0.01 \pm +0.00$ | $+0.01 \pm +0.00$ |
| ID Cov ↑ | $-0.70 \pm -0.11$ | 0 | $+0.05 \pm -0.07$ | $-0.34 \pm +0.17$ | $+0.17 \pm -0.14$ | $-0.10 \pm -0.09$ | $-1.30 \pm +0.58$ | $-3.48 \pm +0.37$ | $-5.22 \pm +1.04$ | $-1.69 \pm +0.20$ | $-3.00 \pm +0.03$ |
| OOD Cov ↓ | $+0.55 \pm +0.05$ | 0 | $+0.31 \pm -0.02$ | $-0.07 \pm -0.20$ | $+0.10 \pm -0.17$ | $-0.02 \pm -0.39$ | $+0.14 \pm +1.59$ | $+8.44 \pm +0.43$ | $+0.46 \pm +0.25$ | $-1.02 \pm -0.18$ | $-1.33 \pm -0.44$ |
| Adv Cov ↓ | $+3.03 \pm -2.25$ | 0 | $-1.31 \pm -1.68$ | $-5.92 \pm -1.84$ | $-0.85 \pm -3.04$ | $-4.48 \pm -1.46$ | $-10.82 \pm -1.74$ | $+6.91 \pm -3.29$ | $-8.48 \pm -3.86$ | $-16.68 \pm -4.02$ | $-17.87 \pm -5.01$ |
| MNIST → EMNIST* | | | | | | | | | | | |
| ID Acc ↑ | $+0.00 \pm +0.00$ | 0 | $+0.01 \pm +0.00$ | $+0.01 \pm +0.00$ | $+0.00 \pm +0.00$ | $+0.00 \pm +0.01$ | $-0.06 \pm +0.03$ | $+0.01 \pm +0.00$ | $+0.01 \pm +0.00$ | $+0.01 \pm +0.00$ | $+0.00 \pm +0.00$ |
| ID Cov ↑ | $+0.41 \pm -0.07$ | 0 | $+0.44 \pm -0.48$ | $-0.08 \pm +0.16$ | $+0.45 \pm +0.22$ | $+0.54 \pm -0.12$ | $-2.20 \pm +0.36$ | $-6.62 \pm +1.92$ | $-9.98 \pm +2.02$ | $-2.08 \pm +0.40$ | $-2.84 \pm +0.50$ |
| OOD Cov ↓ | $+4.22 \pm -0.32$ | 0 | $-0.12 \pm -0.41$ | $-0.43 \pm -0.01$ | $+0.55 \pm +0.25$ | $-0.68 \pm -0.32$ | $+3.40 \pm +7.13$ | $+6.42 \pm -0.00$ | $-3.91 \pm +0.03$ | $-1.86 \pm -0.30$ | $-1.81 \pm -0.59$ |
| Adv Cov ↓ | $+7.94 \pm +0.42$ | 0 | $-1.28 \pm -1.81$ | $-1.21 \pm -1.51$ | $+1.24 \pm +0.58$ | $-1.41 \pm -0.57$ | $+4.49 \pm +7.40$ | $+14.82 \pm -0.79$ | $-0.67 \pm -1.92$ | $-6.34 \pm -2.94$ | $-6.40 \pm -3.13$ |
| CIFAR10 → SVHN | | | | | | | | | | | |
| ID Acc ↑ | $-0.25 \pm 0.18$ | 0 | $+0.31 \pm +0.15$ | $+0.19 \pm +0.15$ | $+0.21 \pm +0.01$ | $+0.94 \pm +0.00$ | $-4.63 \pm +0.42$ | $+1.51 \pm -0.01$ | $+2.52 \pm -0.15$ | $+1.41 \pm -0.23$ | $+2.10 \pm -0.12$ |
| ID Cov ↑ | $-1.60 \pm -0.04$ | **0** | $-0.77 \pm -0.74$ | $-1.77 \pm +0.21$ | $-0.60 \pm -0.40$ | $-4.75 \pm -0.74$ | $-23.51 \pm +13.82$ | $-10.69 \pm -0.14$ | $-20.30 \pm -0.49$ | $-6.81 \pm -0.83$ | $-12.64 \pm -1.00$ |
| OOD Cov ↓ | $+0.91 \pm +0.29$ | 0 | $-1.23 \pm -0.39$ | $-0.51 \pm -0.29$ | $-1.07 \pm +0.60$ | $-1.99 \pm -0.47$ | $+9.49 \pm +7.02$ | $+1.10 \pm -0.35$ | $-4.25 \pm -1.17$ | $-4.32 \pm -1.06$ | $-6.22 \pm -1.23$ |
| Adv Cov ↓ | $-4.43 \pm -0.70$ | 0 | $-0.75 \pm -0.52$ | $-16.68 \pm -5.43$ | $-0.36 \pm -2.03$ | $-5.39 \pm -0.10$ | $+1.48 \pm +2.09$ | $-4.78 \pm -4.57$ | $-10.61 \pm -5.71$ | $-17.65 \pm -5.93$ | $-18.75 \pm -6.33$ |
| CIFAR10 → CIFAR100* | | | | | | | | | | | |
| ID Acc ↑ | $-0.50 \pm -0.34$ | 0 | $+0.01 \pm -0.25$ | $+0.03 \pm -0.16$ | $+0.10 \pm -0.24$ | $+0.74 \pm -0.29$ | $-7.38 \pm +0.11$ | $+1.21 \pm -0.08$ | $+1.98 \pm -0.30$ | $+0.85 \pm -0.30$ | $+1.56 \pm -0.43$ |
| ID Cov ↑ | $-0.42 \pm -1.18$ | 0 | $+0.51 \pm -1.14$ | $-0.20 \pm -1.54$ | $-1.08 \pm -1.78$ | $-3.12 \pm -1.50$ | $+3.65 \pm -0.38$ | $-8.83 \pm -1.18$ | $-18.31 \pm -2.08$ | $-3.86 \pm -1.58$ | $-10.15 \pm -1.58$ |
| OOD Cov ↓ | $+0.49 \pm -1.59$ | 0 | $+0.49 \pm -1.29$ | $+0.24 \pm -1.45$ | $-1.12 \pm -1.01$ | $-2.78 \pm -1.99$ | $+16.61 \pm +0.13$ | $+2.26 \pm -2.27$ | $-5.80 \pm -2.20$ | $-3.90 \pm -2.18$ | $-7.63 \pm -2.75$ |
| Adv Cov ↓ | $-1.72 \pm -2.35$ | 0 | $+0.08 \pm -1.01$ | $-6.29 \pm -3.18$ | $-1.29 \pm -1.89$ | $-4.62 \pm -2.57$ | $+17.86 \pm -0.67$ | $+2.30 \pm -3.05$ | $-6.82 \pm -3.36$ | $-8.56 \pm -3.86$ | $-10.85 \pm -4.55$ |
| Oxford Flowers → Deep Weeds | | | | | | | | | | | |
| ID Acc ↑ | $-0.41 \pm -0.19$ | 0 | $-0.11 \pm -0.08$ | $+0.09 \pm -0.07$ | $-0.03 \pm -0.11$ | $+0.78 \pm -0.42$ | $-6.62 \pm +2.60$ | $+0.72 \pm -0.32$ | $+1.00 \pm -0.57$ | $+0.20 \pm -0.06$ | $+0.59 \pm +0.01$ |
| ID Cov ↑ | $+6.19 \pm -2.08$ | 0 | $-2.94 \pm -3.83$ | $-1.97 \pm -1.49$ | $-0.50 \pm -0.03$ | $+1.11 \pm -1.88$ | $+7.73 \pm +0.99$ | $-9.47 \pm +3.85$ | $-21.34 \pm +1.29$ | $-11.94 \pm -0.28$ | $-16.17 \pm +3.04$ |
| OOD Cov ↓ | $-3.60 \pm -3.54$ | 0 | $-0.70 \pm -3.02$ | $-0.74 \pm -3.07$ | $-2.86 \pm -3.46$ | $-6.53 \pm -3.16$ | $-5.11 \pm -2.52$ | $-2.87 \pm -2.57$ | $-7.85 \pm -4.39$ | $-3.26 \pm -1.99$ | $-7.09 \pm -2.88$ |
| Adv Cov ↓ | $-5.33 \pm -3.39$ | 0 | $-0.83 \pm -2.06$ | $-2.01 \pm -2.27$ | $-4.96 \pm -2.66$ | $-12.32 \pm -5.14$ | $-8.88 \pm -3.02$ | $-8.02 \pm -4.08$ | $-12.43 \pm -5.54$ | $-6.63 \pm -3.11$ | $-11.64 \pm -4.10$ |

on the MNIST → EMNIST task, it reduces adversarial coverage by $-6.40 \pm 2.96$, and on CIFAR10 → SVHN by $-18.75 \pm 1.53$, the most substantial improvement among all approaches. These differences reinforce the point that C-EDL's superior robustness is both due to the added diversity and the conflict-aware adjustment. Furthermore, the consistently positive differences in ID accuracy and coverage for C-EDL (Meta) suggest that these gains do not come at the expense of model reliability on clean data.

Figure 9 shows a boxplot comparison of ID, OOD, and adversarial coverage across a subset of representative ID/near- and far-OOD dataset pairs, these result complement Figure 9a shown in Section 5.

In the MNIST → KMNIST dataset pair (far-OOD), all comparative approaches achieve high ID coverage, with EDL-based approaches generally outperforming the Posterior Network, as expected. C-EDL (Meta) also maintains competitive ID coverage, with only a modest drop of approximately 2% to 3%. In terms of OOD coverage, approaches that extend EDL, such as R-EDL and DA-EDL, achieve moderate reductions. However, C-EDL (Meta) surpasses all baselines, achieving the lowest OOD coverage with $1.90\% \pm 0.32$.

The most pronounced performance gain appears in adversarial coverage, where C-EDL offers a significant improvement over all comparative approaches. Most EDL-based models struggle to abstain on adversarially perturbed data, predicting on approximately 20% of such inputs. For instance, EDL yields an adversarial coverage of $20.88\% \pm 5.95$. In contrast, C-EDL (Meta) reduces this by more than a factor of four, attaining a superior coverage of only $3.01\% \pm 0.94$.

These trends persist in other far-OOD settings, such as CIFAR10 → SVHN, where C-EDL sacrifices only a negligible amount of ID coverage but achieves the lowest OOD coverage and a near-zero adversarial coverage. Similar results are observed in the MNIST → EMNIST pairing (near-OOD), where C-EDL (Meta) once again delivers the lowest OOD coverage and near-zero adversarial coverage, marking a substantial improvement over all comparative baselines.

These results provide strong empirical evidence into the reliability of C-EDL across diverse settings. Despite only a negligible reduction in ID coverage, C-EDL consistently achieves the lowest OOD and adversarial coverage. This makes it particularly well-suited for deployment in environments where distributional shift or adversarial attacks are likely, offering a level of robustness unmatched by existing approaches.

To complement the adversarial coverage plots shown in the main paper (Figure 5), we present a detailed analysis of abstention margins ($\Delta$) across all evaluated dataset pairs, with raw values shown in Table 3. These plots visualise the mean difference between each sample's uncertainty score and the learned ID/OOD decision threshold, alongside the corresponding coverage rates. Positive values

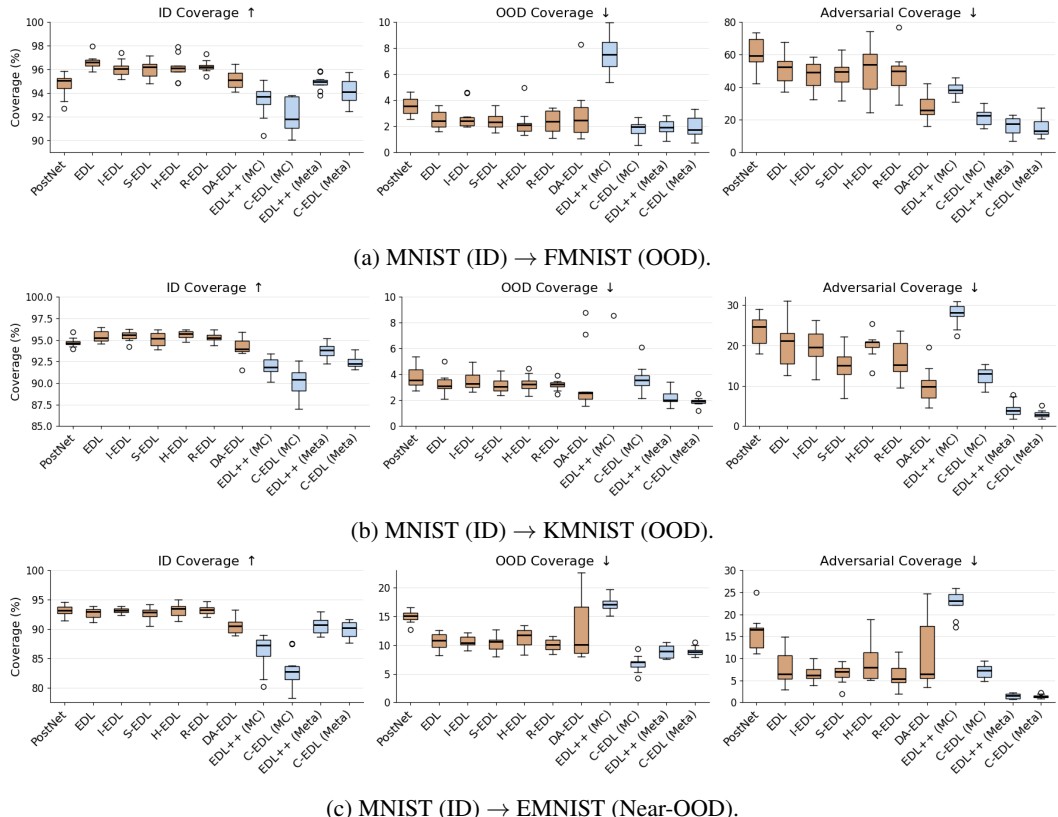

(a) MNIST (ID) → FMNIST (OOD).

(b) MNIST (ID) → KMNIST (OOD).

(c) MNIST (ID) → EMNIST (Near-OOD).

Figure 9: ID, OOD, and adversarial coverage (%) across comparative approaches for evaluated MNIST datasets from Table 1. C-EDL and its variants are in blue and show significantly lower adversarial and OOD coverage than baselines.

of $\Delta$ (expected for ID data) indicate retained predictions lie confidently above the threshold, while negative values (expected for OOD or adversarial inputs) reflect confident abstention.

For ID coverage (Figure 11a), we observe that C-EDL (Meta) achieves a high positive $\Delta$ comparable to the EDL variants. For example, in MNIST → FashionMNIST, EDL achieves $\Delta = 3.2$ whilst C-EDL (Meta) achieves $\Delta = 2.7$, which is competitive. It does this with competitive ID coverage also, only dropping $\approx 2\%$, reflecting confidence retention of ID examples. Interestingly to note is DA-EDL, which achieves extremely high ID $\Delta$ values compared to other approaches, which indicates extreme confidence in the data being ID. However, the coverage does not improve much from EDL results. This result can be explained by looking at Figure 11b simultaneously, which shows the OOD coverage and $\Delta$. DA-EDL shows similar OOD $\Delta$ to comparative approaches. This can explain the high confidence but no/marginal improvement in ID and OOD coverage as the density-based scaling predominantly affects confidence estimates in dense ID regions, rather than uniformly impacting all input regions.

For OOD coverage (Figure 11b), C-EDL (Meta) shows the most desirable behaviour: large negative $\Delta$ and the smallest OOD coverage across every dataset. For example, C-EDL (Meta) achieves $\Delta = -13.3$ on MNIST → FashionMNIST and $\Delta = -7.8$ on CIFAR10 → CIFAR100, which is one of the smallest on these datasets. But unlike other approaches, C-EDL achieves tiny coverage and low $\Delta$ on all datasets equally instead of on some. This means it not only abstains effectively, but does so with high confidence and margin, which is essential for robust deployment.

These $\Delta$ results and visualisations provide analysis to understand the quality of abstentions made by the different models and how C-EDL provides consistent, robust, significant improvements.

Figure 12 presents exclusion-based attention overlays generated by the C-EDL model, comparing OOD inputs from FashionMNIST (left) to ID inputs from MNIST (right). These heatmaps highlight

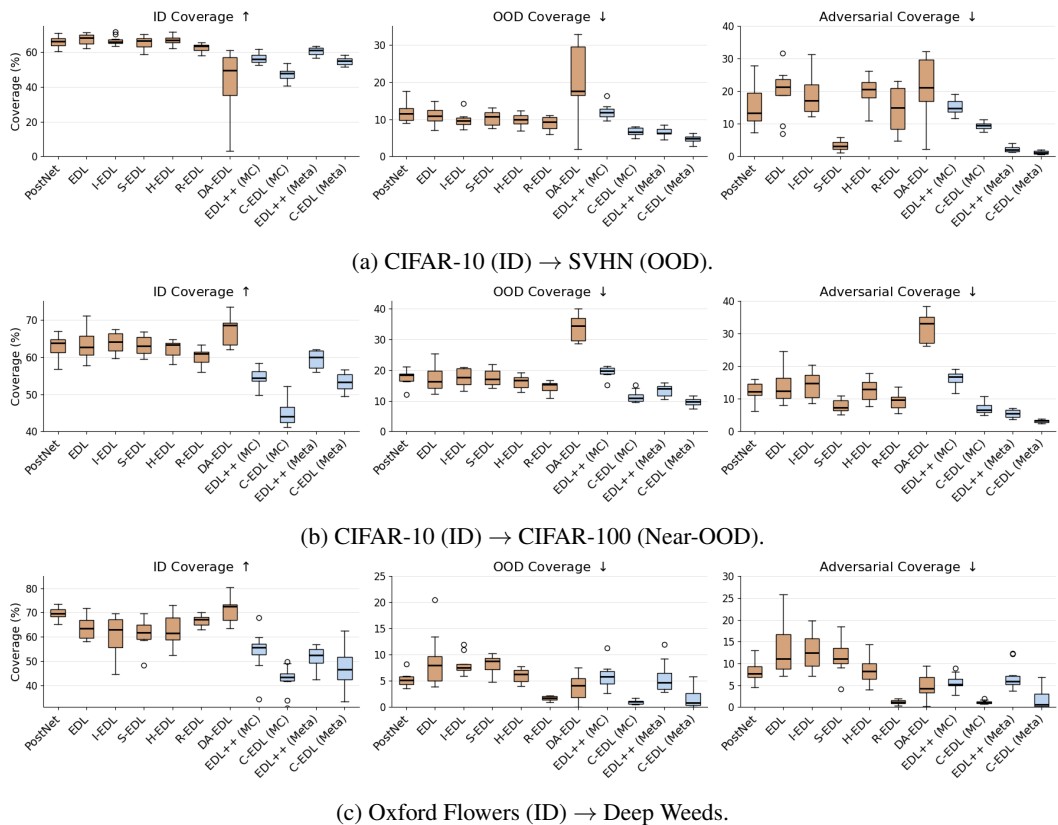

(a) CIFAR-10 (ID) → SVHN (OOD).

(b) CIFAR-10 (ID) → CIFAR-100 (Near-OOD).

(c) Oxford Flowers (ID) → Deep Weeds.

Figure 10: ID, OOD, and adversarial coverage (%) across comparative approaches for evaluated CIFAR-10, and Flowers datasets from Table 1. C-EDL and its variants are in blue and show significantly lower adversarial and OOD coverage than baselines.

Table 3: Mean difference between the computed abstention metric for each sample and the computed ID/OOD threshold for each approach from Table 1. Positive values indicate predictions tend to be above the threshold (expected for ID data), while negative values indicate predictions fall below the threshold (as expected for OOD and Adversarially attacked data). * indicates datasets classed as Near-OOD.

| | Comparative Methods | | | | | | | Ablated C-EDL Methods | | | Proposed C-EDL |
|---|---|---|---|---|---|---|---|---|---|---|---|
| | Posterior Network | EDL | $\mathcal{I}$-EDL | S-EDL | H-EDL | R-EDL | DA-EDL | EDL++ (MC) | C-EDL (MC) | EDL++ (Meta) | C-EDL (Meta) |
| | MNIST → FashionMNIST | | | | | | | | | | |
| ID ↑ | $2.31 \pm 0.37$ | $3.17 \pm 0.39$ | $2.91 \pm 0.44$ | $2.91 \pm 0.46$ | $2.97 \pm 0.51$ | $3.99 \pm 0.39$ | $352.94 \pm 381.70$ | $2.49 \pm 0.33$ | $2.13 \pm 0.43$ | $2.79 \pm 0.38$ | $2.67 \pm 0.42$ |
| OOD ↓ | $-10.38 \pm 1.21$ | $-12.46 \pm 1.11$ | $-12.08 \pm 1.52$ | $-11.32 \pm 0.93$ | $-11.99 \pm 1.22$ | $-16.42 \pm 0.92$ | $-18.06 \pm 13.60$ | $-2.51 \pm 0.51$ | $-5.10 \pm 0.57$ | $-4.47 \pm 1.23$ | $-5.50 \pm 1.24$ |
| Adv ↓ | $-3.06 \pm 1.63$ | $-2.89 \pm 1.13$ | $-3.55 \pm 1.46$ | $-2.57 \pm 1.28$ | $-2.10 \pm 1.19$ | $-0.65 \pm 0.72$ | $1.86 \pm 9.29$ | $-2.51 \pm 0.51$ | $-5.10 \pm 0.57$ | $-4.47 \pm 1.23$ | $-5.50 \pm 1.24$ |
| | MNIST → KMNIST | | | | | | | | | | |
| ID ↑ | $2.37 \pm 0.27$ | $2.47 \pm 0.18$ | $2.59 \pm 0.26$ | $2.46 \pm 0.26$ | $2.64 \pm 0.24$ | $3.38 \pm 0.30$ | $277.81 \pm 172.05$ | $2.05 \pm 0.20$ | $1.75 \pm 0.28$ | $2.33 \pm 0.26$ | $2.00 \pm 0.21$ |
| OOD ↓ | $-11.26 \pm 1.03$ | $-10.55 \pm 0.60$ | $-10.43 \pm 0.50$ | $-10.79 \pm 0.53$ | $-10.43 \pm 0.43$ | $-15.96 \pm 0.32$ | $-22.18 \pm 21.55$ | $-5.63 \pm 0.27$ | $-9.76 \pm 0.34$ | $-10.66 \pm 0.33$ | $-12.55 \pm 0.39$ |
| Adv ↓ | $-6.66 \pm 1.77$ | $-3.70 \pm 0.66$ | $-3.83 \pm 0.49$ | $-4.17 \pm 0.54$ | $-3.82 \pm 0.43$ | $-3.76 \pm 0.49$ | $-13.84 \pm 9.73$ | $-3.13 \pm 0.30$ | $-6.37 \pm 0.37$ | $-6.23 \pm 0.62$ | $-7.97 \pm 0.49$ |
| | MNIST → EMNIST* | | | | | | | | | | |
| ID ↑ | $1.91 \pm 0.21$ | $1.83 \pm 0.19$ | $1.92 \pm 0.12$ | $1.84 \pm 0.20$ | $1.98 \pm 0.26$ | $2.72 \pm 0.24$ | $307.70 \pm 174.56$ | $1.32 \pm 0.30$ | $0.90 \pm 0.22$ | $1.63 \pm 0.26$ | $1.56 \pm 0.27$ |
| OOD ↓ | $-8.81 \pm 0.50$ | $-8.69 \pm 0.38$ | $-8.80 \pm 0.25$ | $-9.00 \pm 0.30$ | $-8.52 \pm 0.35$ | $-12.79 \pm 0.38$ | $-4.27 \pm 25.72$ | $-5.70 \pm 0.28$ | $-9.64 \pm 0.23$ | $-8.80 \pm 0.30$ | $-10.44 \pm 0.23$ |
| Adv ↓ | $-7.84 \pm 1.45$ | $-5.70 \pm 0.46$ | $-5.84 \pm 0.53$ | $-5.88 \pm 0.52$ | $-5.45 \pm 0.61$ | $-6.06 \pm 0.70$ | $-6.32 \pm 27.76$ | $-3.82 \pm 0.31$ | $-7.53 \pm 0.42$ | $-7.54 \pm 0.33$ | $-9.13 \pm 0.54$ |
| | CIFAR10 → SVHN | | | | | | | | | | |
| ID ↑ | $2.20 \pm 0.60$ | $1.97 \pm 0.63$ | $1.81 \pm 0.54$ | $1.66 \pm 0.70$ | $1.82 \pm 0.52$ | $-0.04 \pm 0.99$ | $-281.83 \pm 1489.45$ | $0.66 \pm 0.54$ | $-1.10 \pm 0.79$ | $0.96 \pm 0.49$ | $-0.18 \pm 0.58$ |
| OOD ↓ | $-4.78 \pm 0.61$ | $-4.88 \pm 0.34$ | $-4.86 \pm 0.30$ | $-5.30 \pm 0.37$ | $-4.94 \pm 0.30$ | $-11.52 \pm 0.54$ | $-483.99 \pm 1506.94$ | $-6.72 \pm 0.91$ | $-9.30 \pm 0.69$ | $-5.37 \pm 0.44$ | $-7.81 \pm 0.47$ |
| Adv ↓ | $-3.82 \pm 0.80$ | $-3.03 \pm 0.69$ | $-3.09 \pm 0.48$ | $-5.91 \pm 0.45$ | $-3.00 \pm 0.45$ | $-7.45 \pm 1.55$ | $-478.49 \pm 1488.76$ | $-6.05 \pm 0.90$ | $-8.64 \pm 0.69$ | $-5.24 \pm 0.45$ | $-7.78 \pm 0.53$ |
| | CIFAR10 → CIFAR100* | | | | | | | | | | |
| ID ↑ | $1.60 \pm 0.49$ | $1.21 \pm 0.81$ | $1.28 \pm 0.52$ | $1.12 \pm 0.45$ | $0.97 \pm 0.43$ | $-1.11 \pm 0.68$ | $242.41 \pm 106.62$ | $0.24 \pm 0.44$ | $-1.72 \pm 0.81$ | $0.69 \pm 0.43$ | $-0.67 \pm 0.51$ |
| OOD ↓ | $-4.67 \pm 0.50$ | $-4.82 \pm 0.77$ | $-4.68 \pm 0.50$ | $-4.83 \pm 0.49$ | $-4.97 \pm 0.38$ | $-11.33 \pm 0.65$ | $56.73 \pm 32.73$ | $-4.27 \pm 0.48$ | $-7.43 \pm 0.77$ | $-7.80 \pm 0.48$ | $-8.25 \pm 0.48$ |
| Adv ↓ | $-4.39 \pm 0.51$ | $-3.93 \pm 0.68$ | $-3.93 \pm 0.56$ | $-4.97 \pm 0.43$ | $-4.13 \pm 0.47$ | $-9.18 \pm 0.68$ | $52.42 \pm 31.52$ | $-4.28 \pm 0.48$ | $-7.64 \pm 0.75$ | $-5.46 \pm 0.47$ | $-8.25 \pm 0.48$ |
| | Oxford Flowers → Deep Weeds | | | | | | | | | | |
| ID ↑ | $5.20 \pm 4.87$ | $1.78 \pm 5.00$ | $-0.22 \pm 6.95$ | $0.22 \pm 6.28$ | $2.79 \pm 7.62$ | $-10.60 \pm 7.55$ | $539.10 \pm 195.46$ | $-6.54 \pm 6.92$ | $-19.88 \pm 6.70$ | $-7.83 \pm 3.79$ | $-12.05 \pm 9.10$ |
| OOD ↓ | $-41.61 \pm 4.34$ | $-33.16 \pm 4.97$ | $-30.55 \pm 6.44$ | $-31.71 \pm 5.55$ | $-31.80 \pm 7.54$ | $-62.86 \pm 5.06$ | $-0.61 \pm 0.94$ | $-60.01 \pm 6.49$ | $-85.02 \pm 9.72$ | $-34.99 \pm 7.07$ | $-49.65 \pm 11.09$ |
| Adv ↓ | $-43.08 \pm 5.77$ | $-32.65 \pm 5.71$ | $-29.68 \pm 7.87$ | $-31.64 \pm 7.05$ | $-32.40 \pm 10.28$ | $-68.67 \pm 7.34$ | $-0.54 \pm 0.89$ | $-61.10 \pm 6.97$ | $-83.18 \pm 8.36$ | $-34.89 \pm 7.55$ | $-60.74 \pm 7.45$ |

spatial regions where the model's attention distributions differ across metamorphic transformations, providing an intuitive visual explanation/insight of to where conflict could potentially lead to an increase in the conflict score $C$.

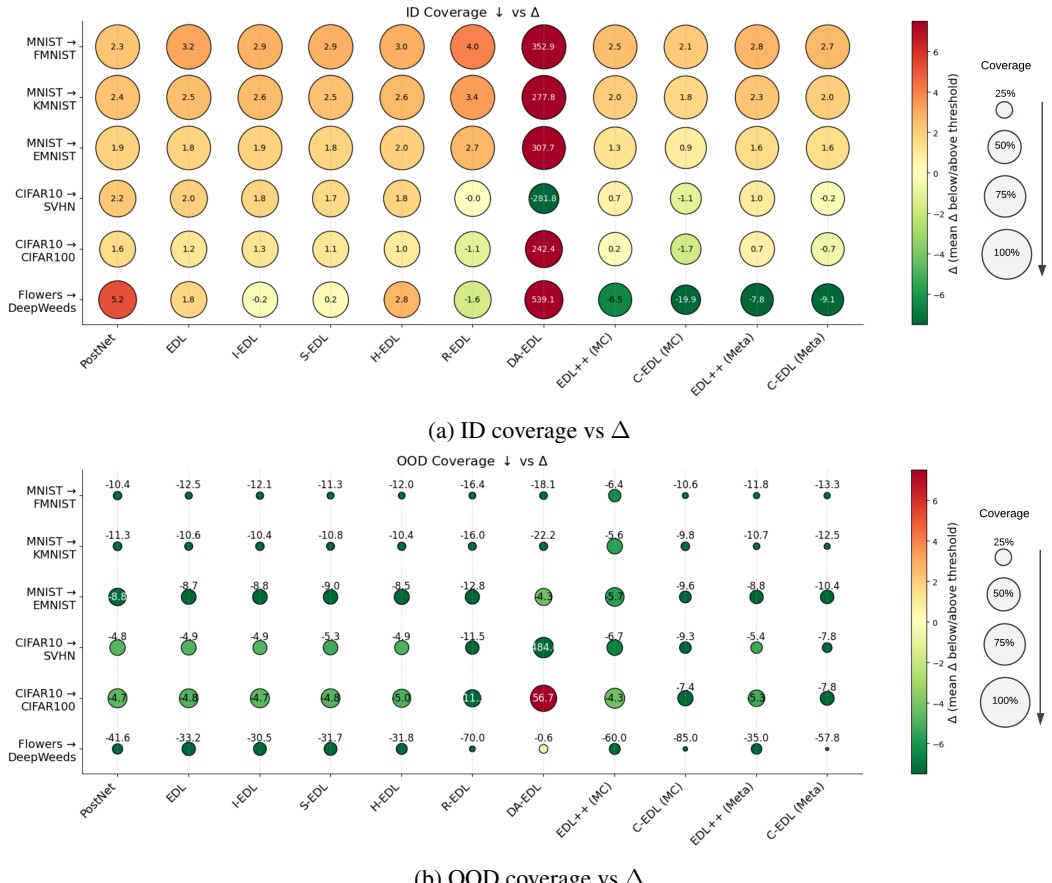

(a) ID coverage vs $\Delta$

(b) OOD coverage vs $\Delta$

Figure 11: Coverage (bubble size) compared to mean difference between the computed abstention metric (bubble colour - Table 3) for all approaches on all evaluated datasets.

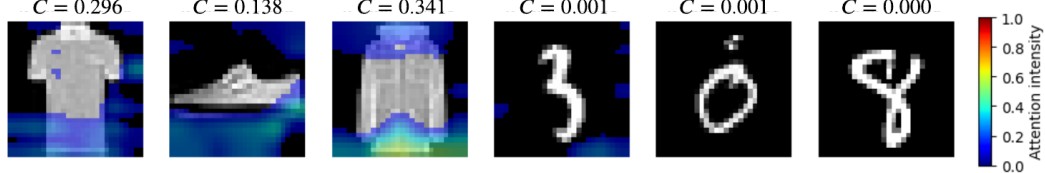

Figure 12: Exclusion-based attention overlays from the C-EDL model for OOD (FashionMNIST, left) and ID (MNIST, right) inputs. The heatmaps highlight spatial regions where attention distributions differ across metamorphic transformations. These overlays qualitatively highlight regions of disagreement that could potentially lead to an increase in the conflict measure $C$.

For OOD samples (left three images), we can observe high regions of disagreement across the OOD images. For example, in the third image showing a coat, the lower and upper regions show disagreement across the transformation $T$, which in turn leads to a high conflict score $C = 0.341$. In contrast, ID samples (right three images) exhibit little to no areas of disagreement across the transformations. Correspondingly, the conflict scores are near-zero (e.g., $C = 0.000$ to $C = 0.001$), confirming that the model's evidence remains stable and coherent for familiar inputs.

AUROC plots and adversarial AUROC plots for all comparative methods can be visualised in Figures 13 and 14 respectively.

C-EDL and its variants were also compared against broader UQ methods that comprise state-of-the-art in a broader UQ sense; results are shown in Table 4. We compared C-EDL and its ablated variants

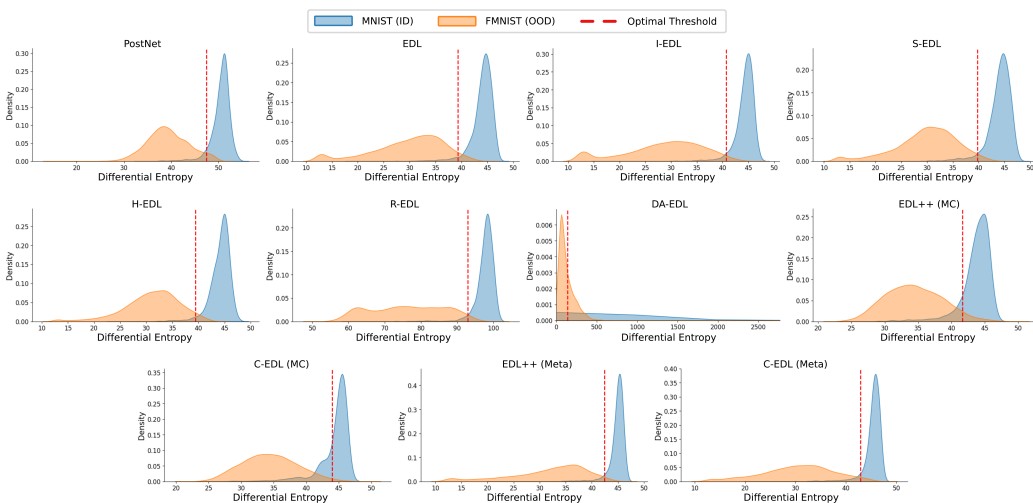

Figure 13: Visualised AUROC plots, including the binary decision threshold for OOD/Adv rejection, for comparative methods where the ID dataset is MNIST and the OOD dataset is FashionMNIST.

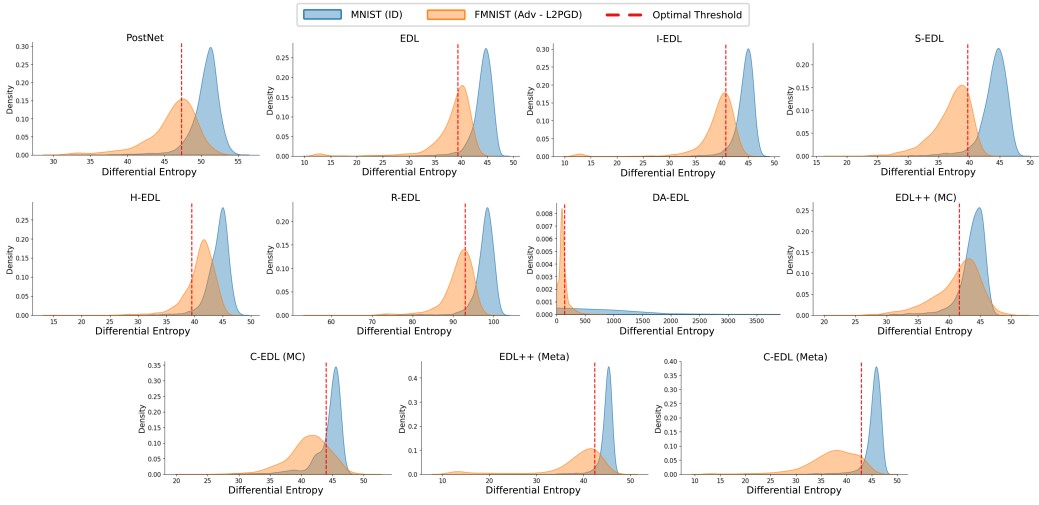

Figure 14: Visualised adversarial AUROC plots, including the binary decision threshold for OOD/Adv rejection, for comparative methods where the ID dataset is MNIST and the OOD dataset is Fashion-MNIST.

against a standard deterministic network, ABNN (Franchi et al., 2024), a standard deterministic backbone that is then fine-tuned with an EDL head (Sensoy et al., 2018), and EMM (Shen et al., 2023).

Table 4: The accuracy, OOD detection, and adversarial attack detection performance of the C-EDL and broader UQ methods on the MNIST to FashionMNIST pairing. The adversarial attack is an L2PGD attack. **Highlighted** cells denote the best performance for each metric.

| | Standard Network | MC-Dropout | ABNN | EDL-Head | EMM | EDL++ (MC) | C-EDL (MC) | EDL++ (Meta) | C-EDL (Meta) |
|---|---|---|---|---|---|---|---|---|---|
| ID Acc ↑ | $99.80\% \pm 0.05$ | $99.94 \pm 0.03$ | $99.78\% \pm 0.81$ | $99.43\% \pm 0.11$ | $99.55\% \pm 0.06$ | $99.98\% \pm 0.01$ | **$99.98\% \pm 0.01$** | $99.97\% \pm 0.02$ | $99.96\% \pm 0.02$ |
| ID Cov ↑ | $79.26\% \pm 3.12$ | $88.46 \pm 2.61$ | $74.88\% \pm 5.41$ | $77.44\% \pm 1.21$ | $90.58\% \pm 2.78$ | $93.51\% \pm 1.12$ | $92.05\% \pm 1.72$ | **$95.03\% \pm 0.71$** | $94.22\% \pm 1.05$ |
| OOD Cov ↓ | $24.89\% \pm 6.53$ | $12.82 \pm 3.06$ | $18.23\% \pm 5.83$ | $22.51\% \pm 3.76$ | $31.87\% \pm 17.98$ | $35.73\% \pm 3.62$ | $5.80\% \pm 1.42$ | $1.92\% \pm 0.93$ | **$1.53\% \pm 0.80$** |
| Adv Cov ↓ | $25.84\% \pm 5.34$ | $27.79 \pm 4.70$ | $24.35\% \pm 6.98$ | $19.86\% \pm 2.96$ | $22.18\% \pm 15.34$ | $54.42\% \pm 3.77$ | $23.39\% \pm 5.70$ | **$17.62\% \pm 7.64$** | $17.68\% \pm 4.81$ |

Table 5: The accuracy, OOD detection, AUROC, and adversarial attack detection performance of EDL and C-EDL (meta) on few-shot Tiny-ImageNet and CUB pairing using a pretrained ResNet50-Wide backbone. The adversarial attack is L2PGD (0.1 maximum perturbation). Highlighted cells denote the best performance for each metric.

| | 5-Way 1-Shot | | 5-Way 5-Shot | |
| --- | --- | --- | --- | --- |
| | EDL | C-EDL (Meta) | EDL | C-EDL (Meta) |
| ID Acc ↑ | $51.63\% \pm 3.17$ | $\mathbf{53.19\% \pm 4.67}$ | $\mathbf{77.90\% \pm 8.23}$ | $74.85\% \pm 5.07$ |
| ID Cov ↑ | $\mathbf{67.20\% \pm 18.54}$ | $65.07\% \pm 18.37$ | $44.53\% \pm 7.58$ | $\mathbf{45.60\% \pm 21.71}$ |
| OOD Cov ↓ | $50.13\% \pm 16.67$ | $\mathbf{34.67\% \pm 23.25}$ | $29.87\% \pm 8.77$ | $\mathbf{16.14\% \pm 10.84}$ |
| AUROC ↑ | $64.50\% \pm 4.27$ | $\mathbf{69.76\% \pm 5.00}$ | $62.94\% \pm 6.40$ | $\mathbf{67.48\% \pm 4.56}$ |
| Adv Cov ↓ | $43.33\% \pm 19.41$ | $\mathbf{31.11\% \pm 27.94}$ | $26.67\% \pm 4.65$ | $\mathbf{20.56\% \pm 15.16}$ |

Table 6: The accuracy, OOD detection, and adversarial attack detection performance of the C-EDL compared to RED on the MNIST to FashionMNIST and CIFAR-10 to SVHN pairings. The adversarial attack is an L2PGD attack.

| | EDL | RED | C-EDL (Meta) |
| --- | --- | --- | --- |
| | MNIST → FMNIST | | |
| ID Acc ↑ | $99.96\% \pm 0.02$ | $99.98\% \pm 0.02$ | $99.96\% \pm 0.01$ |
| ID Cov ↑ | $96.61\% \pm 0.57$ | $92.17\% \pm 1.77$ | $94.18\% \pm 1.03$ |
| OOD Cov ↓ | $2.52\% \pm 0.68$ | $6.30\% \pm 9.44$ | $2.00\% \pm 0.80$ |
| Adv Cov ↓ | $52.21\% \pm 9.49$ | $19.05\% \pm 9.03$ | $15.51\% \pm 6.09$ |
| | CIFAR-10 → SVHN | | |
| ID Acc ↑ | $95.88\% \pm 0.54$ | $95.48\% \pm 1.26$ | $97.98\% \pm 0.42$ |
| ID Cov ↑ | $67.34\% \pm 3.15$ | $60.24\% \pm 5.35$ | $54.70\% \pm 2.11$ |
| OOD Cov ↓ | $10.91\% \pm 2.23$ | $11.68\% \pm 3.13$ | $4.69\% \pm 1.00$ |
| Adv Cov ↓ | $20.00\% \pm 6.89$ | $4.39\% \pm 2.96$ | $1.25\% \pm 0.56$ |

C-EDL and it EDL were also tested against Tiny-ImageNet and CUB in a few-shot setting to test their ability and robustness under a tiny training regime, a poorly trained classifier, and extremely difficult dataset pairing; results are shown in Table 5.

Table 6 compares the proposed C-EDL (meta) against RED Pandey & Yu (2023). Across both dataset shifts, both methods preserve near-identical ID accuracy, while exhibiting clear separability between ID and non-ID inputs: the ID coverage remains high, whereas the corresponding OOD and adversarial coverages are substantially lower. This separation is particularly pronounced for C-EDL (Meta), which shows a larger ID to OOD and ID to adversarial gap than RED on CIFAR-10→SVHN, indicating stronger discrimination between in-distribution samples and shifted or attacked inputs. On MNIST→FMNIST, the two methods are closer, but still maintain a consistent ordering where ID remains highest and OOD/adversarial are markedly reduced, supporting the claim that both approaches meaningfully differentiate ID from non-ID regimes.

## B.2 ADVERSARIAL ATTACK ANALYSIS

Table 7 complements Figure 7 by providing results of adversarial coverage across the range of attacks and perturbation levels. These results further strengthen the case for C-EDL (Meta). Across all attack types and perturbation strengths, C-EDL (Meta) consistently achieves the lowest adversarial coverage, frequently by a substantial margin. For the strongest gradient-based attack (L2PGD at $\epsilon = 1.0$), C-EDL (Meta) attains a remarkably low adversarial coverage of $15.51\% \pm 6.09$, compared to $52.21\% \pm 9.49$ for EDL. Even S-EDL, a similar post-hoc EDL approach designed specifically for adversarial robustness, only reaches $48.80\% \pm 8.75$, further underscoring the advantage of C-EDL's conflict-aware calibration.

Notably, C-EDL (Meta) also retains its superiority under the non-gradient-based Salt-and-Pepper noise attack, where it achieves the lowest coverage at every tested perturbation. In non-gradient-based attacks, smaller perturbations are typically harder to detect than greater ones and therefore have a greater chance to fool the model; the inverse of gradient-based attacks. For instance, at $\epsilon = 1.0$, C-EDL (Meta) achieves $0.14\% \pm 0.00$, a considerable reduction.

Table 7: Adversarial attack detection of the comparative approaches for a variety of adversarial attacks and maximum perturbations $\epsilon$ where the ID dataset is MNIST and the OOD dataset is FashionMNIST. **Highlighted** cells denote the best performance.

| | Comparative Methods | | | | | | | Ablated C-EDL Methods | | | Proposed C-EDL |
|---|---|---|---|---|---|---|---|---|---|---|---|
| $\epsilon$ | Posterior Network | EDL | $\mathcal{I}$-EDL | S-EDL | H-EDL | R-EDL | DA-EDL | EDL++ (MC) | C-EDL (MC) | EDL++ (Meta) | C-EDL (Meta) |
| | L2PGD | | | | | | | | | | |
| 0.1 | 1.97%±1.04 | 1.16%±0.35 | 1.15%±0.45 | 1.15%±0.41 | 1.13%±0.55 | **0.94%±0.20** | 5.69%±10.56 | 5.63%±1.34 | 1.34%±0.41 | 1.04%±0.43 | 1.04%±0.41 |
| 0.5 | 3.84%±2.44 | 1.51%±0.74 | 1.65%±0.96 | 1.01%±0.60 | 1.67%±0.96 | 1.63%±1.86 | 2.55%±2.29 | 14.43%±2.43 | 5.28%±2.29 | 0.38%±0.26 | **0.33%±0.29** |
| 1.0 | 61.14%±9.38 | 52.21%±9.49 | 47.58%±8.79 | 48.80%±8.75 | 50.40%±14.60 | 49.05%±11.90 | 28.74%±7.93 | 38.81%±4.23 | 21.62%±4.89 | 16.43%±5.52 | **15.51%±6.09** |
| | FGSM | | | | | | | | | | |
| 0.1 | 6.84%±2.41 | 2.53%±1.08 | 2.43%±1.46 | 2.68%±0.01 | 2.50%±0.01 | 3.56%±0.03 | 3.10%±0.05 | 12.13%±0.03 | 4.70%±0.02 | 1.35%±0.01 | **1.01%±0.00** |
| 0.5 | 5.48%±4.32 | 2.10%±1.63 | 2.12%±1.51 | 2.76%±0.04 | 1.31%±0.01 | 4.06%±0.08 | 2.27%±0.05 | 3.89%±0.01 | **0.97%±0.00** | 2.78%±0.13 | 1.13%±0.04 |
| 1.0 | 2.72%±2.41 | 4.36%±6.01 | 5.00%±6.89 | 3.67%±0.10 | 2.04%±0.03 | 3.93%±0.12 | 1.38%±0.10 | 1.50%±0.00 | **0.25%±0.00** | 6.16%±1.24 | 0.67%±0.01 |
| | SaltAndPepperNoise | | | | | | | | | | |
| 0.1 | 2.99%±0.00 | 1.72%±0.00 | 2.45%±0.02 | 1.91%±0.01 | 2.63%±0.01 | 2.04%±0.01 | 1.95%±0.01 | 6.29%±0.02 | **1.34%±0.00** | 1.78%±0.00 | 1.53%±0.00 |
| 0.5 | 1.08%±0.00 | 0.63%±0.00 | 0.74%±0.00 | 0.55%±0.00 | 0.68%±0.00 | 0.57%±0.00 | 1.26%±0.01 | 3.23%±0.01 | 0.65%±0.00 | 0.62%±0.00 | **0.49%±0.00** |
| 1.0 | 0.39%±0.00 | 0.20%±0.00 | 0.22%±0.00 | 0.17%±0.00 | 0.20%±0.00 | 0.18%±0.00 | 0.88%±0.01 | 2.54%±0.01 | 0.24%±0.00 | 0.20%±0.00 | **0.14%±0.00** |

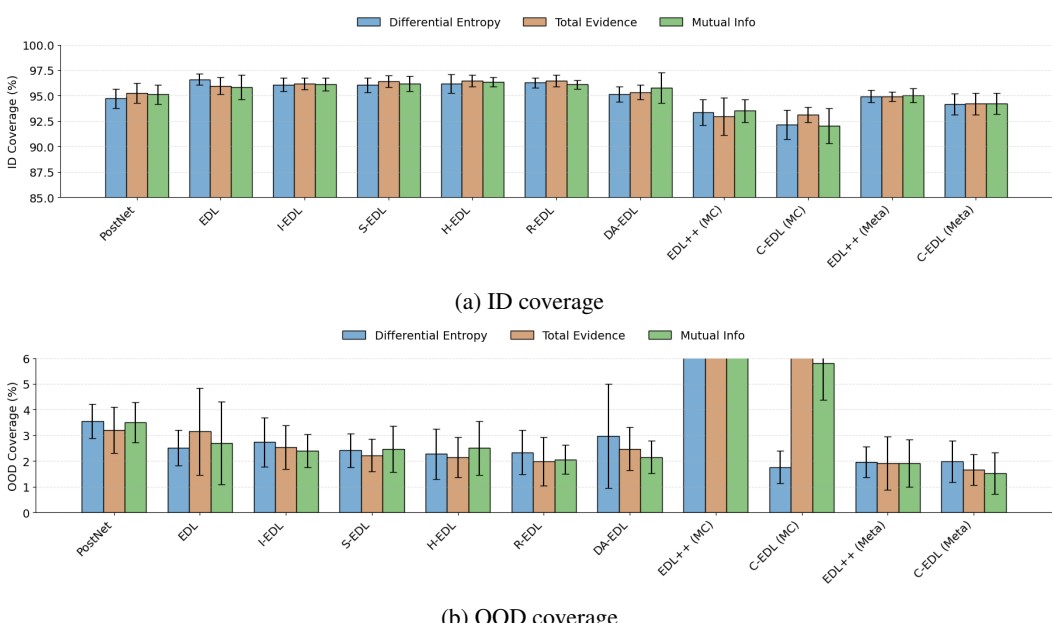

(a) ID coverage

(b) OOD coverage

Figure 15: Coverage (%) across comparative approaches for different ID-OOD threshold metrics where the ID dataset is MNIST, the OOD dataset is FashionMNIST, and the adversarial attack is an L2PGD attack (maximum $\epsilon = 1.0$).

### B.3 THRESHOLD ANALYSIS

Table 8 complements Figure 8 from the main body by providing the full quantitative breakdown of how different thresholding strategies affect ID, OOD, and adversarial coverage across approaches. Figure 15 shows the ID and OOD coverage of all approaches under different decision thresholds.

In Figure 15a, despite its conservative abstention strategy, C-EDL (Meta) preserves a high ID coverage rate across all thresholds, indicating that it experiences little to no conflict across transformations corroborating the results shown in Figure 12. While EDL, I-EDL, and R-EDL achieve slightly higher ID coverage, they do so at the cost of worse OOD and adversarial rejection, often allowing $2\check{\ }3\times$ more OOD examples through.

In Figure 15b, OOD coverage for C-EDL (Meta) remains tightly clustered below 2% across Differential Entropy, Total Evidence, and Mutual Information, with the lowest value being $1.80\% \pm 0.89$ under Mutual Information. In contrast, other approaches such as DA-EDL and EDL exhibit far greater variance (depending on the chosen metric), often with elevated OOD leakage above 3–4%, highlighting weaker separation between ID and OOD inputs.

Table 8: Accuracy, OOD detection, and adversarial attack detection performance of the comparative approaches with different threshold metrics. The adversarial attack is L2PGD (1.0 maximum perturbation), and the ID and OOD datasets are MNIST and FashionMNIST. **Highlighted** cells denote the best performance for each metric.

| | Comparative Methods | | | | | | | Ablated C-EDL Methods | | | Proposed C-EDL |
|---|---|---|---|---|---|---|---|---|---|---|---|
| | Posterior Network | EDL | $\mathcal{I}$-EDL | S-EDL | H-EDL | R-EDL | DA-EDL | EDL++ (MC) | C-EDL (MC) | EDL++ (Meta) | C-EDL (Meta) |
| Differential Entropy | | | | | | | | | | | |
| ID Acc ↑ | 99.96% ± 0.01 | 99.96% ± 0.02 | 99.95% ± 0.02 | 99.95% ± 0.02 | 99.96% ± 0.02 | 99.96% ± 0.01 | 99.89% ± 0.05 | 99.97% ± 0.01 | **99.98% ± 0.02** | 99.97% ± 0.02 | 99.96% ± 0.01 |
| ID Cov ↑ | 94.72% ± 0.96 | **96.61% ± 0.57** | 96.08% ± 0.67 | 96.06% ± 0.72 | 96.18% ± 0.92 | 96.27% ± 0.48 | 95.16% ± 0.76 | 93.35% ± 1.28 | 92.14% ± 1.43 | 94.93% ± 0.60 | 94.18% ± 1.03 |
| OOD Cov ↓ | 3.55% ± 0.67 | 2.52% ± 0.68 | 2.74% ± 0.96 | 2.41% ± 0.66 | 2.28% ± 0.98 | 2.34% ± 0.86 | 2.98% ± 2.02 | 7.55% ± 1.33 | **1.77% ± 0.63** | 1.96% ± 0.59 | 2.00% ± 0.80 |
| Adv Cov ↓ | 61.14% ± 9.38 | 52.21% ± 9.49 | 47.58% ± 8.79 | 48.80% ± 8.75 | 50.40% ± 14.60 | 49.05% ± 11.90 | 28.74% ± 7.93 | 38.81% ± 4.23 | 21.62% ± 4.89 | 16.43% ± 5.52 | **15.51% ± 6.09** |
| Total Evidence | | | | | | | | | | | |
| ID Acc ↑ | 99.97% ± 0.01 | 99.96% ± 0.01 | 99.96% ± 0.01 | 99.96% ± 0.03 | 99.96% ± 0.02 | 99.95% ± 0.02 | 99.94% ± 0.04 | **99.99% ± 0.02** | 99.98% ± 0.01 | 99.96% ± 0.02 | 99.97% ± 0.01 |
| ID Cov ↑ | 95.25% ± 1.00 | 95.97% ± 0.82 | 96.18% ± 0.56 | 96.42% ± 0.56 | **96.48% ± 0.57** | 96.47% ± 0.57 | 95.32% ± 0.72 | 92.95% ± 1.85 | 93.10% ± 0.75 | 94.90% ± 0.47 | 94.21% ± 1.07 |
| OOD Cov ↓ | 3.20% ± 0.90 | 3.15% ± 1.69 | 2.53% ± 0.85 | 2.23% ± 0.63 | 2.15% ± 0.79 | 2.00% ± 0.94 | 2.48% ± 0.84 | 38.32% ± 2.74 | 7.90% ± 1.76 | 1.92% ± 1.03 | **1.67% ± 0.61** |
| Adv Cov ↓ | 58.34% ± 7.98 | 43.41% ± 10.42 | 47.41% ± 14.94 | 42.18% ± 6.51 | 49.67% ± 8.29 | 43.03% ± 7.43 | 39.95% ± 6.69 | 53.36% ± 2.67 | 29.42% ± 7.05 | 14.68% ± 3.56 | **14.28% ± 5.81** |
| Mutual Information | | | | | | | | | | | |
| ID Acc ↑ | 99.96% ± 0.02 | 99.96% ± 0.02 | 99.97% ± 0.04 | 99.96% ± 0.01 | 99.96% ± 0.02 | 99.96% ± 0.01 | 99.92% ± 0.03 | **99.98% ± 0.01** | **99.98% ± 0.01** | 99.97% ± 0.02 | 99.96% ± 0.02 |
| ID Cov ↑ | 95.13% ± 0.95 | 95.83% ± 1.19 | 96.11% ± 0.64 | 96.18% ± 0.73 | **96.52% ± 0.41** | 96.11% ± 0.43 | 95.79% ± 1.50 | 93.51% ± 1.12 | 92.05% ± 1.72 | 95.03% ± 0.71 | 94.22% ± 1.05 |
| OOD Cov ↓ | 3.51% ± 0.78 | 2.71% ± 1.61 | 2.40% ± 0.65 | 2.47% ± 0.90 | 2.95% ± 1.39 | 2.06% ± 0.56 | 2.16% ± 0.63 | 35.73% ± 3.62 | 5.80% ± 1.42 | 1.92% ± 0.93 | **1.53% ± 0.80** |
| Adv Cov ↓ | 61.12% ± 8.29 | 49.13% ± 13.12 | 52.02% ± 10.24 | 43.79% ± 12.54 | 44.07% ± 8.79 | 45.64% ± 6.96 | 43.19% ± 10.65 | 54.42% ± 3.77 | 23.39% ± 5.70 | 17.68% ± 7.64 | **17.62% ± 4.81** |

Figure 16: Visualisation of the conflict score $C$ as a function of $C_{\text{intra}}$, $C_{\text{inter}}$, and $\lambda$. As shown, $C$ increases smoothly with both $C_{\text{intra}}$ and $C_{\text{inter}}$, with monotonicity preserved for $\lambda \leq 0.5$.

## B.4 ABLATION ANALYSIS

Since C-EDL introduces a small number of additional hyperparameters, we conduct an ablation study to understand their individual effects on model performance. This analysis helps assess the robustness of C-EDL to hyperparameter choice and provides practical guidance for tuning in real-world deployments.

Figure 16 illustrates the theoretical behaviour of the conflict score $C$ as formalised in Theorem 1. The surface plots confirm that $C$ remains bounded in the open interval $(0, 1]$ and increases monotonically with both $C_{\text{intra}}$ and $C_{\text{inter}}$ when $\lambda \leq \frac{1}{2}$, validating the theorem's conditions. As $\lambda$ increases beyond this bound, the quadratic penalty term $-\lambda(C_{\text{inter}} - C_{\text{intra}})^2$ induces a curvature that distorts the surface and weakens monotonicity, especially along the diagonal. This justifies our design choice to set $\lambda = 0.5$, balancing the influence of asymmetric disagreement while preserving the desirable theoretical properties of the conflict score.

Table 9 presents an ablation study over the core hyperparameters introduced by C-EDL: the sharpness of $C_{inter}$ penalty $\beta$, the $C$ penalty $\lambda$, the scaling factor $\delta$, and the number of metamorphic transformations $T$.

We observe that as $\beta \in [0.5, 2.5]$ increases, both C-EDL variants show more conservatism. Adversarial coverage reduces, but so does ID coverage. The $C$ penalty $\lambda$, as seen in Figure 16, controls how increases in $C_{intra}$ and $C_{inter}$ affect the total measure $C$. We observe that as $\lambda$ increases up to 0.5, ID coverage increases at the expense of adversarial coverage, showing that both $C_{intra}$ and $C_{inter}$ need to be higher to have a greater effect on $C$. As $\lambda$ increases $< 0.5$, the guarantee of monotonicity is non-existent, and therefore, results begin to vary. The scaling factor $\delta$ controls how much evidence is reduced under high conflict. As $\delta$ increases, adversarial coverage drops for both C-EDL variants. For example, at $\delta = 0.25$, C-EDL (MC) has an adversarial coverage of $31.83\% \pm 4.92$, while at $\delta = 2.00$ it improves to $11.63\% \pm 6.36$. OOD coverage improves in a similar trend, at the expense of small drops in ID coverage. Finally, as the number of transformations $T$ increases, the adversarial and OOD coverage decreases, again, at the expense of small drops in ID coverage. However, inference time also grows, highlighting the trade-off between performance and efficiency. For reference, EDL has an average inference time of $1.15\text{s} \pm 0.10$. This and the inference time results displayed in Table 9 are based on inferring on the dataset as a whole, not individual data.

Table 9: The accuracy, OOD detection, and adversarial attack detection performance of C-EDL variants under ablations of hyperparameters ($\beta$, $\lambda$, $\delta$, $T$) introduced. The adversarial attack is L2PGD (1.0 maximum perturbation), and the ID and OOD datasets are MNIST and FashionMNIST.

| | C-EDL (MC) | | | | | C-EDL (Meta) | | | | |
|---|---|---|---|---|---|---|---|---|---|---|
| | $\beta$ Ablation | | | | | | | | | |
| | 0.50 | 1.00 | 1.50 | 2.00 | 2.50 | 0.50 | 1.00 | 1.50 | 2.00 | 2.50 |
| ID Acc ↑ | 99.99% ± 0.02 | 99.99% ± 0.01 | 99.98% ± 0.02 | 99.98% ± 0.01 | 99.99% ± 0.01 | 99.97% ± 0.02 | 99.97% ± 0.01 | 99.96% ± 0.01 | 99.98% ± 0.01 | 99.97% ± 0.01 |
| ID Cov ↑ | 92.10% ± 1.58 | 92.40% ± 1.44 | 92.14% ± 1.43 | 92.57% ± 1.40 | 92.01% ± 1.53 | 93.77% ± 0.76 | 94.25% ± 0.65 | 94.18% ± 1.03 | 93.83% ± 1.05 | 93.41% ± 1.00 |
| OOD Cov ↓ | 1.92% ± 0.41 | 2.15% ± 0.77 | 1.77% ± 0.63 | 1.77% ± 0.49 | 1.88% ± 0.48 | 1.83% ± 0.77 | 2.04% ± 0.85 | 2.00% ± 0.80 | 1.66% ± 0.80 | 1.61% ± 0.94 |
| Adv Cov ↓ | 22.38% ± 4.90 | 26.04% ± 5.21 | 21.62% ± 4.89 | 21.51% ± 6.30 | 21.51% ± 4.31 | 10.31% ± 2.81 | 12.60% ± 5.52 | 15.51% ± 6.09 | 12.74% ± 5.37 | 13.99% ± 7.45 |
| | $\lambda$ Ablation | | | | | | | | | |
| | 0.00 | 0.25 | 0.50 | 0.75 | 1.00 | 0.00 | 0.25 | 0.50 | 0.75 | 1.00 |
| ID Acc ↑ | 99.99% ± 0.01 | 99.99% ± 0.01 | 99.99% ± 0.02 | 99.99% ± 0.01 | 99.97% ± 0.01 | 99.97% ± 0.02 | 99.99% ± 0.01 | 99.96% ± 0.01 | 99.97% ± 0.02 | 99.97% ± 0.01 |
| ID Cov ↑ | 92.01% ± 1.08 | 92.11% ± 1.08 | 92.14% ± 1.43 | 91.92% ± 1.25 | 92.95% ± 0.59 | 94.40% ± 0.62 | 92.11% ± 1.08 | 94.18% ± 1.03 | 93.96% ± 0.59 | 94.55% ± 0.52 |
| OOD Cov ↓ | 1.37% ± 0.57 | 1.63% ± 0.57 | 1.77% ± 0.63 | 2.17% ± 0.69 | 2.76% ± 0.54 | 2.21% ± 1.02 | 1.63% ± 0.57 | 2.00% ± 0.80 | 1.70% ± 0.48 | 1.70% ± 0.65 |
| Adv Cov ↓ | 14.23% ± 3.95 | 19.74% ± 6.20 | 21.62% ± 4.89 | 21.55% ± 5.19 | 21.76% ± 2.86 | 14.66% ± 4.77 | 19.74% ± 6.20 | 15.51% ± 6.09 | 13.43% ± 5.19 | 15.56% ± 3.96 |
| | $\delta$ Ablation | | | | | | | | | |
| | 0.25 | 0.50 | 1.00 | 1.50 | 2.00 | 0.25 | 0.50 | 1.00 | 1.50 | 2.00 |
| ID Acc ↑ | 99.98% ± 0.02 | 99.98% ± 0.02 | 99.98% ± 0.02 | 99.98% ± 0.01 | 99.98% ± 0.01 | 99.98% ± 0.01 | 99.97% ± 0.01 | 99.96% ± 0.01 | 99.96% ± 0.01 | 99.98% ± 0.01 |
| ID Cov ↑ | 93.57% ± 1.34 | 92.43% ± 1.78 | 92.14% ± 1.43 | 90.59% ± 0.91 | 92.43% ± 1.89 | 94.64% ± 0.80 | 94.25% ± 0.71 | 94.18% ± 1.03 | 93.83% ± 0.93 | 93.87% ± 0.87 |
| OOD Cov ↓ | 4.82% ± 1.40 | 3.93% ± 1.06 | 1.77% ± 0.63 | 1.32% ± 0.61 | 0.80% ± 0.27 | 1.62% ± 0.56 | 1.62% ± 0.44 | 2.00% ± 0.80 | 1.79% ± 0.68 | 1.24% ± 0.64 |
| Adv Cov ↓ | 31.83% ± 4.92 | 27.21% ± 4.92 | 21.62% ± 4.89 | 17.14% ± 5.98 | 11.63% ± 6.36 | 12.08% ± 4.89 | 17.20% ± 6.16 | 15.51% ± 6.09 | 15.24% ± 6.93 | 10.54% ± 3.29 |
| | $T$ Ablation | | | | | | | | | |
| | 2.00 | 5.00 | 10.00 | 25.00 | 50.00 | 2.00 | 5.00 | 10.00 | 25.00 | 50.00 |
| ID Acc ↑ | 99.97% ± 0.02 | 99.98% ± 0.02 | 99.99% ± 0.01 | 100.00% ± 0.01 | 100.00% ± 0.00 | 99.97% ± 0.02 | 99.96% ± 0.01 | 99.97% ± 0.02 | 99.99% ± 0.01 | 99.99% ± 0.01 |
| ID Cov ↑ | 91.82% ± 1.53 | 92.14% ± 1.43 | 91.69% ± 1.73 | 84.77% ± 4.06 | 77.51% ± 5.86 | 94.85% ± 1.16 | 94.18% ± 1.03 | 93.08% ± 0.86 | 91.60% ± 0.53 | 90.79% ± 1.04 |
| OOD Cov ↓ | 7.46% ± 1.34 | 1.77% ± 0.63 | 0.78% ± 0.31 | 1.25% ± 0.35 | 1.66% ± 0.54 | 2.75% ± 1.20 | 2.00% ± 0.80 | 1.54% ± 0.60 | 1.25% ± 0.71 | 1.50% ± 1.01 |
| Adv Cov ↓ | 31.95% ± 4.93 | 21.62% ± 4.89 | 11.42% ± 3.62 | 15.49% ± 3.59 | 25.07% ± 4.50 | 23.39% ± 7.27 | 15.51% ± 6.09 | 8.22% ± 3.47 | 6.23% ± 3.44 | 4.32% ± 1.95 |
| Inference Time | 1.51s ± 0.11 | 2.26s ± 0.12 | 3.86s ± 0.12 | 8.32s ± 0.01 | 16.93s ± 0.16 | 2.53s ± 0.14 | 5.12s ± 0.10 | 9.18s ± 0.21 | 24.32s ± 0.24 | 48.32s ± 0.46 |

Table 10: The accuracy, OOD detection, and adversarial attack detection performance of dropout strength in the EDL++ (MC) and C-EDL (MC) models. The adversarial attack is L2PGD (1.0 maximum perturbation), and the ID and OOD datasets are MNIST and FashionMNIST.

| | EDL++ (MC) | | | C-EDL (MC) | | |
|---|---|---|---|---|---|---|
| | Dropout Rate | | | | | |
| | 0.10 | 0.25 | 0.50 | 0.10 | 0.25 | 0.50 |
| ID Acc ↑ | 99.97% ± 0.02 | 99.97% ± 0.01 | 99.99% ± 0.01 | 99.97% ± 0.01 | 99.98% ± 0.02 | 99.99% ± 0.01 |
| ID Cov ↑ | 94.00% ± 1.33 | 93.35% ± 1.28 | 87.74% ± 1.63 | 93.79% ± 1.37 | 92.14% ± 1.43 | 85.60% ± 1.88 |
| OOD Cov ↓ | 13.76% ± 2.55 | 7.55% ± 1.33 | 4.82% ± 0.97 | 3.82% ± 1.77 | 1.77% ± 0.63 | 1.42% ± 0.46 |
| Adv Cov ↓ | 48.26% ± 6.38 | 38.81% ± 4.23 | 26.16% ± 3.78 | 33.83% ± 7.85 | 21.62% ± 4.89 | 14.59% ± 3.91 |

Table 10 investigates the sensitivity of EDL++ (MC) and C-EDL (MC) to varying dropout strengths. Across both models, increasing the dropout strength from 0.1 to 0.50 induces a consistent trend: OOD and adversarial coverage decrease rapidly, while ID coverage moderately decreases. For EDL++ (MC) that lacks the conflict reduction, the trend is less pronounced, once again showing that the conflict-aware reduction is necessary to improves robustness to anomalous data.

Following the investigation of the dropout strength in EDL++ (MC) and C-EDL (MC) (Table 10), Table 11 investigates the impact of metamorphic transformations in the EDL++ (Meta) and C-EDL (Meta) variants. Across all transformation types and intensities, we observe that both EDL++ (Meta) and C-EDL (Meta) exhibit less OOD and adversarial coverage at the expense of ID coverage. For example, when only the rotation transformation is used and increases from $\pm 5°$ to $\pm 30°$, adversarial coverage for EDL++ (Meta) reduces by $-21.03\% \pm 2.66$ at the expense of a $-8.62\% \pm -1.33$ reduction in ID coverage.

The ablation of individual transformations shows that on their own, all transformation types aid in detecting anomalous data with different magnitudes in reduction of ID, OOD, and adversarial coverage. However, when combined, ID coverage remains moderately high, with a large prediction in OOD and adversarial coverage showing that the combination of transformations effectively balances all coverage types.

The inference time against the adversarial coverage of comparative methods can be seen in Table 12. C-EDL (MC $T = 2$) achieves an adversarial coverage of 31.95% with a total inference time of 1.51 seconds for the entire dataset, corresponding to approximately 0.11 seconds per input. In contrast, C-EDL (Meta $T = 5$) yields substantially lower adversarial coverage of 15.51% while requiring 5.12 seconds for full-dataset inference, or about 0.15 seconds per input. For reference, the baseline EDL method consumes roughly 0.103 seconds per input (equivalent to around ten inferences per second), whereas the alternative post-hoc approach S-EDL (with five perturbed samples to ensure comparability) requires about 0.49 seconds per input, amounting to only two inferences per second. These results indicate that while C-EDL introduces some computational overhead, the increase is

Table 11: The accuracy, OOD detection, and adversarial attack detection performance of augmentation types and strength in the EDL++ (Meta) and C-EDL (Meta) models. The adversarial attack is L2PGD (1.0 maximum perturbation), and the ID and OOD datasets are MNIST and FashionMNIST.

| | EDL++ (Meta) | | | C-EDL (Meta) | | |
|---|---|---|---|---|---|---|
| | Rotate Only | | | | | |
| | $\pm 5°$ | $\pm 15°$ | $\pm 30°$ | $\pm 5°$ | $\pm 15°$ | $\pm 30°$ |
| ID Acc ↑ | 99.95% ± 0.02 | 99.96% ± 0.01 | 99.97% ± 0.02 | 99.96% ± 0.02 | 99.96% ± 0.02 | 99.97% ± 0.02 |
| ID Cov ↑ | 95.99% ± 0.60 | 94.84% ± 0.77 | 87.37% ± 1.93 | 95.72% ± 0.94 | 94.28% ± 0.67 | 84.04% ± 1.49 |
| OOD Cov ↓ | 2.12% ± 0.50 | 1.70% ± 0.60 | 1.32% ± 0.71 | 2.33% ± 0.97 | 1.75% ± 0.68 | 0.78% ± 0.37 |
| Adv Cov ↓ | 22.48% ± 4.39 | 6.34% ± 4.55 | 1.45% ± 1.73 | 18.55% ± 7.41 | 4.07% ± 2.00 | 0.97% ± 0.95 |
| | Shift Only | | | | | |
| | $\pm 1$ | $\pm 2$ | $\pm 4$ | $\pm 1$ | $\pm 2$ | $\pm 4$ |
| ID Acc ↑ | 99.96% ± 0.02 | 99.98% ± 0.01 | 99.99% ± 0.01 | 99.97% ± 0.01 | 99.98% ± 0.02 | 99.99% ± 0.02 |
| ID Cov ↑ | 95.63% ± 0.77 | 92.84% ± 1.31 | 71.71% ± 2.42 | 95.12% ± 0.86 | 91.80% ± 0.92 | 62.58% ± 3.10 |
| OOD Cov ↓ | 2.19% ± 0.81 | 2.09% ± 1.17 | 0.93% ± 0.30 | 2.25% ± 0.73 | 1.46% ± 0.49 | 1.07% ± 0.44 |
| Adv Cov ↓ | 16.92% ± 5.88 | 11.68% ± 5.49 | 3.53% ± 1.56 | 12.10% ± 3.66 | 8.87% ± 3.88 | 2.06% ± 0.89 |
| | Noise Only | | | | | |
| | 0.005 | 0.010 | 0.020 | 0.005 | 0.010 | 0.020 |
| ID Acc ↑ | 99.96% ± 0.01 | 99.96% ± 0.02 | 99.96% ± 0.01 | 99.95% ± 0.02 | 99.95% ± 0.02 | 99.96% ± 0.01 |
| ID Cov ↑ | 95.88% ± 1.02 | 96.43% ± 0.68 | 96.25% ± 0.47 | 96.03% ± 0.79 | 96.46% ± 0.24 | 95.99% ± 0.64 |
| OOD Cov ↓ | 2.28% ± 0.63 | 2.07% ± 0.85 | 1.86% ± 0.77 | 2.40% ± 0.99 | 1.81% ± 0.66 | 2.37% ± 1.12 |
| Adv Cov ↓ | 49.88% ± 9.61 | 43.44% ± 9.52 | 31.64% ± 7.10 | 45.59% ± 7.03 | 42.20% ± 12.14 | 34.28% ± 11.10 |
| | All Combined (Rotate $\pm 15°$, Shift $\pm 2$, Noise 0.010) | | | | | |
| ID Acc ↑ | 99.98% ± 0.02 | | | 99.96% ± 0.01 | | |
| ID Cov ↑ | 92.14% ± 1.43 | | | 94.18% ± 1.03 | | |
| OOD Cov ↓ | 1.77% ± 0.63 | | | 2.00% ± 0.80 | | |
| Adv Cov ↓ | 21.62% ± 4.89 | | | 15.51% ± 6.09 | | |

Table 12: The adversarial attack detection performance and inference time of comparative methods. The adversarial attack is L2PGD (1.0 maximum perturbation), and the ID and OOD datasets are MNIST and FashionMNIST.

| | Post Net | EDL | I-EDL | S-EDL | H-EDL | R-EDL | DA-EDL | C-EDL (MC T=2) | C-EDL (MC T=5) | C-EDL (Meta T=2) | C-EDL (Meta T=5) |
|---|---|---|---|---|---|---|---|---|---|---|---|
| Adversarial Coverage | 61.14% | 52.21% | 47.58% | 48.80% | 50.40% | 49.05% | 28.74% | 31.95% | 21.62% | 23.39% | 15.51% |
| Inference Time (entire dataset) | 1.17s | 1.15s | 1.19s | 63.36s | 1.17s | 1.17s | 0.78s | 1.51s | 2.26s | 2.53s | 5.12s |

relatively modest and remains significantly more efficient than S-EDL, the only other state-of-the-art post-hoc method considered. This highlight a clear trade-off between inference time and adversarial coverage, with a practical balance emerging around $T = 5$. Nevertheless, the choice of $T$ should ultimately be guided by the anticipated real-time constraints of the deployment setting, allowing practitioners to adjust inference time per input while maintaining the best attainable coverage.

To show that inference time scales with model size and dataset size, we measure inference time (seconds) for C-EDL (Meta) on the MNIST and CIFAR-10 datasets for various $T$ values for both the large LeNet and SeNet models in Table 13.

## C  EXPERIMENT DETAILS

This section provides comprehensive details of our experimental setup to support reproducibility and clarify the evaluation procedures used throughout the paper. We begin by describing the datasets used across ID and OOD in the evaluation, followed by details of the threshold scoring metrics used for uncertainty-based rejection. We then describe the comparative approaches included in our evaluation

Table 13: Inference time (seconds) for C-EDL (Meta) on the MNIST and CIFAR-10 datasets for various $T$ values for both the large LeNet and SeNet models.

| | C-EDL (Meta) - LeNet (0.45M params) | | | | | C-EDL (Meta) - SeNet (9.79M params) | | | | |
|---|---|---|---|---|---|---|---|---|---|---|
| Dataset | 2.00 | 5.00 | 10.00 | 25.00 | 50.00 | 2.00 | 5.00 | 10.00 | 25.00 | 50.00 |
| MNIST | 2.53s ± 0.14 | 5.12s ± 0.10 | 9.18s ± 0.21 | 24.32s ± 0.24 | 48.32s ± 0.46 | 35.20s ± 0.58 | 71.79s ± 1.12 | 135.90s ± 1.26 | 324.24s ± 2.33 | 621.82s ± 4.31 |
| CIFAR-10 | 3.40s ± 0.42 | 6.96s ± 0.56 | 13.62s ± 0.70 | 33.81s ± 0.22 | 58.32s ± 0.54 | 41.24s ± 0.36 | 82.91s ± 1.31 | 151.09s ± 1.59 | 392.16s ± 2.76 | 687.05s ± 6.17 |

and provide detailed information about model training. Finally, we include task-specific experimental configurations and implementation details relevant to each setup.

## C.1 DATASETS

We evaluate our approaches across a diverse collection of widely used computer vision benchmarks, spanning a range of domains and task difficulties. This enables a comprehensive evaluation of model robustness and uncertainty estimation under varying conditions. To rigorously test the ability of models to distinguish ID from OOD samples, we evaluate in both near-OOD and far-OOD settings. Near-OOD datasets exhibit some degree of class overlap with the ID data, while far-OOD datasets involve distinct visual domains. Dataset-specific details, including sample sizes, input resolutions, class labels, and split protocols, are provided below. Example images from each dataset can be found in Figure 17.

- **MNIST (LeCun et al., 1998):** consists of 28x28 greyscale images of handwritten digits (0-9) spanning 10 classes. We utilise the widely used standard split of 60,000 training samples, 8000 test samples, and 2000 validation samples. Pixel normalisation was applied, bounded $[0, 1]$. In our experimental evaluation, MNIST serves as one of the ID datasets. MNIST was paired with FashionMNIST and KMNIST as far-OOD datasets due to them sharing no overlapping classes, but exhibit similar visual characteristics, including greyscale appearance, low resolution, and hand-drawn style. MNIST was also paired with EMNIST as a near-OOD dataset due to it sharing the same visual characteristics and sharing overlapping classes with EMNIST containing handwritten digits in addition to handwritten letters.
- **FashionMNIST (Xiao et al., 2017):** consists of 28x28 greyscale images of clothing items (e.g., coats, bags, t-shirts, etc.) spanning 10 classes. We utilise the widely used standard split of 60,000 training sample, 8000 test samples, and 2000 validation samples. Pixel normalisation was applied, bounded $[0, 1]$. In our experimental evaluation, we use this dataset as a far-OOD pairing with the ID MNIST dataset.
- **KMNIST (Clanuwat et al., 2018):** consists of 28x28 greyscale images of handwritten Japanese characters (specifically Kuzushiji characters from classical Japanese literature) spanning 10 classes. We utilise the widely used standard split of 60,000 training samples, 8000 test samples, and 2000 validation samples. Pixel normalisation was applied, bounded $[0, 1]$. In our experimental evaluation, we use this dataset as a far-OOD pairing with the ID MNIST dataset.
- **EMNIST (Cohen et al., 2017):** consists of 28x28 greyscale images of both handwritten letters (from the Latin alphabet) and digits (0-9) spanning 47 classes. We use a split of 112,800 training samples, 15,040 test samples, and 3760 validation samples. Pixel normalisation was applied, bounded $[0, 1]$. In our experimental evaluation, we use this dataset as a near-OOD pairing with the ID MNIST dataset due to it sharing the same visual characteristics and overlapping classes with MNIST (handwritten digits).
- **CIFAR10 (Krizhevsky et al., 2009):** consists of 32x32 RGB images of real-world objects (e.g, birds, trucks, airplanes, frogs, etc) spanning 10 classes. We utilise the widely used standard split of 50,000 training samples, 8000 test samples, and 2000 validation samples. Pixel normalisation was applied, bounded $[0, 1]$ across all three RGB channels. In our experimental evaluation, CIFAR10 serves as one of the ID datasets. CIFAR10 was paired with FashionMNIST and SVHN as a far-OOD dataset due to them sharing no overlapping classes, but exhibiting similar visual characteristics, including coloured appearance, real-world pictures. CIFAR-10 was also paired with CIFAR-100 as a near-OOD dataset, due to the datasets sharing the same visual characteristics and overlapping classes. Specifically, CIFAR-10 contains broad object categories (e.g., cat, dog, truck, ship), which correspond to finer-grained classes like (e.g., house cat, beagle, pickup truck, cruise ship) in CIFAR-100.
- **CIFAR100 (Krizhevsky et al., 2009):** consists of 32x32 RGB images of real-world objects spanning of 100 classes that are fine-grained counterparts of the classes from CIFAR10. For example, CIFAR10 has the class *truck*, whilst CIFAR100 has the classes *pickup truck* and *train*. We utilise the widely used standard split of 50,000 training samples, 8000 test samples, and 2000 validation samples. Pixel normalisation was applied, bounded $[0, 1]$ across all three RGB channels. In our experimental evaluation, we use this dataset as a near-OOD pairing with the ID CIFAR10 dataset due to it sharing the same visual characteristics and overlapping classes.

- **SVHN (Netzer et al., 2011):** consists of 32x32 RGB images of house numbers collected from Google Street View spanning 10 classes (0-9). We use a split of 73,257 training samples, 10,832 test samples, and 5200 validation samples. Pixel normalisation was applied, bounded $[0, 1]$ across all three RGB channels. In our experimental evaluation, we use this dataset as a far-OOD pairing with the ID CIFAR10 dataset due to the substantial domain shift between street-level digit photographs and object-centric natural images.
- **Oxford Flowers (Nilsback & Zisserman, 2008):** consists of 64x64 RGB images of flowers commonly found in the United Kingdom, spanning 102 (e.g., Water Lily, Wild Pansy, etc) classes. We utilise a split of 9826 training samples, 1638 test samples, and 1638 validation samples. This is double the size of the original dataset, as each image has been duplicated with a random augmentation to aid generalisation within training. Pixel normalisation was applied, bounded $[0, 1]$ across all three RGB channels. In our experimental evaluation, we use this dataset as an ID dataset. Oxford Flowers was paired with Deep Weeds because they share no overlapping classes but exhibit similar visual characteristics, including features found in nature (leaves, flowers, etc.). Though smaller in total images, this dataset offers higher class granularity and lower per-class shot count than TinyImageNet, making it a compelling test for fine-grained UQ.
- **Deep Weeds (Olsen et al., 2019):** consists of 64x64 RGB images of species of weeds found in Australia, spanning over 8 classes (e.g., Snake weed, Rubber vine, etc). We use the standard split of 10,505 training samples, 3502 test samples, and 3502 validation samples. Pixel normalisation was applied, bounded $[0, 1]$ across all three RGB channels. In our experimental evaluation, Deep Weeds serves as a far-OOD pairing with the ID Oxford Flowers dataset due to the substantial domain shift between photographs of flowers and photographs of weeds.
- **Tiny-ImageNet (Le & Yang, 2015):** consists of 64x64 RGB images across 200 object classes, each corresponding to a subset of ImageNet. The dataset contains 100,000 training samples (500 per class), 10,000 validation samples (50 per class), and 10,000 test samples without publicly available labels. Pixel values were normalised to the range $[0, 1]$ across all RGB channels. In our experimental evaluation, Tiny-ImageNet is used as an ID dataset. Tiny-ImageNet was paired with Caltech-UCSD Birds-200-2011 because they share very limited overlapping classes but exhibit similar visual characteristics.
- **Caltech-UCSD Birds-200-2011 (CUB) (Welinder et al., 2010):** consists of 64x64 RGB images of 200 bird species, collected from various natural environments. The dataset contains a total of 11,788 images, with a standard split of 5,994 training samples and 5,794 test samples. Pixel values were normalised to the range $[0, 1]$ across all RGB channels. In our experimental evaluation, CUB is used as an OOD dataset paired with Tiny-ImageNet.

In some cases where the dataset splits were not already provided, manual splits were stratified to preserve class distribution within each subset. Additionally, for all experiments, validation splits from each dataset were used to determine the ID-OOD threshold for uncertainty-based abstention, which allowed ID and OOD coverage to be determined on the test splits.

## C.2 ID-OOD Thresholds

To enable abstention of uncertain predictions, an optimal ID–OOD threshold was calculated using the validation datasets associated with each experimental setting. We consider multiple ID-OOD scoring metrics in our experimental evaluation to allow for a robust evaluation of C-EDL and the comparative approaches.

- **Differential Entropy:** quantifies the spread or uncertainty of the Dirichlet distribution. It is computed as:

$$\sum_{k=1}^{K} \ln \Gamma(\alpha_k) - \ln \Gamma(S) - \sum_{k=1}^{K} (\alpha_k - 1)(\Psi(\alpha_k) - \Psi(S))$$

where $B(\boldsymbol{\alpha})$ is the multivariate Beta function, $S$ is the total concentration parameter, and $\Psi(\cdot)$ denotes the digamma function. A higher differential entropy corresponds to greater uncertainty, and thus larger values are expected for OOD samples. This scoring metric only works on Dirichlet-based models (e.g., posterior networks, evidential networks).
- **Total Evidence:** quantifies the sum of total Dirichlet parameters $\alpha$ across all classes $K$. It is computed as $\sum_{k=1}^{K} \alpha_k$. A higher total evidence corresponds to less uncertainty in

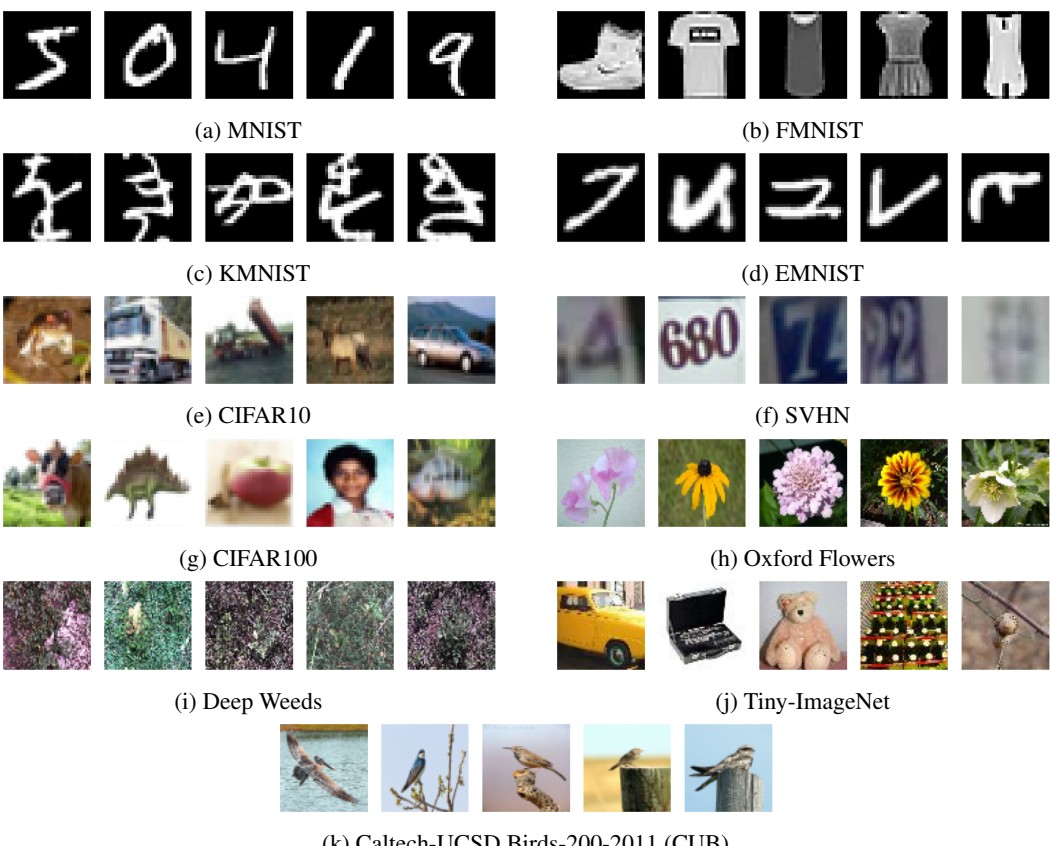

(a) MNIST

(b) FMNIST

(c) KMNIST

(d) EMNIST

(e) CIFAR10

(f) SVHN

(g) CIFAR100

(h) Oxford Flowers

(i) Deep Weeds

(j) Tiny-ImageNet

(k) Caltech-UCSD Birds-200-2011 (CUB)

Figure 17: Example images from the datasets evaluated upon.

Dirichlet-based models, and thus larger values are expected for ID samples. This scoring metric only works on Dirichlet-based models (e.g., posterior networks, evidential networks).

- **Mutual Information:** quantifies epistemic uncertainty by measuring how much information the model parameters provide about the predictive distribution, computed as the difference between the total predictive entropy and the expected conditional entropy under the posterior over parameters.

$$-\sum_{k=1}^{K} \frac{\alpha_k}{S} \left( \ln \frac{\alpha_k}{S} - \psi(\alpha_k + 1) + \psi(S + 1) \right)$$

where $\Psi(\cdot)$ denotes the digamma function.

Following prior work (Deng et al., 2023; Kopetzki et al., 2021; Yoon & Kim, 2024), to calculate the optimal ID-OOD threshold, a scoring metric (chosen above) was used to derive scores for the ID and OOD validation datasets. Then, a receiver operating characteristic (ROC) curve based on these scores is constructed where ID samples are treated as positive instances, and OOD samples are treated as negative instances. This curve provides a way to visualise the separation between the ID and OOD distributions based on the chosen scoring metric. We extend this analysis by computing the score threshold that maximises TPR − FPR to provide the optimal ID-OOD separation point. This procedure ensures a balanced trade-off between correctly retaining ID samples and correctly rejecting OOD samples without manual tuning and facilitates the reporting of ID and OOD coverage for deployment/realistic scenarios.

Once obtained, the optimal threshold is fixed and used during evaluation on the test datasets. Test samples are rejected if their score (based on the chosen scoring metric) exceeds the threshold, allowing for consistent evaluation of both ID retention and OOD rejection performance across all approaches and datasets.

C.3 COMPARATIVE APPROACHES

To evaluate the effectiveness of C-EDL, we compare it against a range of recent EDL and uncertainty quantification approaches that are representative of the current state-of-the-art. Each approach is briefly described below, alongside any implementation-specific details or hyperparameter choices used in our experiments:

- **Posterior Networks (Charpentier et al., 2020):** represent a foundational approach to uncertainty estimation in classification that predates and informs the development of evidential deep learning. They avoid the need for OOD data during training by directly modelling a closed-form posterior over categorical distributions. Specifically, it predicts Dirichlet concentration parameters $\alpha = \beta_{\text{prior}} + \beta$ for each input $x$, where $\beta$ are pseudo-counts derived from class-conditional density estimates in a learned latent space, and $\beta_{\text{prior}}$ is a fixed symmetric prior. The resulting Dirichlet distribution encodes both aleatoric and epistemic uncertainty in closed form.

- **Evidential Deep Learning (EDL) (Sensoy et al., 2018):** proposes a deterministic alternative to Bayesian neural networks for uncertainty estimation by modelling class probabilities with a Dirichlet distribution whose parameters are derived from the model's non-negative outputs. The model is trained to minimise a loss by combining squared prediction error, variance, and a KL divergence with a uniform prior. This evidential formulation allows the model to quantify both aleatoric and epistemic uncertainty in closed form, without requiring sampling or OOD examples during training. This approach serves as the base EDL approach for the following comparative approaches, including our proposed EDL++ and C-EDL approaches. Due to this being the base approach, hyperparameters are shared across the remaining comparative approaches for fair comparison and are discussed in Section C.5.

- **Fisher Information-Based Evidential Deep Learning ($\mathcal{I}$-EDL) (Deng et al., 2023):** extends EDL by incorporating the Fisher Information Matrix (FIM) to adaptively weight the loss based on the informativeness of predicted evidence. The key idea is that classes with higher evidence carry less Fisher information and should be regularised less strictly. The approach introduces a regularisation term that penalises the log-determinant of the FIM to prevent overconfident predictions:

$$\mathcal{L}_i^{|\mathcal{I}|} = \sum_{j=1}^K \log \psi^{(1)}(\alpha_{ij}) + \log\left(1 - \frac{\psi^{(1)}(\alpha_{i0})}{\sum_{j=1}^K \psi^{(1)}(\alpha_{ij})}\right)$$

  In our experiments, we set $\lambda_{|\mathcal{I}|} = 0.001$ and annealed it linearly over the first 10 epochs, following guidance in Appendix C.2 of the original paper and our manual tuning.

- **Smoothed Evidential Deep Learning (S-EDL) (Kopetzki et al., 2021):** enhances the robustness of standard evidential models by applying median smoothing to their uncertainty estimates. Given an input $x$, a set of noisy samples $\{x_s\} \sim \mathcal{N}(x, \sigma)$ is generated. The final uncertainty estimate is obtained by taking the median over this set, which improves robustness to adversarial perturbations by mitigating the influence of outliers. In our experiments, we followed the original setup and set the Gaussian noise scale to $\sigma = 0.01$, using 50 samples per input following guidance of the original paper and our manual tuning. S-EDL is a purely post-hoc approach, similar to the proposed EDL++ and C-EDL approaches.

- **Hyper-Opinion Evidential Deep Learning (H-EDL) (Qu et al., 2024):** augments classical EDL by incorporating *hyper-opinions*, a generalisation of multinomial opinions from Subjective Logic that can express uncertainty over both singleton classes and composite subsets. This enables H-EDL to extract not only sharp evidence (supporting one class) but also vague evidence (supporting multiple plausible classes), thereby improving robustness to ambiguous inputs. A novel opinion projection mechanism is introduced to convert hyper-opinions into standard Dirichlet-based predictions, enabling training within the conventional EDL framework. This projection addresses the vanishing gradient issue that limits traditional EDL on fine-grained tasks. In our experiments, we adopted the two-stage training procedure described in the original paper, without introducing additional hyperparameters.

- **Relaxed Evidential Deep Learning (R-EDL) (Chen et al., 2024):** addresses overconfidence in standard EDL by relaxing two nonessential assumptions. First, it replaces the fixed prior weight (typically set to the number of classes) with a tunable scalar hyperparameter $\lambda$, controlling the contribution of the base rate $\alpha$ in the Dirichlet construction:

$$\alpha(x) = e(x) + \lambda$$

Second, R-EDL removes the variance-penalising regularisation from the EDL loss, instead directly optimising the projected class probabilities $P(x) = \frac{\alpha(x)}{S}$ to match one-hot labels. This simplification better balances evidence magnitude and proportion, particularly in OOD settings. In our experiments, we followed the guidance from Appendix C.2 and Figure 1(b) of the original paper and set $\lambda = 0.1$ as a fixed prior weight throughout training.

- **Density-Aware Evidential Deep Learning (DA-EDL) (Yoon & Kim, 2024):** extends EDL by incorporating feature space density into the uncertainty estimation process, thereby enabling distance-aware predictions. During training, DA-EDL follows the same loss function as standard EDL. At test time, it estimates the feature space density $s(x) \in [0, 1]$ of a test input using Gaussian Discriminant Analysis (GDA) fitted on the training features. This density is then used to scale the logits before applying the exponential activation, yielding concentration parameters:

$$\alpha(x) = \exp(g_\phi(f_\theta(x)) \cdot s(x))$$

where $f_\theta$ is the feature extractor and $g_\theta$ is the classifier. This adjustment ensures predictive uncertainty increases as the distance from the training data increases.

## C.4 Adversarial Attacks

This section outlines the adversarial attacks evaluated in our study to assess the robustness of uncertainty-based models. Adversarial attacks refer to small, intentionally crafted perturbations that can cause a model to behave incorrectly while maintaining high prediction confidence. This poses a significant threat to uncertainty-aware systems, as such perturbations can manipulate both aleatoric and epistemic uncertainty, making adversarial examples appear indistinguishable from ID samples or evading detection thresholds.

To comprehensively evaluate robustness, we consider both gradient-based and non-gradient-based attacks. Gradient-based attacks are typically used in white-box scenarios and leverage access to the model's gradients to construct targeted perturbations. Non-gradient-based attacks, in contrast, apply random or structured noise to inputs and are more commonly used in black-box threat models. We utilise Foolbox (Rauber et al., 2017) to implement the following attack strategies:

- **L2 Projected Gradient Descent (L2PGD):** is an iterative, white-box adversarial attack that perturbs inputs within a bounded $L_2$-norm ball to maximise the model's loss. At each iteration, the input is updated in the direction of the gradient of the loss with respect to the input, followed by projection back onto the $L_2$-ball of radius $\epsilon$. This results in smooth, high-precision perturbations that remain less perceptible to humans:

$$x'_{t+1} = \text{Proj}_\epsilon^{L_2}(x'_t + \alpha \cdot \nabla_x J(x'_t, y))$$

where $\alpha$ is the step size and $J(x, y)$ is the loss function.
- **Fast Gradient Sign Method (FGSM):** is a single-step white-box adversarial attack that perturbs the input in the direction of the sign of the gradient of the loss:

$$x' = x + \epsilon \cdot \text{sign}(\nabla_x J(x, y))$$

where $\epsilon$ controls the perturbation magnitude. FGSM generates perceptible but targeted perturbations with minimal computational overhead.
- **Salt & Pepper Noise:** is a non-gradient-based black-box perturbation that randomly sets a proportion of input pixels to their minimum or maximum value. This form of structured noise simulates impulsive corruption and tests the model's resilience to sparse, high-intensity artefacts. It does not rely on model gradients and is agnostic to internal model parameters.

## C.5 Training Details

To ensure consistency and reproducibility across all experiments, we adopt a fixed training configuration and architectural setup for all evidential models, including our proposed C-EDL variants and baseline comparators.

All models were trained using a custom convolutional neural network architecture, very closely in design to a LeNet architecture. The model consists of two convolutional blocks with 32 and 64 filters respectively, each using $3 \times 3$ kernels with ReLU activations and L2 regularisation ($\lambda = 1e - 4$). Each block include batch normalisation layers, $2 \times 2$ max pooling layers, and dropout layers

($\lambda = 0.25$) (Srivastava et al., 2014). The final feature maps are flattened and passed through two fully connected dense layers of $120$ and $84$ units, each followed by the same batch normalisation and dropout layers used earlier. The output layer consists of $K$ units (one per class according to the respective ID dataset), with a Softplus activation to ensure non-negative evidence outputs for evidential modelling.

Each evidential model is trained using the evidential loss (Sensoy et al., 2018) (except where additional modifications are specified in Section C.3) based on the mean squared error between the target one-hot label and the expected class probabilities under the Dirichlet posterior, augmented with a Kullback–Leibler (KL) divergence regularisation term between the predicted Dirichlet distribution and a non-informative prior:

$$\mathcal{L} = \underbrace{\mathbb{E}_{\text{Dir}(\boldsymbol{\alpha})}\left[\|\mathbf{y} - \mathbf{p}\|^2\right]}_{\text{Prediction error}} + \underbrace{\mathbb{E}_{\text{Dir}(\boldsymbol{\alpha})}\left[\text{Var}(\mathbf{p})\right]}_{\text{Uncertainty penalty}} + \lambda \cdot \underbrace{\text{KL}\left[\text{Dir}(\boldsymbol{\alpha}) \parallel \text{Dir}(\mathbf{1})\right]}_{\text{Regularisation}}$$

$$= \sum_{k=1}^{K}\left((y_k - m_k)^2 + \frac{m_k(1 - m_k)}{S + 1}\right) + \lambda \cdot \text{KL}(\text{Dir}(\boldsymbol{\alpha}) \parallel \text{Dir}(\mathbf{1})) \tag{26}$$

where $\alpha = \mathbf{e} + 1$ is the Dirichlet concentration, $m_k = \alpha_k/S$ is the expected probability, Dir(1) is the non-informative prior ($\beta$ in Posterior Networks), and $\lambda$ is the annealed KL weight. The KL divergence term is weighted by a scalar coefficient, which is annealed linearly from 0 to 1 over the first 10 epochs to encourage stable training. The KL formulation follows the standard evidence-based Dirichlet variational regularisation commonly used in evidential deep learning:

$$\text{KL}(\text{Dir}(\boldsymbol{\alpha}) \parallel \text{Dir}(\mathbf{1})) = \log \frac{\Gamma\left(\sum_{k=1}^{K} \alpha_k\right)}{\prod_{k=1}^{K} \Gamma(\alpha_k)} + \sum_{k=1}^{K} (\alpha_k - 1)\left[\psi(\alpha_k) - \psi\left(\sum_{j=1}^{K} \alpha_j\right)\right] \tag{27}$$

where $\Gamma(\cdot)$ is the gamma function, $\psi(\cdot)$ is the digamma function, and $\mathbf{1}$ is a uniform Dirichlet prior.

Models are trained using the Adam optimiser (Kingma & Ba, 2014) with an initial learning rate of $0.001$. We apply a learning rate scheduling strategy via ReduceLROnPlateau, which halves the learning rate upon stagnation in validation loss (patience $= 5$ epochs, min learning rate $= 1e-8$). All models are trained for 250 epochs with a batch size of 64. A KL annealing callback is implemented to gradually scale the regularisation weight across early epochs, improving convergence.

All experiments were implemented in TensorFlow 2.15 and executed on a large performance GPU cluster using a maximum of three Nvidia A40 GPUs, 32 CPU cores, 167GB of memory (per GPU). All models were trained from scratch, and no pretraining or external data was used. All runs used random seeds.

## C.6 EXPERIMENTAL SETUPS

This subsection outlines additional implementation details and experimental choices that are specific to certain experiments that may help with reproducibility and transparency.

Firstly, for all experiments unless stated otherwise, EDL++ utilises the hyperparameter combination of $T = 5$. EDL++ (MC) has a dropout strength of 0.25, and EDL++ (Meta) uses the rotate ($\pm 15°$), shift ($\pm 2$), and noise (0.01) transformations. Both C-EDL variants use the same setup but with the additional hyperparameter combination of $\beta = 1.5$, $\lambda = 0.5$, and $\delta = 1.0$.

In Table 1, the adversarial attack used is a gradient-based L2PGD attack. For MNIST to FashionMNIST, KMNIST, and EMNIST, a maximum perturbation of $1.0$ was used. For CIFAR10 to SVHN, CIFAR10 to CIFAR100, and Oxford Flowers to Deep Weeds, a maximum perturbation of $0.1$ was used. This choice reflects the fact that adversarial examples on natural image datasets such as CIFAR-10 require smaller perturbations to significantly degrade model performance, due to the increased complexity and lower inherent separability of the visual features. In contrast, MNIST-like datasets typically require larger perturbations to achieve a comparable adversarial effect.

For the $\Delta$ experiments (Table 3, and Figures 5, 11a, and 11b), we consider all data within the test dataset not just the data that falls above or below the abstention threshold.

