# OpenReview forum: "Robust Adversarial Quantification via Conflict-Aware Evidential Deep Learning"
_ICLR.cc/2026/Conference — ICLR 2026 Poster_

### Official Review · Reviewer_g4uc · 2025-10-22

**Soundness:** 4
**Presentation:** 4
**Contribution:** 4
**Rating:** 8
**Confidence:** 5

**Summary:**

The paper addresses the sensitivity of the evidential deep learning approach to input perturbations, particularly to the adversarial ones. The gist of the proposed solution is to apply a variant of self-supervised learning criterion to the adversarial robustness use case in the evidential learning context for post-hoc calibration. Inputs are augmented in diverse ways and repeatedly fed into an evidential network. The resulting Dirichlet parameters are calibrated after training with a conflict adjustment formula that promotes consistently low uncertainty scores among the perturbed versions of the same input for all classes (Eq 4) and high scores in other cases (Eq 5).

**Strengths:**

This is a very solid piece of work which makes a spot-on problem definition, provides a well-grounded solution to it no matter how simple, and demonstrates the efficacy of the solution on a comprehensive set of experiments where actually everything I can think of as necessary has been already considered. The C-EDL model clearly improves the state of the art in robust implementations of EDL as visible from the main results (Table 1) as well as all the downstream side analyses.

**Weaknesses:**

I can follow the rationale behind Eq 5 and it makes sense. I am only not sure that this is the simplest possible way the intended goal is achieved. Extending the related part of the paper (around Lines 239-256) will contribute to its readability. I do understand that Theorem 1 clearly demonstrates that the criterion behaves in the intended way. The question here is how close the proposed solution is the simplest possible one.

**Questions:**

Did the authors consider alternative criteria such as an adaptation of InfoNCE, Barlow Twins, or IMAX?

Evidential learning is also being used in regression [1]. In that case, as we don't have a Dirichlet distribution, I understand that the suggested conflict adjustment criterion will not directly translate. Do the authors have any insight about which basic principles of the suggested contribution can be extended to the regression case?

[1] Amini et al., Deep evidential regression, NeurIPS, 2020

---

> ### Author Response · Authors · 2025-11-20
>
> We thank the reviewer for their positive review of C-EDL and their valuable time. Please find answers to your weaknesses below:
>
> > I can follow the rationale behind Eq 5 and it makes sense. I am only not sure that this is the simplest possible way the intended goal is achieved. Extending the related part of the paper (around Lines 239-256) will contribute to its readability. I do understand that Theorem 1 clearly demonstrates that the criterion behaves in the intended way. The question here is how close the proposed solution is the simplest possible one.
>
> A: We thank the reviewer for this comment and agree that further intuition around Eq. 5 would improve clarity. Our goal was to construct an inter-class conflict measure that is symmetric, bounded, and increases only when different classes receive both (i) comparable support and (ii) sufficiently strong evidence. Other simpler formulations we explored failed to respect this ordering in edge cases (for example, assigning higher conflict to weak, noisy evidence than to two strongly supported competing classes). Thus, Eq. 5 represents the smallest extension that captures both the relative balance between classes and the absolute strength of their evidence, while remaining analytically tractable for Theorem 1. We will expand the discussion around lines 239–256 to better convey this rationale.
>
> > Did the authors consider alternative criteria such as an adaptation of InfoNCE, Barlow Twins, or IMAX?
>
> A: Thank you for these interesting suggestions. Enhancing C-EDL with criteria like those mentioned above would definitely be a fruitful investigation that would enable us to assess the applicability and generalisability of our approach.  We will pursue this avenue as part of our future work.
>
> > Evidential learning is also being used in regression [1]. In that case, as we don't have a Dirichlet distribution, I understand that the suggested conflict adjustment criterion will not directly translate. Do the authors have any insight about which basic principles of the suggested contribution can be extended to the regression case?
> [1] Amini et al., Deep evidential regression, NeurIPS, 2020
>
> A: This is a very interesting suggestion, and we would like to investigate this as part of a collaborative effort in the future.

---

> > ### Comment · Reviewer_g4uc · 2025-11-26
> >
> > Thanks. My original opinion about the paper is quite positive and my questions were rather to conduct a scientific discussion. Although it doesn't affect my grade, I have to mention that the authors appear not to answer my questions and delegated all to future work. This doesn't quite fit the purpose of the open review system we are exercising. Having read the other reviews, I don't see any misunderstanding that may have led me to a too positive scores as the raised issues appear to stem from some missed appendix details and some others are addressed by additional experiments. Hence I keep my score.

---

### Official Review · Reviewer_jVfx · 2025-10-31

**Soundness:** 3
**Presentation:** 3
**Contribution:** 3
**Rating:** 4
**Confidence:** 4

**Summary:**

The authors propose Conflict-aware Evidential Deep Learning (C-EDL), a post-hoc method to improve the robustness of pre-trained EDL models against OOD and adversarial inputs. The core idea is to generate multiple, semantically equivalent views of an input using label-preserving transformations, pass them through the base EDL model to get a set of evidence vectors, and then quantify the conflict or disagreement among these evidence sets. This conflict score is then used to adjust (reduce) the aggregated evidence, thereby increasing the model's expressed uncertainty for contentious inputs.

**Strengths:**

*   The problem of improving the adversarial robustness of EDL models is significant, as the base method is known to be vulnerable despite its efficiency.
*   The proposed post-hoc approach is a practical strength, as it can be applied to any pre-trained EDL model without requiring costly retraining, making it easier to deploy.
*   The core idea of leveraging disagreement across multiple generated views to calibrate uncertainty is intuitive and simple.
*   The experimental results presented are extensive and demonstrate large performance gains.

**Weaknesses:**

My main concerns are regarding the related work, the scale of the experimental evaluation, and the analysis of the method's scalability.

In terms of related work, the authors seem to have missed recent work on improving EDL robustness. Pandey et al. (ICML 2023) also addresses the shortcomings of EDL for OOD detection by introducing a novel regularizer, RED. A comparison to this closely related baseline is missing and would be necessary to properly situate C-EDL's contribution within the current state-of-the-art.

While the experiments are comprehensive across many metrics, they are limited to small-scale vision datasets like MNIST, CIFAR-10, and their variants. The claims of robustness and the general applicability of the method would be much stronger if demonstrated on large-scale, more complex datasets such as ImageNet and ideally real world datasets.

On a related note, I would like to see a more thorough scalability analysis, investigating how runtime scales not just with the number of transformations, but also with model size and input dimensionality.

The formulation of the conflict score C, combining intra-class and inter-class measures, seems somewhat over-engineered. While the authors provide a theoretical analysis of its properties in the appendix, the paper would benefit from a clearer motivation for this specific formulation over other, perhaps simpler, measures of distributional disagreement (e.g., average Jensen-Shannon divergence). It is also unclear how a practitioner would select the set of "metamorphic transformations" for a new domain, which seems critical to the method's success but is not discussed in detail.

Finally, the paper's relationship with Test-Time Augmentation (TTA) could be made more explicit. The core idea of generating multiple transformed views of a test input is functionally equivalent to TTA. While the authors acknowledge TTA, they could better frame their contribution in this context. C-EDL can be seen as a novel aggregation strategy for TTA applied specifically to EDL models. Whereas standard TTA involves a simple averaging of output predictions C-EDL's primary novelty is in its aggregation of intermediate evidence vectors and the subsequent use of a "conflict score" to modulate the final uncertainty. The authors'  ablation study on "EDL++" (which effectively performs TTA on EDL evidence vectors without conflict adjustment) demonstrates that this conflict-aware step is critical for the method's success. Clarifying this aspect would help to more precisely specify the paper's novelty.

**Questions:**

See above

---

> ### Author Response · Authors · 2025-11-20
>
> We thank the reviewer for their time and valuable review. Please find answers to your weaknesses below:
>
> > In terms of related work, the authors seem to have missed recent work on improving EDL robustness. Pandey et al. (ICML 2023) also addresses the shortcomings of EDL for OOD detection by introducing a novel regularizer, RED. A comparison to this closely related baseline is missing and would be necessary to properly situate C-EDL's contribution within the current state-of-the-art.
>
> A: We thank the reviewer for pointing out RED (Pandey et al., ICML 2023). We agree it is a valuable extension of EDL, and a comparison would be good. RED is not designed as an OOD‐detection method: it regularises training by increasing evidence for the true class when vacuity is high, thereby mitigating zero-evidence collapse and improving the base classifier. In contrast, C-EDL is a post-hoc uncertainty calibration method applied at inference time, specifically targeting OOD and adversarial robustness.
>
> Following the reviewer’s suggestion, and for completeness, we have evaluated RED against EDL and C-EDL in the Table below. Results show that while RED improves ID accuracy, it provides weaker OOD and adversarial rejection compared to C-EDL (in the table below, lower is better, indicating that the percentage of C-EDL making a decision for OOD and Adversarial samples is lower than RED).
>
> |                    | EDL               | C-EDL (Meta)      | RED                |
> |--------------------|-------------------|--------------------|--------------------|
> | MNIST → FMNIST     |                   |                    |                    |
> | ID Acc             | $99.96 \pm 0.02$  | $99.96 \pm 0.01$   | $99.98 \pm 0.02$   |
> | ID Cov             | $96.61 \pm 0.57$  | $94.18 \pm 1.03$   | $92.17 \pm 1.77$   |
> | OOD Cov            | $2.52 \pm 0.68$   | $2.00 \pm 0.80$    | $6.30 \pm 9.44$    |
> | Adv Cov            | $52.21 \pm 9.49$  | $15.51 \pm 6.09$   | $19.05 \pm 9.03$   |
> | CIFAR10 → SVHN     |                   |                    |                    |
> | ID Acc             | $95.88 \pm 0.54$  | $97.98 \pm 0.42$   | $95.48 \pm 1.26$   |
> | ID Cov             | $67.34 \pm 3.15$  | $54.70 \pm 2.11$   | $60.24 \pm 5.35$   |
> | OOD Cov            | $10.91 \pm 2.23$  | $4.69 \pm 1.00$    | $11.68 \pm 3.13$   |
> | Adv Cov            | $20.00 \pm 6.89$  | $1.25 \pm 0.56$    | $4.39 \pm 2.96$    |
>
> > While the experiments are comprehensive across many metrics, they are limited to small-scale vision datasets like MNIST, CIFAR-10, and their variants. The claims of robustness and the general applicability of the method would be much stronger if demonstrated on large-scale, more complex datasets such as ImageNet and ideally real world datasets.
>
> A: In addition to MNIST and CIFAR-scale benchmarks, we already evaluate C-EDL on more complex datasets in a few-shot ResNet50 setting using Tiny-ImageNet and CUB (Appendix B.1, Table 5). These results exceed the dataset scale typically used in prior EDL and OOD-EDL work, enabling a fair comparison to existing baselines while demonstrating that C-EDL remains effective on higher-dimensional, real-world images. We will highlight this aspect in the main text of our paper.

---

> > ### Author Response · Authors · 2025-11-20
> >
> > > On a related note, I would like to see a more thorough scalability analysis, investigating how runtime scales not just with the number of transformations, but also with model size and input dimensionality.
> >
> > A: We thank the reviewer for this suggestion. Appendix B.4 (Table 11) already includes an ablation over the number of transformations $T$ , showing that runtime scales with $T$. To further address scalability, we have now added experiments varying both model size and input dimensionality, comparing a small LeNet (0.45M params) and a larger SeNet (9.79M params) on MNIST and CIFAR-10. As shown below, the runtime grows with $T$ and proportionally with model capacity and input resolution, matching the expected complexity of $T$ forward passes.
> >
> > |          | C-EDL (Meta) - LeNet (0.45M params) | -                | -                 | -                 | -                 | C-EDL (Meta) - SeNet (9.79M params) |         -         |          -         |          -         |          -         |
> > |----------|:-----------------------------------:|------------------|-------------------|-------------------|-------------------|:-----------------------------------:|:-----------------:|:------------------:|:------------------:|:------------------:|
> > |  Dataset |                 2.00                |       5.00       |       10.00       |       25.00       |       50.00       |                 2.00                |        5.00       |        10.00       |        25.00       |        50.00       |
> > |   MNIST  |           $2.53s \pm 0.14$          | $5.12s \pm 0.10$ |  $9.18s \pm 0.21$ | $24.32s \pm 0.24$ | $48.32s \pm 0.46$ | $35.20s \pm 0.58$                   | $71.79s \pm 1.12$ | $135.90s \pm 1.26$ | $324.24s \pm 2.33$ | $621.82s \pm 4.31$ |
> > | CIFAR-10 |           $3.40s \pm 0.42$          | $6.96s \pm 0.56$ | $13.62s \pm 0.70$ | $33.81s \pm 0.22$ | $58.32s \pm 0.54$ | $41.24s \pm 0.36$                   | $82.91s \pm 1.31$ | $151.09s \pm 1.59$ | $392.16s \pm 2.76$ | $687.05s \pm 6.17$ |
> >
> > |                      | C-EDL (Meta) - LeNet (0.45M params) | C-EDL (Meta) - SeNet (9.79M params) |
> > |:--------------------:|:-----------------------------------:|:-----------------------------------:|
> > |   ID Acc $\uparrow$  |          $99.96\% \pm 0.01$         |           $99.94 \pm 0.01$          |
> > |   ID Cov $\uparrow$  |          $94.18\% \pm 1.03$         |          $96.64\% \pm 0.54$         |
> > | OOD Cov $\downarrow$ |          $2.00\% \pm 0.80$          |           $1.86 \pm 0.48$           |
> > | Adv Cov $\downarrow$ |          $15.51\% \pm 6.09$         |          $5.50\% \pm 2.81$          |
> >
> > > The formulation of the conflict score C, combining intra-class and inter-class measures, seems somewhat over-engineered. While the authors provide a theoretical analysis of its properties in the appendix, the paper would benefit from a clearer motivation for this specific formulation over other, perhaps simpler, measures of distributional disagreement (e.g., average Jensen-Shannon divergence).
> >
> > A: We thank the reviewer for this observation. While simpler measures, such as the Jensen–Shannon divergence, can quantify global distributional differences, they overlook the finer-grained disagreement patterns that arise. Our formulation explicitly separates two complementary forms of conflict: intra-class variability, reflecting fluctuations in evidence magnitude across transformations, and inter-class contradiction, capturing competition among supported classes. The inclusion–exclusion combination (Eq. 6) provides a bounded and interpretable union of these effects [1], while the $\lambda$ term introduces controllable sensitivity between symmetric and asymmetric disagreement. This structure enables C-EDL to detect subtle representational inconsistencies that simpler divergences would miss. See also Figure 14, Apeendix B.1, that illustrates the difference between EDL and C-EDL (clarifying related comment by Reviewer odmc), which illustrates the usefulness of our conflict-aware measure.
> >
> > [1] Szpankowski, W., 2011. Average case analysis of algorithms on sequences. John Wiley & Sons.
> >
> > > It is also unclear how a practitioner would select the set of "metamorphic transformations" for a new domain, which seems critical to the method's success but is not discussed in detail.
> >
> > A: We discuss the choice of metamorphic transformations in Appendix B.4, where we additionally provide an ablation (Table 10) and practical recommendations for selecting transformations in new domains. In practice, we find that simple, label-preserving transformations are sufficient and even very weak or only single-choice transformations suffice. We will make this guidance more explicit in the main text to clarify how practitioners can adapt the transformation set to their own domain.

---

> > > ### Author Response · Authors · 2025-11-20
> > >
> > > > Finally, the paper's relationship with Test-Time Augmentation (TTA) could be made more explicit. The core idea of generating multiple transformed views of a test input is functionally equivalent to TTA. While the authors acknowledge TTA, they could better frame their contribution in this context. C-EDL can be seen as a novel aggregation strategy for TTA applied specifically to EDL models. Whereas standard TTA involves a simple averaging of output predictions C-EDL's primary novelty is in its aggregation of intermediate evidence vectors and the subsequent use of a "conflict score" to modulate the final uncertainty. The authors' ablation study on "EDL++" (which effectively performs TTA on EDL evidence vectors without conflict adjustment) demonstrates that this conflict-aware step is critical for the method's success. Clarifying this aspect would help to more precisely specify the paper's novelty.
> > >
> > > A: EDL++ in our ablation (Tables 2 and 3) does indeed correspond to a TTA-style aggregation at the evidence level, and our results show that this alone provides only modest improvements. The core novelty of C-EDL lies in the conflict-aware adjustment of evidence: instead of averaging views uniformly, we quantify their disagreement and modulate the final uncertainty accordingly. This conflict step is what enables C-EDL to outperform EDL++ (that uses only TTA-style augmentation) across all robustness metrics (Fig. 9). We agree with the reviewer’s observation and will make this connection more explicit.

---

### Official Review · Reviewer_z85G · 2025-11-03

**Soundness:** 2
**Presentation:** 2
**Contribution:** 2
**Rating:** 2
**Confidence:** 4

**Summary:**

The paper introduces a post-hoc calibration method for evidential deep learning (EDL) by adjusting the predicted total evidence based on output disagreement across augmented input variants. Specifically, the method estimates prediction conflicts between metamorphically transformed inputs and scales the evidence accordingly to improve robustness against adversarial and out-of-distribution (OOD) data. The authors claim that the proposed approach enhances uncertainty reliability while maintaining high in-distribution accuracy and without requiring model retraining.

**Strengths:**

1. The proposed method is post-hoc and architecture-agnostic. It can be directly applied to any trained EDL model without modifying the network structure or retraining, making it highly practical and easy to integrate.
2. The approach is computationally efficient compared to full retraining or Bayesian ensemble methods, as it only involves a few forward passes with simple input transformations.

**Weaknesses:**

1. The paper lacks sufficient implementation details for reproducibility. For example, the specific metamorphic transformations applied to each dataset, the exact uncertainty metric used in Table 1, and the threshold definition for coverage reporting are not clearly specified. The choice and tuning process for hyperparameters (e.g., conflict weighting and decay factors) should also be explained.
2. The motivation requires further clarification. EDL naturally distinguishes between aleatoric and epistemic uncertainty in a single forward pass, yet the proposed post-hoc calibration does not discuss how this property is affected. Input augmentations may alter aleatoric uncertainty due to increased noise, so the rationale behind interpreting disagreement among augmented predictions as a reliable proxy for epistemic uncertainty should be explicitly analyzed.
3. Regarding efficiency, the method requires multiple forward passes during inference. Thus, comparisons with other multi-pass methods such as Monte Carlo Dropout or ensemble-based EDL variants would provide a more meaningful evaluation of trade-offs.
4. For OOD detection, reporting AUROC and AUPR metrics in addition to coverage would provide a clearer and more comprehensive picture of the model’s uncertainty quality and detection capability.

**Questions:**

See Weaknesses.

---

> ### Author Response · Authors · 2025-11-20
>
> We thank the reviewer for their time and valuable review. Please find answers to your weaknesses below:
>
> > The paper lacks sufficient implementation details for reproducibility. For example, the specific metamorphic transformations applied to each dataset, the exact uncertainty metric used in Table 1, and the threshold definition for coverage reporting are not clearly specified. The choice and tuning process for hyperparameters (e.g., conflict weighting and decay factors) should also be explained.
>
> A: The implementation details are fully specified in the appendix. The metamorphic transformations across all datasets are listed in Appendix B.4. The uncertainty metric and threshold used for coverage in Table 1 follow the standard EDL protocol and are described in Appendix C.6. Hyperparameter choices, including conflict weighting and decay factors, are also given in Appendix C with ablation results in Appendix B.4. We will cross-reference these sections in the updated version of our paper to improve clarity and reproducibility.
>
> > The motivation requires further clarification. EDL naturally distinguishes between aleatoric and epistemic uncertainty in a single forward pass, yet the proposed post-hoc calibration does not discuss how this property is affected. Input augmentations may alter aleatoric uncertainty due to increased noise, so the rationale behind interpreting disagreement among augmented predictions as a reliable proxy for epistemic uncertainty should be explicitly analysed.
>
> A: Our post-hoc calibration does not change how EDL models uncertainty. It evaluates several label-preserving variants of the same input, including small and imperceptible geometric and noise-based augmentations, to assess how sensitive the model’s evidence is. Disagreement among these predictions mainly reflects epistemic uncertainty, because a confident model should remain stable under small, label-preserving perturbations. While certain augmentations, such as salt-and-pepper noise, may slightly increase aleatoric variability, this effect is limited by keeping the noise level low (see Table 10), and the overall disagreement still primarily captures the model’s epistemic uncertainty.
>
> > Regarding efficiency, the method requires multiple forward passes during inference. Thus, comparisons with other multi-pass methods such as Monte Carlo Dropout or ensemble-based EDL variants would provide a more meaningful evaluation of trade-offs.
>
> A: We thank the reviewer for raising this point, and we agree that comparing against MC Dropout would be valuable. It should be noted that our evaluation already includes multi-pass baselines: S-EDL (Table 1) is itself a multi-pass method, and Table 4 further includes diverse UQ approaches, giving a total of 11 comparative methods (excluding ablations such as EDL++). We have also added a comparison from Reviewer jVfx, which can be seen in their discussion.
>
> We have also added a comparison against MC-Dropout in Table 4, Appendix B.1 of our revised paper. As shown below, MC Dropout improves upon EDL in adversarial coverage (Adv Cov), showing some strengths, but performs substantially worse than C-EDL in both OOD and adversarial coverage, despite requiring even more forward passes.
>
> |            | EDL              | C-EDL (Meta T=5)    | MC Dropout (T=100) |
> |------------|------------------|----------------------|----------------------|
> | ID Acc     | $99.96 \pm 0.02$ | $99.96 \pm 0.01$     | $99.94 \pm 0.03$     |
> | ID Cov     | $96.61 \pm 0.57$ | $94.18 \pm 1.03$     | $88.46 \pm 2.61$     |
> | OOD Cov    | $2.52 \pm 0.68$  | $2.00 \pm 0.80$      | $12.82 \pm 3.06$     |
> | Adv Cov    | $52.21 \pm 9.49$ | $15.51 \pm 6.09$     | $27.79 \pm 4.70$     |
>
> > For OOD detection, reporting AUROC and AUPR metrics in addition to coverage would provide a clearer and more comprehensive picture of the model’s uncertainty quality and detection capability.
>
> A: Appendix C explains that our coverage metric is a thresholded form of AUROC that aligns more closely with real-world abstention settings, and therefore is used as the primary metric. The submitted version of our manuscripts already includes AUROC snapshots in the main paper (Figure 4) and provide additional AUROC curves in Appendix B.1, providing the broader picture requested. To improve the clarity of our paper, we will make these references more explicit and clarify the relationship between coverage and AUROC/AUPR for transparency.

---

### Official Review · Reviewer_odmc · 2025-11-03

**Soundness:** 3
**Presentation:** 3
**Contribution:** 3
**Rating:** 6
**Confidence:** 3

**Summary:**

The paper introduces C-EDL, a post-hoc uncertainty quantification method that enhances the robustness of Evidential Deep Learning (EDL) models against out-of-distribution (OOD) and adversarial inputs. C-EDL generates multiple metamorphic transformations of each input, quantifies model disagreement (conflict) via intra- and inter-class evidence variability, and adjusts the Dirichlet parameters accordingly. Extensive experiments show that C-EDL significantly reduces OOD and adversarial coverage while maintaining in-distribution (ID) accuracy.

**Strengths:**

1. C-EDL is simple and clear, which achieves promising results.
2. The conflict measure  $C$ is formally bounded, providing a principled basis for evidence adjustment.
3. This work provides a comprehensive evaluation.

**Weaknesses:**

1. Standard EDL was originally used for the most basic classification tasks. Can the proposed C-EDL be directly applied to improve the level of uncertainty quantification and performance?
2. The citation format needs to be modified to improve readability; for example, using \citep.
3. C-EDL essentially utilizes multiple views and improves the accuracy of uncertainty quantification by quantifying the conflict of these views. However, there is a lack of discussion regarding some related work, e.g., [1].
4. In Table 8, the experimental results for the optimal parameters are recommended to be bolded.
5. I found that the parameter $\\epsilon$ in Equation 4 seems to be quite important in the subsequent proof; however, the intuition and motivation behind it lack a clear discussion.
6. I suggest providing some intuitive examples to demonstrate its advantages over the baseline in quantifying uncertainty. For example, using the simplest 3-dimensional simplex.

> Ref:
> [1] Reliable Conflictive Multi-View Learning, AAAI'24.

**Questions:**

See Weaknesses.

---

> ### Author Response · Authors · 2025-11-20
>
> We thank the reviewer for their positive review of C-EDL and their valuable time. Please find answers to your weaknesses below:
>
> > Standard EDL was originally used for the most basic classification tasks. Can the proposed C-EDL be directly applied to improve the level of uncertainty quantification and performance?
>
> A: Indeed, this is exactly the ultimate goal of C-EDL. Our approach is a drop-in post-hoc upgrade to standard EDL, and our results show that it substantially improves uncertainty estimates and robustness without retraining.
>
> > The citation format needs to be modified to improve readability; for example, using \citep.
>
> A: Thank you for this useful suggestion. We have updated the citation format in the updated version of our paper.
>
> > C-EDL essentially utilizes multiple views and improves the accuracy of uncertainty quantification by quantifying the conflict of these views. However, there is a lack of discussion regarding some related work, e.g., [1].
> Ref: [1] Reliable Conflictive Multi-View Learning, AAAI'24.
>
> A: This work is orthogonal. ECML assumes multiple input views (for example, an image and its caption) and trains a separate evidential model for each view. It then measures conflict when these view-specific models make confident but different predictions, and uses this signal during training. C-EDL instead operates within a single modality and uses label-preserving transformations to assess representational disagreement. Thank you for pointing us towards this paper; for completeness, we will add this discussion to our related work section.
>
> > In Table 8, the experimental results for the optimal parameters are recommended to be bolded.
>
> A: Thank you for this suggestion. We have implemented your suggestion in the updated version of our paper.
>
> > I found that the parameter epsilon in Equation 4 seems to be quite important in the subsequent proof; however, the intuition and motivation behind it lack a clear discussion.
>
> A: We thank the reviewer for bringing this to our attention. The small constant $\epsilon$ in Eq. 4 is not only a numerical stabiliser but also ensures the denominator in $C_{intra}$ is strictly positive, which is then required in Appendix A to guarantee boundedness and continuity of $C_{intra}$. We will update this point in the paper to clarify the effect of epsilon.
>
> > I suggest providing some intuitive examples to demonstrate its advantages over the baseline in quantifying uncertainty. For example, using the simplest 3-dimensional simplex.
>
> A: We thank the reviewer for this suggestion. We have now added a 3-class simplex visualisation (see Figure 14, Appendix B.1 in the updated revision of our paper) for the CIFAR-10 dataset comparing EDL and C-EDL. Whilst EDL produces strong uncertainty regions across the simplex, C-EDL further boosts these uncertain regions and in the centre outputs uncertainty close to 1 (full uncertainty). We also observe that C-EDL has higher uncertainty when there is equal disagreement between classes (directly our $C_{inter}$ conflict). Further, C-EDL preserves the low uncertainty regions in the corners of the simplex. Accordingly, C-EDL preserves low uncertainty in high-evidence and low-conflict regions just like EDL, but increases uncertainty in uncertain and high-conflict regions, unlike EDL. This insight provides an intuitive illustration of how conflict-aware adjustment used by our C-EDL approach yields more credible uncertainty estimates than EDL.

---

### Official Review · Reviewer_y7KH · 2025-11-14

**Soundness:** 3
**Presentation:** 3
**Contribution:** 3
**Rating:** 6
**Confidence:** 4

**Summary:**

The paper, “Robust Adversarial Quantification via Conflict-Aware Evidential Deep Learning (C-EDL)”, introduces a new post-hoc uncertainty quantification (UQ) method that enhances the robustness of Evidential Deep Learning (EDL) models to out-of-distribution (OOD) and adversarial inputs.

Motivation
Standard EDL provides efficient uncertainty estimation by modeling class probabilities as Dirichlet distributions in a single forward pass. However, it remains vulnerable to adversarial perturbations, often producing overconfident predictions on corrupted or unfamiliar inputs. Prior extensions (e.g., S-EDL, R-EDL, H-EDL, DA-EDL) improve OOD detection but fail to adequately address adversarial overconfidence, and most require retraining.

Proposed Method: Conflict-aware EDL (C-EDL)
C-EDL is a lightweight, post-hoc framework that can be applied to pretrained EDL models. It operates by:
	1.	Generating multiple label-preserving metamorphic transformations of each input to create diverse evidence sets.
	2.	Quantifying intra-class variability (how much evidence for a class fluctuates) and inter-class contradiction (how much competing classes are equally supported).
	3.	Combining these into a conflict score (C) that measures representational disagreement across transformations.
	4.	Using this score to downscale the Dirichlet evidence via an exponential decay rule, thereby increasing uncertainty when conflict is high and maintaining confidence when conflict is low.

Mathematically, the adjusted evidence is expressed as
$$\tilde{\alpha}_k = \bar{\alpha}_k \times \exp(-\delta C)$$
where $\bar{\alpha}_k$ is the aggregated evidence and $\delta$ is a sensitivity parameter.

The authors also provide a theoretical guarantee that the conflict measure C is bounded, monotonic, and consistent with Dempster–Shafer theory principles.

Results
	•	Evaluations across multiple ID/OOD dataset pairs (MNIST→FashionMNIST, CIFAR10→SVHN, Oxford Flowers→DeepWeeds, etc.) and attack types (L2PGD, FGSM, Salt-and-Pepper) show that C-EDL substantially improves robustness:
	•	OOD coverage reduced by up to ≈55%
	•	Adversarial coverage reduced by up to ≈90%
	•	In-distribution accuracy maintained (~99%)
	•	C-EDL variants (Meta vs MC) demonstrate that metamorphic transformations outperform Monte Carlo sampling for generating meaningful uncertainty signals.
	•	The method is threshold-agnostic and retains negligible computational overhead.

Contributions
	1.	A conflict-aware, post-hoc calibration method for EDL uncertainty estimation.
	2.	Theoretical analysis of the conflict metric’s properties.
	3.	Comprehensive experimental validation across datasets, attack types, and decision thresholds showing consistent superiority over prior EDL methods.

In summary:
C-EDL presents a principled and efficient approach to making evidential deep learning models more robust to adversarial and OOD conditions, achieving strong empirical results while maintaining efficiency and avoiding retraining.

**Strengths:**

Originality:
The paper introduces a conflict-aware uncertainty adjustment mechanism for evidential deep learning (EDL), which is a meaningful extension to prior post-hoc calibration and OOD robustness methods. The idea of leveraging metamorphic transformations to measure evidence disagreement is both conceptually intuitive and novel within the EDL literature.
	•	Quality:
The experimental evaluation is comprehensive, spanning multiple datasets (MNIST, CIFAR10, Flowers102, etc.) and adversarial settings (FGSM, PGD, Salt-and-Pepper). The results consistently support the central claim that C-EDL improves robustness without retraining, confirming the technical soundness of the method.
	•	Clarity:
The paper is clearly structured with a good balance between formalism and intuition. The figures effectively illustrate the mechanism and the results, particularly the conflict-based uncertainty scaling and comparative coverage plots.
	•	Significance:
C-EDL contributes to an active and relevant research direction — trustworthy and uncertainty-aware deep learning. By improving EDL robustness in a post-hoc and computationally light manner, the method holds potential practical value for real-world AI reliability and safety-critical applications.

Overall Strength:
A well-motivated, well-executed, and clearly presented contribution that meaningfully extends evidential learning toward robustness and reliability.

**Weaknesses:**

Limited Theoretical Depth:
While the paper provides an intuitive justification and basic boundedness proof for the conflict metric, the mathematical grounding of how conflict relates to epistemic and aleatoric uncertainty decomposition is underdeveloped. A stronger theoretical connection to Dempster–Shafer theory or Bayesian evidence accumulation would strengthen the claim of principled robustness.
	•	Overlap with Prior Work:
The contribution, though practical, feels incremental relative to recent variants such as R-EDL (Robust EDL) and H-EDL (Hierarchical EDL). The manuscript could better differentiate its novelty beyond being a post-hoc fusion of metamorphic testing and EDL evidence scaling.
	•	Evaluation Breadth:
The experiments are extensive but lacking in diversity of model architectures. Most results are presented on simple CNNs and small datasets (MNIST, CIFAR10). Including modern architectures (ResNet, ViT) would improve generalization credibility and relevance for the ICLR audience.
	•	Ablation Clarity:
While several variants (C-EDL-MC, C-EDL-Meta) are tested, ablation on key hyperparameters (e.g., transformation count, conflict scaling δ) is minimal. It’s unclear how sensitive the method is to these design choices, limiting reproducibility and interpretability.
	•	Qualitative Analysis Missing:
The paper lacks visual or interpretive examples of how conflict behaves across inputs — e.g., how uncertainty distributions change for adversarial vs. OOD samples. Such qualitative insight could strengthen readers’ understanding of what the conflict term captures.
	•	Reproducibility Gaps:
Implementation details are briefly described, but no code release or reproducibility checklist is mentioned. For a post-hoc method claiming simplicity, an open implementation would be essential to validate the reported improvements.

Overall Weakness Summary:
The paper is technically solid but conceptually incremental. Strengthening theoretical justification, expanding experiments to more challenging datasets and architectures, and including detailed ablations would substantially increase its impact and readiness for top-tier publication.

**Questions:**

1.	Conflict Measure Definition and Theoretical Rationale
	•	The conflict score C is central to the method. Could the authors clarify whether it formally corresponds to a measure of epistemic uncertainty (i.e., model disagreement) or a combination of epistemic and aleatoric components?
	•	How does it relate to established evidential theory constructs such as conflict mass in Dempster–Shafer theory or belief/plausibility intervals? A clearer theoretical bridge would strengthen the conceptual contribution.
2.	Choice of Transformations
	•	The metamorphic transformations used (rotation, contrast, etc.) appear heuristic.
	•	How sensitive is the method to the type and strength of these transformations?
	•	Could the authors provide an ablation showing how different transformation sets affect performance on adversarial and OOD benchmarks?
3.	Computational Cost and Efficiency
	•	The paper states that C-EDL is “lightweight” and post-hoc, but no explicit computational cost analysis is provided.
	•	How does inference time or memory footprint scale with the number of transformations?
	•	Could the authors provide a table comparing wall-clock latency versus baseline EDL and Monte Carlo dropout methods?
4.	Comparison with Recent Robust EDL Variants
	•	How does C-EDL compare to S-EDL (Shannon-EDL) and R-EDL (Regularized-EDL) on the same benchmarks?
	•	These methods also modify evidence to reduce overconfidence. Are there specific cases where C-EDL performs worse or complements them?
5.	Impact on Calibration Metrics
	•	While OOD and adversarial coverage are reported, calibration metrics such as Expected Calibration Error (ECE) or Brier score are not shown.
	•	Could the authors report these to confirm that C-EDL improves not just robustness but also calibration fidelity?
6.	Applicability Beyond Classification
	•	The approach is tailored for classification tasks. Could it extend to regression or structured prediction problems?
	•	How would the conflict term be redefined in a continuous-output evidential setting?
7.	Sensitivity to δ (Scaling Parameter)
	•	The exponential scaling parameter \delta plays a crucial role in modulating evidence.
	•	How was it chosen, and how sensitive is model performance to this hyperparameter?
	•	Would a learnable δ (e.g., via validation loss optimization) yield more consistent improvements?
8.	Interpretability of Conflict Scores
	•	Have the authors visualized conflict maps or examined which regions of an image contribute most to high conflict?
	•	This could enhance understanding of whether C-EDL is detecting semantic uncertainty or simply pixel-level perturbations.
9.	Robustness Under Distributional Shift
	•	Beyond adversarial examples, have the authors tested on natural distribution shifts (e.g., CIFAR-10-C, ImageNet-C)?
	•	This would strengthen claims of general robustness rather than adversarial robustness alone.
10.	Reproducibility and Implementation Availability
	•	Is there a plan to release code or pretrained models?
	•	Given that the paper emphasizes post-hoc simplicity, open-source availability would allow the community to verify these promising results quickly.

Summary:
The main clarifications concern the theoretical interpretation of conflict, robustness to transformation choices, and quantitative trade-offs between accuracy, calibration, and inference cost. Addressing these would substantially improve the paper’s completeness and strengthen its position for acceptance.

**Details Of Ethics Concerns:**

This paper focuses entirely on a technical contribution — improving robustness and uncertainty quantification in Evidential Deep Learning (EDL).
It does not involve human subjects, personal data, sensitive attributes, or potentially harmful applications. All datasets used (MNIST, CIFAR10, SVHN, etc.) are public, well-established benchmarks with no privacy or ethical risks.

---

> ### Author Response · Authors · 2025-11-20
>
> We thank the reviewer for their time and positive review of C-EDL. Please find answers to your weaknesses below:
>
> > Limited Theoretical Depth: While the paper provides an intuitive justification and basic boundedness proof for the conflict metric, the mathematical grounding of how conflict relates to epistemic and aleatoric uncertainty decomposition is underdeveloped. A stronger theoretical connection to Dempster–Shafer theory or Bayesian evidence accumulation would strengthen the claim of principled robustness.
>
> A: We appreciate this suggestion. C-EDL is a post-hoc calibration method and does not alter the epistemic/aleatoric decomposition already encoded by the underlying EDL model. Our contribution is to probe this decomposition by evaluating the stability of the evidence under multiple label-preserving transformations. Disagreement across these transformations primarily reflects epistemic uncertainty, since a well-trained model should generalise and produce consistent evidence for small perturbations of the same input. Although certain transformations (e.g., light noise) introduce a minor aleatoric component, we contain their intensity (see Table 10), and the resulting conflict still mainly reflects the model’s sensitivity to input-level perturbations. We will expand the theoretical discussion to clarify this connection and to better relate our conflict measure to the principles of evidence accumulation used in evidential frameworks.
>
> > Overlap with Prior Work: The contribution, though practical, feels incremental relative to recent variants such as R-EDL (Robust EDL) and H-EDL (Hierarchical EDL). The manuscript could better differentiate its novelty beyond being a post-hoc fusion of metamorphic testing and EDL evidence scaling.
>
> A: We thank the reviewer for this comment. Our goal was precisely to position C-EDL against recent EDL variants, and we therefore already compare against several strong baselines (11 methods in total, plus RED added in this rebuttal). We were unable to locate additional methods explicitly named “Robust EDL” or “Hierarchical EDL”; if the reviewer could provide citations for these EDL variants, we would be happy to cite, try to compare, and discuss them in the camera-ready version of our paper.
>
> Conceptually, C-EDL differs from these prior variants in that it is a purely post-hoc method applied to any pretrained EDL model. Instead of changing the training loss or network architecture, C-EDL uses label-preserving metamorphic transformations together with a conflict-aware evidence scaling mechanism, explicitly designed to improve OOD and adversarial detection while preserving ID accuracy.
>
> > Evaluation Breadth: The experiments are extensive but lacking in diversity of model architectures. Most results are presented on simple CNNs and small datasets (MNIST, CIFAR10). Including modern architectures (ResNet, ViT) would improve generalization credibility and relevance for the ICLR audience.
>
> A: We appreciate this suggestion. In addition to the CNNs on MNIST/CIFAR10, we already evaluate C-EDL on a more modern and larger-scale setup using a ResNet50 backbone in a Tiny-ImageNet / CUB few-shot setting (Appendix B.1, Table 5). We also performed experiments using SeNet (see results and inference times below)
>
> This demonstrates our approach generalises beyond small CNNs to deeper architectures and higher-resolution data.
>
> |          | C-EDL (Meta) - LeNet (0.45M params) | -                | -                 | -                 | -                 | C-EDL (Meta) - SeNet (9.79M params) |         -         |          -         |          -         |          -         |
> |----------|:-----------------------------------:|------------------|-------------------|-------------------|-------------------|:-----------------------------------:|:-----------------:|:------------------:|:------------------:|:------------------:|
> |  Dataset |                 2.00                |       5.00       |       10.00       |       25.00       |       50.00       |                 2.00                |        5.00       |        10.00       |        25.00       |        50.00       |
> |   MNIST  |           $2.53s \pm 0.14$          | $5.12s \pm 0.10$ |  $9.18s \pm 0.21$ | $24.32s \pm 0.24$ | $48.32s \pm 0.46$ | $35.20s \pm 0.58$                   | $71.79s \pm 1.12$ | $135.90s \pm 1.26$ | $324.24s \pm 2.33$ | $621.82s \pm 4.31$ |
> | CIFAR-10 |           $3.40s \pm 0.42$          | $6.96s \pm 0.56$ | $13.62s \pm 0.70$ | $33.81s \pm 0.22$ | $58.32s \pm 0.54$ | $41.24s \pm 0.36$                   | $82.91s \pm 1.31$ | $151.09s \pm 1.59$ | $392.16s \pm 2.76$ | $687.05s \pm 6.17$ |
>
> |  | C-EDL (Meta) - LeNet (0.45M params) | C-EDL (Meta) - SeNet (9.79M params) |
> |:---:|:---:|:---:|
> | ID Acc $\uparrow$ | $99.96 \pm 0.01$ | $99.94 \pm 0.01$ |
> | ID Cov $\uparrow$ | $94.18 \pm 1.03$ | $96.64 \pm 0.54$ |
> | OOD Cov $\downarrow$ | $2.00 \pm 0.80$ | $1.86 \pm 0.48$ |
> | Adv Cov $\downarrow$ | $15.51 \pm 6.09$ | $5.50 \pm 2.81$ |

---

> > ### Author Response · Authors · 2025-11-20
> >
> > > Ablation Clarity: While several variants (C-EDL-MC, C-EDL-Meta) are tested, ablation on key hyperparameters (e.g., transformation count, conflict scaling δ) is minimal. It’s unclear how sensitive the method is to these design choices, limiting reproducibility and interpretability.
> >
> > A: We provide a detailed hyperparameter ablation in Appendix B.4 (Table 8), where we vary all key parameters (β,λ,δ,T) and report their impact on ID accuracy, ID/OOD/adversarial coverage, and inference time. These results show that C-EDL is relatively robust to a wide range of settings (e.g., stable performance for δ ∈ [0.5,1.5] and T ∈ [10,25]), with larger T mainly trading off runtime for marginal robustness gains. We also include further discussion alongside this table, including recommendations for the hyperparameter selections.
> >
> > > Qualitative Analysis Missing: The paper lacks visual or interpretive examples of how conflict behaves across inputs — e.g., how uncertainty distributions change for adversarial vs. OOD samples. Such qualitative insight could strengthen readers’ understanding of what the conflict term captures.
> >
> > A: Qualitative analysis is provided in Figure 11 in our Appendix. It visualises exclusion-based attention overlays from the C-EDL model for OOD (FashionMNIST, left) and ID (MNIST, right) inputs. The heatmaps highlight spatial regions where attention distributions differ across metamorphic transformations. These overlays qualitatively highlight regions of disagreement that could potentially lead to an increase in the conflict measure $C$.
> >
> > > Reproducibility Gaps: Implementation details are briefly described, but no code release or reproducibility checklist is mentioned. For a post-hoc method claiming simplicity, an open implementation would be essential to validate the reported improvements.
> >
> > A: We thank the reviewer for this suggestion. We already provide an anonymised open-source repository link in the main paper (line 290), containing scripts to reproduce all experiments. Upon acceptance, we will de-anonymise the repository to ensure it is fully reproducible.
> >
> > > Overall Weakness Summary: The paper is technically solid but conceptually incremental. Strengthening theoretical justification, expanding experiments to more challenging datasets and architectures, and including detailed ablations would substantially increase its impact and readiness for top-tier publication.
> >
> > A: While C-EDL is intentionally lightweight as a post-hoc method, we note that our experiments already span a broad range of datasets and architectures: MNIST, Fashion-MNIST, KMNIST, EMNIST, CIFAR-10, CIFAR-100, SVHN, Oxford Flowers, DeepWeeds, Tiny-ImageNet, and CUB, using models from a large LeNet to ResNet-50 and a 9.8M-parameter SeNet. This range goes beyond the typical scope of prior EDL and evidential-robustness work. In addition, Appendix B contains detailed ablations of all key hyperparameters (including conflict weighting, transformation sets, decay factors, and number of views), as well as sensitivity analyses and runtime scaling experiments. If the reviewer has specific datasets, architectures, or ablation types in mind that would strengthen the work further, we would be grateful for their suggestions.
> >
> > > Conflict Measure Definition and Theoretical Rationale • The conflict score C is central to the method. Could the authors clarify whether it formally corresponds to a measure of epistemic uncertainty (i.e., model disagreement) or a combination of epistemic and aleatoric components? • How does it relate to established evidential theory constructs such as conflict mass in Dempster–Shafer theory or belief/plausibility intervals? A clearer theoretical bridge would strengthen the conceptual contribution.
> >
> > A: By construction, $C$ captures model disagreement across label-preserving transformations of the same input, and thus primarily reflects epistemic uncertainty: a well-specified model should produce stable evidence for small, task-preserving perturbations, so large intra-/inter-class variation across these views indicates a lack of knowledge rather than input noise. While some transformations (e.g., mild noise) can introduce a small aleatoric component, we keep their magnitude low and fixed, so their effect is largely uniform across views and does not dominate the conflict signal.
> >
> > Our use of the term “conflict” is inspired by Dempster–Shafer theory, in the sense that both our $C$ and the DST conflict mass quantify inconsistency between multiple sources of evidence. However, we operate directly on Dirichlet evidence vectors produced by a single EDL model under metamorphic views, rather than on separate belief mass functions combined via Dempster’s rule. The underlying mapping from evidence to belief/plausibility intervals in EDL remains unchanged; $C$ only modulates the overall Dirichlet strength to downweight highly conflicting evidence, and thus boost uncertainty by construction.

---

> > > ### Author Response · Authors · 2025-11-20
> > >
> > > > Choice of Transformations • The metamorphic transformations used (rotation, contrast, etc.) appear heuristic. • How sensitive is the method to the type and strength of these transformations? • Could the authors provide an ablation showing how different transformation sets affect performance on adversarial and OOD benchmarks?
> > >
> > > A: The choice of metamorphic transformations is discussed in Appendix B.4, where we also provide an ablation over type and strength (Table 10) together with practical recommendations. In practice, we find that C-EDL is not highly sensitive to these choices: simple label-preserving rotations, shifts and noise levels all yield similar robustness, and even very mild or single-type transformations already give strong gains over EDL. Only overly aggressive transformations that start to violate label preservation noticeably degrade ID coverage.
> > >
> > > > Computational Cost and Efficiency • The paper states that C-EDL is “lightweight” and post-hoc, but no explicit computational cost analysis is provided. • How does inference time or memory footprint scale with the number of transformations? • Could the authors provide a table comparing wall-clock latency versus baseline EDL and Monte Carlo dropout methods?
> > >
> > > A: We provide an explicit computational cost analysis in Appendix B.4 (Table 8), where we report wall-clock inference time for C-EDL compared to the number of transformations T. These results show that C-EDL’s latency scales with T as expected, and that even a small number of transformations (T=2) C-EDL can still effectively reduce the coverage on OOD and adversarial examples. We also discuss the runtime of base EDL, 1.15s ± 0.10, proving only a marginal increase in run-time when you calculate per-sample rather than the whole test dataset.
> > >
> > > > Comparison with Recent Robust EDL Variants • How does C-EDL compare to S-EDL (Shannon-EDL) and R-EDL (Regularized-EDL) on the same benchmarks? • These methods also modify evidence to reduce overconfidence. Are there specific cases where C-EDL performs worse or complements them?
> > >
> > > A: We are unable to locate the additional methods explicitly named “Shannon-EDL” or “Regularized-EDL”; if the reviewer could provide citations for these, we would be happy to cite, try to compare, and discuss them in a camera-ready version. We note that we already compare C-EDL against several strong baselines (11 methods in total, plus RED which was added in this rebuttal) demonstrating the effectiveness of C-EDL against state-of-the-art EDL-based approaches.
> > >
> > > > Impact on Calibration Metrics • While OOD and adversarial coverage are reported, calibration metrics such as Expected Calibration Error (ECE) or Brier score are not shown. • Could the authors report these to confirm that C-EDL improves not just robustness but also calibration fidelity?
> > >
> > > A: We thank the reviewer for this suggestion. In the current version, we focus on coverage-based metrics because they directly reflect abstention behaviour in safety-critical settings and are standard in recent EDL/OOD work. That said, C-EDL’s evidence decay is explicitly designed to down-weight overconfident predictions under conflict, so we expect it to also impact calibration. We will, therefore, look into further calibration metrics as future work.
> > >
> > > > Applicability Beyond Classification • The approach is tailored for classification tasks. Could it extend to regression or structured prediction problems? • How would the conflict term be redefined in a continuous-output evidential setting?
> > >
> > > A: We view this as promising future work and would be very interested in exploring it, potentially as part of a collaborative effort.
> > >
> > > > Sensitivity to δ (Scaling Parameter) • The exponential scaling parameter \delta plays a crucial role in modulating evidence. • How was it chosen, and how sensitive is model performance to this hyperparameter? • Would a learnable δ (e.g., via validation loss optimization) yield more consistent improvements?
> > >
> > > A: We thank the reviewer for highlighting the role of the scaling parameter δ. In all main experiments we fix δ=1.0 (Section C.6), chosen as a simple default after a small sweep guided by the ablation in Appendix B.4 (Table 8). This ablation varies δ ∈ [0.25,2.0] and shows that C-EDL is not highly sensitive to this choice: performance consistently improves over base EDL across this range, with larger δ acting more strictly and thus trading a small reduction in ID coverage for stronger OOD and adversarial abstention. While a learnable or automatically tuned δ could further reduce user effort, we do not expect it to fundamentally change the results. This choice is not a limitation specific to C-EDL; all compared methods rely on tunable hyperparameters, and δ plays an analogous role for C-EDL.

---

> > > > ### Author Response · Authors · 2025-11-20
> > > >
> > > > > Interpretability of Conflict Scores • Have the authors visualised conflict maps or examined which regions of an image contribute most to high conflict? • This could enhance understanding of whether C-EDL is detecting semantic uncertainty or simply pixel-level perturbations.
> > > >
> > > > A: Qualitative analysis is provided in Figure 11 in our Appendix B.1. It visualises exclusion-based attention overlays from the C-EDL model for OOD (FashionMNIST, left) and ID (MNIST, right) inputs. The heatmaps highlight spatial regions where attention distributions differ across metamorphic transformations. These overlays qualitatively highlight regions of disagreement that could potentially lead to an increase in the conflict measure $C$.
> > > >
> > > > > Robustness Under Distributional Shift • Beyond adversarial examples, have the authors tested on natural distribution shifts (e.g., CIFAR-10-C, ImageNet-C)? • This would strengthen claims of general robustness rather than adversarial robustness alone.
> > > >
> > > > A: We agree that natural distribution-shift benchmarks such as CIFAR-10-C or ImageNet-C would be valuable additions. While we do not include corruption benchmarks in this version, our evaluation already spans a wide range of natural domain shifts across MNIST, FashionMNIST, KMNIST, EMNIST, CIFAR-10, CIFAR-100, SVHN, Oxford Flowers, DeepWeeds, Tiny-ImageNet, and CUB. These dataset changes induce substantial appearance and distributional variation beyond adversarial perturbations. We will look to compare against these additional datasets as part of our future work.
> > > >
> > > > > Reproducibility and Implementation Availability • Is there a plan to release code or pretrained models? • Given that the paper emphasizes post-hoc simplicity, open-source availability would allow the community to verify these promising results quickly.
> > > >
> > > > A: We thank the reviewer for emphasising this point. We already provide an anonymised open-source repository link in the main paper (line 290 - footnote on page 6), containing scripts to reproduce all experiments. Upon acceptance, we will de-anonymise the repository to ensure it is fully reproducible.

---

### Meta-Review · Area_Chair_D634 · 2025-12-22

**Summary:**

The reviewers have concerns about novelty of the method compared to previous EDL extensions, the completeness and scale of the experiments (architectures/datasets/runtime), reproducibility, ablation studies, and the choices of evaluation metrics. Most of them were addressed during the rebuttal through additional experiments or by clarifying that the details are already in the appendix.

**Reviewer Concerns:**

The concerns about reproducibility, experimental completeness, missing baselines, and ablation studies were largely addressed by additional experiments and by pointing to details in the appendix. The concerns about novelty of the method and breadth of model architectures (still no ViT-based evaluations) were partially addressed. Some concerns remain unaddressed, such as lack of additional metrics (ECE, Brier score), evaluation on large scale datasets (Imagenet level).

**Reviewer Scores:**

Reviewer g4uc, y7KH, and odmc are likely to keep the positive scores. Reviewer jVfx may increase the score as additional comparison with RED, scalability analyses and TTA relationship clarification were provided. Reviewer z85G may or may not increase the score as the concerns were partially addressed.

---

### Decision · Program_Chairs · 2026-01-26

Accept (Poster)